# Context is Key: A Benchmark for Forecasting with Essential Textual Information

[★]Andrew R. Williams [1 2 3]   [★]Arjun Ashok [1 2 3]
[†]Étienne Marcotte [1]   [†]Valentina Zantedeschi [1 4]   Jithendaraa Subramanian [1 2 5]   Roland Riachi [2]
James Requeima [6]   Alexandre Lacoste [1]   Irina Rish [2 3]   [†]Nicolas Chapados [1 2 7]   [†]Alexandre Drouin [1 2 4]

## Abstract

Forecasting is a critical task in decision-making across numerous domains. While historical numerical data provide a start, they fail to convey the complete context for reliable and accurate predictions. Human forecasters frequently rely on additional information, such as background knowledge and constraints, which can efficiently be communicated through natural language. However, in spite of recent progress with LLM-based forecasters, their ability to effectively integrate this textual information remains an open question. To address this, we introduce "Context is Key" (CiK), a time series forecasting benchmark that pairs numerical data with diverse types of carefully crafted textual context, requiring models to integrate both modalities; crucially, every task in CiK requires understanding textual context to be solved successfully. We evaluate a range of approaches, including statistical models, time series foundation models, and LLM-based forecasters, and propose a simple yet effective LLM prompting method that outperforms all other tested methods on our benchmark. Our experiments highlight the importance of incorporating contextual information, demonstrate surprising performance when using LLM-based forecasting models, and also reveal some of their critical shortcomings. This benchmark aims to advance multimodal forecasting by promoting models that are both accurate and accessible to decision-makers with varied technical expertise. The benchmark can be visualized at https://servicenow.github.io/context-is-key-forecasting/v0/.

[★]Equal contribution; order decided by coin toss. [†]Core contributors. [1]ServiceNow Research [2]Mila - Québec AI Institute [3]Université de Montréal [4]Université Laval [5]McGill University [6]University of Toronto [7]Polytechnique Montréal. Correspondence to: Andrew R. Williams <andrew.williams1@servicenow.com>, Arjun Ashok <arjun.ashok@servicenow.com>.

*Proceedings of the 42nd International Conference on Machine Learning*, Vancouver, Canada. PMLR 267, 2025. Copyright 2025 by the author(s).

## 1. Introduction

The prediction of future states of the world is a cornerstone of decision making (Hyndman & Athanasopoulos, 2018) and intelligence (Wang, 2019). Articulated as time series forecasting, this problem pervades much of science and commerce.

Accurate forecasting relies on several decisions up to the practitioner (Hyndman & Athanasopoulos, 2018): 1. *Model selection*: choosing an appropriate forecasting model for a given problem, and 2. *Incorporating prior information*: determining what relevant information to integrate into the model and how to do so effectively. This involves decisions about statistical priors, inductive biases in the model architecture, and other forms of domain knowledge integration, all of which traditionally rely on expert knowledge and manual intervention. However, recent advancements in machine learning have shown promise in automating both model selection and the incorporation of prior information, accelerating the democratization of time series forecasting.

In the wake of the foundation model paradigm shift (Bommasani et al., 2021), several works such as Liang et al. (2024); Chen et al. (2023); Lim & Zohren (2021), have addressed automatic model selection by learning flexible, adaptable models applicable across various problem scenarios. However, these approaches are much more costly than traditional statistical methods, and provide debatable improvements in performance (Garza & Mergenthaler-Canseco, 2024). Typically, these models process purely numerical time series, excluding the context that human forecasters rely on to incorporate prior information.

An alternative class of recent approaches (Jin et al., 2024; Liu et al., 2024c; Requeima et al., 2024) adapt large language models (LLMs) for forecasting and leverage natural language (NL) as an intuitive interface to integrate side information. These methods overcome a significant limitation of traditional forecasting techniques by eliminating the need to manually encode priors or design specialized models. They further hold the promise of capturing a broader range of prior knowledge and context, potentially leading to more comprehensive and accurate forecasts.

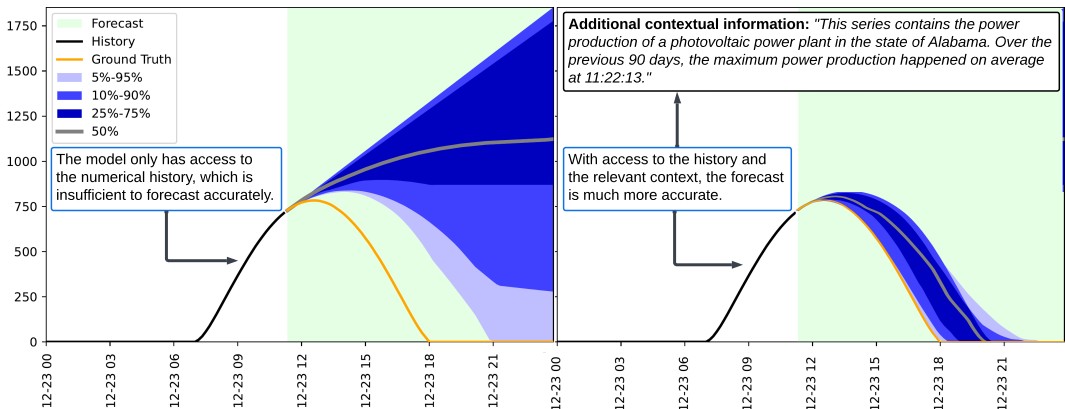

*Figure 1.* An example task from the proposed Context is Key (CiK) benchmark with `GPT-4o` forecasts in blue and the ground truth in yellow. **Left:** Forecasts based on the numerical history alone are inaccurate, as nothing indicates a reversion to zero. **Right:** The context enables better forecasts because it reveals that the series represents photovoltaic power production. Hence, the model can deduce that no power will be produced at night. The context also enables better estimation of the peak hour of production by providing statistics from the history.

Unfortunately, there are as of yet no systematic evaluations of these models' abilities to jointly leverage historical observations and natural language for forecasting. While several benchmarks for context-aided forecasting have been recently released (Zhang et al., 2023; Liu et al., 2024a; Xu et al., 2024; Emami et al., 2024; Merrill et al., 2024), their contexts are not guaranteed to be useful for improving performance. As such, it remains unknown whether existing models can enhance their forecasts by utilizing crucially relevant textual context.

To this end, we propose the Context is Key (CiK, pronounced *kick*) benchmark of forecasting tasks. CiK consists of tasks designed to assess a forecasting model's ability to use both 1. numerical input-output pairs and 2. essential textual context. As shown in Figure 1, accurate forecasts in CiK are made possible only by effectively leveraging both numerical data and key information contained within the accompanying text.

Our contributions are:

- *CiK Benchmark*: A collection of 71 manually designed forecasting tasks spanning seven real-world domains, each requiring the integration of diverse contextual information that has a non-trivial impact on forecasts (Sec. 3).
- *Region of Interest CRPS (RCRPS)*: A scoring rule to evaluate context-aided forecasting performance, which prioritizes context-sensitive windows and accounts for constraint satisfaction (Sec. 4).
- *Direct Prompt Forecasters*: A simple yet effective prompt-based approach to using LLMs as context-aided forecasters, which serves as a surprisingly strong baseline on CiK (Sec. 5.2).
- *Extensive evaluation* of diverse models on CiK, including statistical models, time series foundation models using only numerical data, and LLM-based forecasters capable of incorporating natural language context. Our analysis explores key factors such as the impact of context conditioning, tradeoffs in model size and performance, and discusses failure modes of models (Sec. 5).

## 2. Problem Setting

**Context-Aided Forecasting** This work addresses the problem of *context-aided forecasting*, where the goal is to produce statistical forecasts by incorporating relevant side information (i.e., context). Let $\mathbf{X}_H = [X_1, \ldots, X_t]$ represent a series of random variables corresponding to historical observations in discrete time, where each $X_\tau \in \mathcal{X} \subseteq \mathbb{R}$, and let $\mathbf{X}_F = [X_{t+1}, \ldots, X_T]$ represent future observations. In classical statistical forecasting, the goal is to estimate the joint distribution of future observations given the historical ones, $P(\mathbf{X}_F \mid \mathbf{X}_H)$. We further assume access to *context*, denoted $\mathbf{C}$, which consists of additional data of arbitrary nature containing information relevant for predicting $\mathbf{X}_F$ and complementary to the history $\mathbf{X}_H$. The task then becomes estimating the distribution $P(\mathbf{X}_F \mid \mathbf{X}_H, \mathbf{C})$. Crucially, we restrict our focus to *relevant context*, which we define as context that does not degrade forecasts. Formally, for $\mathbf{x}_F \sim \mathbf{X}_F \mid \mathbf{X}_H, \mathbf{C}$, given some loss function $\mathcal{L}$ assessing a predictive distribution over $\mathbf{X}_F$ against a realization $\mathbf{x}_F$, $\mathcal{L} : P(\mathbf{X}_F) \times \mathbf{x}_F \to \mathbb{R}$, we are interested in models where, in expectation, forecasts that leverage context perform better:

$$\mathbb{E}_{\mathbf{x}_F} \mathcal{L}(P(\mathbf{X}_F \mid \mathbf{X}_H, \mathbf{C}), \mathbf{x}_F) \leq \mathbb{E}_{\mathbf{x}_F} \mathcal{L}(P(\mathbf{X}_F \mid \mathbf{X}_H), \mathbf{x}_F).$$

Furthermore, although the nature of the context can vary widely, we specifically concentrate on *context communicated through natural language*, which we refer to as "context" or "contextual information".

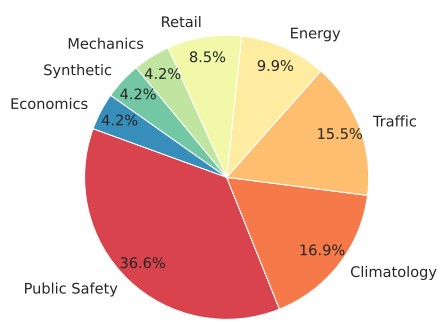

Figure 2. The tasks in the CiK benchmark rely on real-world numerical data from 7 domains.

# 3. Context is Key: a Natural Language Context-Aided Forecasting Benchmark

We introduce the *Context is Key* (CiK) benchmark, a collection of *probabilistic forecasting* tasks where accurate predictions require integrating both numerical data and natural language contextual information. CiK comprises 71 distinct tasks spanning seven real-world application domains (see Sec. 3.1), each featuring various stochastic components that can be instantiated into thousands of task instances (e.g., time series, time windows, natural language formulation). These tasks encompass diverse types of contextual information that reveal various aspects of dynamical processes (see Sec. 3.2). Moreover, they are designed such that *context is key* in that it non-trivially *unlocks* accurate forecasts, e.g., by conveying causal relationships that reveal the effect of a covariate on the time series of interest. An example task is shown in Figure 1 and others can be found in Appendix B.

**Availability:** CiK is open source. The complete set of tasks can be explored at https://servicenow.github.io/context-is-key-forecasting/v0/ and the source code, at https://github.com/ServiceNow/context-is-key-forecasting. All data sources used for CiK's tasks are openly available (see Appendix A.1).

## 3.1. Domains and Numerical Data Sources

As illustrated in Figure 2, the majority (95%) of tasks in CiK are based on real-world application domains. We leverage 2,644 time series sourced from publicly available datasets across seven domains: Climatology (solar irradiance and cloud coverage (Sengupta et al., 2018)); Economics (unemployment rates across states and counties (U.S. Bureau of Labor Statistics, 2024)); Energy (electricity consumption and production (Godahewa et al., 2021)); Mechanics (experimental properties of physical systems (Gamella et al., 2024)); Public Safety (fire department intervention counts (Ville de Montréal, 2020)); Transportation (highway segment occupancy rates and average speeds (Chen et al., 2001)); and Retail (cash withdrawals from various ATMs (Godahewa et al., 2021)). The remaining 5% of tasks use simulated data from dynamical systems crafted specifi-

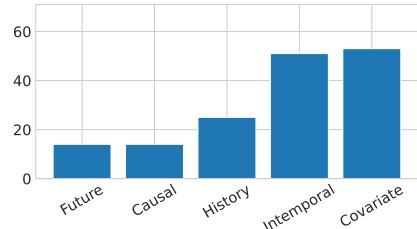

Figure 3. Number of tasks per context type in CiK.

cally for the tasks. Overall, the time series in CiK exhibit diverse sampling frequencies, with observations ranging from every 10 minutes to monthly intervals; additional details on data sources can be found in Appendix A.1.

**Memorization mitigation:** Building tasks with publicly available data introduces contamination risk: pretrained LLMs and time series foundation models may have memorized portions of the data, potentially inflating evaluation performance. We employ several mitigation strategies. First, we prioritize live data sources that are continuously updated, such as Chen et al. (2001) and Ville de Montréal (2020), ensuring the data is collected after the training cut-off dates of the models that we evaluate. Second, where applicable, we use derived series that are not directly available in the raw data, such as incident logs converted into time series (Ville de Montréal, 2020). Finally, as a last resort, we apply minor transformations, such as adding noise or shifting timestamps, but use these sparingly to limit their potential impact on the tasks, and to avoid misalignment between common-sense knowledge (e.g., holiday dates) and the numerical data. The exact mitigation methods used, per data source, together with the number of tasks on which they are applied are given in Appendix A.1.

## 3.2. Natural Language Context

For each task in the benchmark, we jointly sample numerical data from one of the series described in Sec. 3.1 and then *manually* craft the natural language context necessary to unlock accurate forecasts. In some cases, this context is purely descriptive, providing information about the general nature of the target variable and its historical behavior, as seen in the task illustrated in Figure 1. In other cases, the raw numerical data is adjusted to reflect the influence of the context. For example, in one task based on data from Godahewa et al. (2021), an ATM is expected to be inaccessible during a specific time period in the future, leading to zero withdrawals (see Appendix B.3). In another task, electricity demand is projected to surge due to an incoming weather event (see Appendix B.2). For such cases, we modify the series to incorporate patterns described by the context.

Overall, we include diverse forms of natural language context, each capturing a different aspect of the process underlying the time series and providing complementary knowledge that a human expert could leverage for more accurate

forecasting. The types of context are described below and exemplified in the task illustrated in Figure 4. For additional clarity, further examples are provided in Appendix B and the distribution of tasks per context type is shown in Figure 3.

**Intemporal information ($c_I$)** Information about the process that remains invariant in time. For example, a description of the process and the nature of the target variable, as in Figure 4 (point ②). This includes patterns that cannot be inferred from the available numerical data, such as long-period seasonalities, or constraints on values, such as positivity.

**Future information ($c_F$)** Information about the future behavior of the time series. For example, a scenario to be simulated as in Figure 4 point ③, or expected events along with any entailed constraints, such as an inventory shortage restricting future sales amounts.

**Historical information ($c_H$)** Information about the past behavior of the series that the available numerical history does not reveal. For example, statistics on past values of the series, as in Figure 4 (point ④), or an explanation for spurious patterns that should be be disregarded at inference, such as periodic anomalies caused by sensor maintenance.

**Covariate information ($c_{cov}$)** Information about additional variables that are statistically associated with the variable of interest. For example, a series correlated with the target values (as in Figure 4 point ⑤).

**Causal information ($c_{causal}$)** Information about causal relationships between covariates and the target variable. For example, if the covariates are known to cause or are confounded with the target variable, as in Figure 4 point ⑥.

Finally, for completeness, Appendix A.7 provides the distributions of lengths of the numerical historical data, prediction horizons and natural language context.

### 3.3. Validating the Relevance of The Context

Related efforts on context-aided forecasting, outlined in Sec. 6, rely on scraping and/or LLMs to obtain natural language context (Zhang et al., 2023; Merrill et al., 2024; Liu et al., 2024a; Emami et al., 2024). In contrast, to ensure both the quality and relevance of tasks in CiK, we manually craft all contextual information and associated data transformations according to the procedure described in Appendix A.2. To validate the importance of the context, we subject each task to review by a panel of human and LLM evaluators tasked with assessing if the context enables better forecasts. The results are overwhelmingly positive, with humans finding the context to be relevant for 95% of evaluated instances (details in Appendix A.4).

## 4. Region of Interest CRPS

Alongside the tasks, we introduce the Region of Interest Continuous Ranked Probability Score (RCRPS), a novel

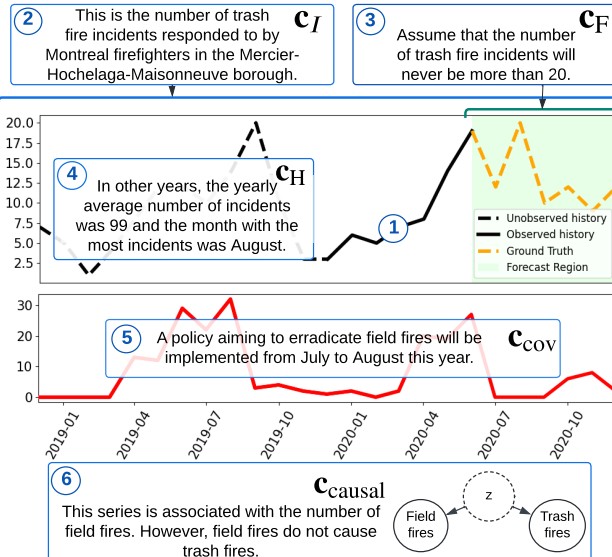

*Figure 4.* Illustration of a CiK task annotated with types of natural language context: ① The short numerical history is misleading, suggesting an increasing trend. However, contextual information compensates and enables accurate forecasts: ② The intemporal information ($c_I$) reveals the nature of the series, implying a seasonal pattern with greater prevalence in the summer months due to weather. ③ The future information ($c_F$) reveals that the series cannot continue its increasing trend. ④ The historical information ($c_H$) complements the short history by providing high-level statistics on past values. ⑤ The covariate information ($c_{cov}$) reveals an association with another quantity: field fires. Could its behavior impacts future values of the target series? ⑥ No, the causal information ($c_{causal}$) provides the answer.

proper scoring rule designed specifically for context-aided probabilistic forecasting. This new scoring rule is an extension of the Continuous Ranked Probability Score (CRPS; Gneiting & Raftery (2007)), a proper scoring rule that provides a comprehensive assessment of forecast quality by evaluating the entire predictive distribution rather than point forecasts. Since it is based on the CRPS, the RCRPS can be calculated using only samples from the predictive distribution, and so can be used even in cases where closed-form distributions are unavailable. The RCRPS extends the CRPS via two key components: a *region of interest* and a measure of *constraint satisfaction*.

**Region of interest (RoI):** The RCRPS assigns more weight to errors in a task's RoI, which is a subset of time steps $\mathcal{I} \subseteq [t+1, \ldots, T]$ for which the context is particularly relevant. For example, in the ATM task from Sec. 3.2 (visualized in Appendix B.3), the RoI denotes the time steps during which the ATM is expected to be unavailable. In other tasks, such as those in Figures 1 and 4, where the context informs the value of all future time points, the RCRPS assigns equal weights to all time steps (for readability, we report the definition of RCRPS for this special case in Ap-

*Table 1.* Results of selected models on the CiK benchmark. Starting from the left, the first column shows the RCRPS averaged over all tasks. The second column shows the rank of each method w.r.t. other models, averaged over all tasks. The remaining columns show the average RCRPS stratified by types of context (Sec. 3.2). All averages are weighted according to the scheme described in Sec. 5.1 and accompanied by standard errors. Lower is better and the best averages are in bold. An asterisk (*) denotes models that do not use natural language context. For results on all models and with alternative aggregation strategies, see Appendix C.

| MODEL | AVERAGE RCRPS | AVERAGE RANK | INTEMPORAL INFORMATION | HISTORICAL INFORMATION | FUTURE INFORMATION | COVARIATE INFORMATION | CAUSAL INFORMATION |
|---|---|---|---|---|---|---|---|
| **DIRECT PROMPT (ours)** | | | | | | | |
| Llama-3.1-405B-Inst | **0.159 ± 0.008** | **4.516 ± 0.233** | **0.174 ± 0.010** | 0.146 ± 0.001 | **0.075 ± 0.005** | **0.164 ± 0.010** | 0.398 ± 0.045 |
| Llama-3-70B-Inst | 0.286 ± 0.004 | 7.803 ± 0.106 | 0.336 ± 0.006 | 0.180 ± 0.003 | 0.194 ± 0.006 | 0.228 ± 0.004 | 0.629 ± 0.019 |
| Mixtral-8x7B-Inst | 0.523 ± 0.023 | 14.473 ± 0.147 | 0.723 ± 0.037 | 0.236 ± 0.002 | 0.241 ± 0.001 | 0.359 ± 0.028 | 0.875 ± 0.128 |
| Qwen-2.5-7B-Inst | 0.290 ± 0.003 | 11.330 ± 0.253 | 0.290 ± 0.004 | 0.176 ± 0.003 | 0.287 ± 0.007 | 0.240 ± 0.002 | 0.525 ± 0.003 |
| Qwen-2.5-0.5B-Inst | 0.463 ± 0.012 | 12.694 ± 0.173 | 0.609 ± 0.019 | 0.165 ± 0.004 | 0.218 ± 0.012 | 0.476 ± 0.015 | 0.429 ± 0.006 |
| GPT-4o | 0.274 ± 0.010 | **4.381 ± 0.159** | 0.218 ± 0.007 | **0.118 ± 0.001** | 0.121 ± 0.001 | 0.250 ± 0.011 | 0.858 ± 0.053 |
| GPT-4o-mini | 0.354 ± 0.022 | 9.056 ± 0.194 | 0.475 ± 0.035 | 0.139 ± 0.002 | 0.143 ± 0.002 | 0.341 ± 0.028 | 0.644 ± 0.128 |
| **LLMP** | | | | | | | |
| Llama-3-70B-Inst | 0.539 ± 0.013 | 8.243 ± 0.231 | 0.438 ± 0.017 | 0.516 ± 0.028 | 0.847 ± 0.024 | 0.546 ± 0.016 | 0.392 ± 0.028 |
| Llama-3-70B | 0.236 ± 0.006 | 6.522 ± 0.244 | 0.212 ± 0.005 | 0.121 ± 0.008 | 0.299 ± 0.017 | 0.193 ± 0.004 | **0.360 ± 0.011** |
| Mixtral-8x7B-Inst | 0.264 ± 0.004 | 8.519 ± 0.264 | 0.242 ± 0.007 | 0.173 ± 0.004 | 0.324 ± 0.005 | 0.219 ± 0.005 | 0.437 ± 0.007 |
| Mixtral-8x7B | 0.262 ± 0.008 | 8.540 ± 0.198 | 0.250 ± 0.008 | **0.119 ± 0.003** | 0.310 ± 0.019 | 0.229 ± 0.006 | 0.457 ± 0.011 |
| Qwen-2.5-7B-Inst | 1.974 ± 0.027 | 18.443 ± 0.276 | 2.509 ± 0.044 | 2.857 ± 0.056 | 1.653 ± 0.008 | 1.702 ± 0.035 | 1.333 ± 0.144 |
| Qwen-2.5-7B | 0.910 ± 0.037 | 16.051 ± 0.341 | 1.149 ± 0.047 | 1.002 ± 0.053 | 0.601 ± 0.071 | 0.639 ± 0.047 | 0.928 ± 0.129 |
| Qwen-2.5-0.5B-Inst | 1.937 ± 0.024 | 20.136 ± 0.191 | 2.444 ± 0.038 | 1.960 ± 0.063 | 1.443 ± 0.010 | 1.805 ± 0.030 | 1.199 ± 0.129 |
| Qwen-2.5-0.5B | 1.995 ± 0.024 | 19.686 ± 0.275 | 2.546 ± 0.039 | 2.083 ± 0.052 | 1.579 ± 0.015 | 1.821 ± 0.030 | 1.225 ± 0.128 |
| **MULTIMODAL MODELS** | | | | | | | |
| UniTime | 0.370 ± 0.001 | 14.675 ± 0.091 | 0.457 ± 0.002 | 0.155 ± 0.000 | 0.194 ± 0.003 | 0.395 ± 0.001 | 0.423 ± 0.001 |
| Time-LLM (ETTh1) | 0.476 ± 0.001 | 17.932 ± 0.075 | 0.518 ± 0.002 | 0.183 ± 0.000 | 0.403 ± 0.002 | 0.441 ± 0.001 | 0.482 ± 0.001 |
| **TS FOUNDATION MODELS\*** | | | | | | | |
| Lag-Llama | 0.327 ± 0.004 | 13.370 ± 0.233 | 0.330 ± 0.005 | 0.167 ± 0.005 | 0.292 ± 0.009 | 0.294 ± 0.004 | 0.495 ± 0.014 |
| Chronos-Large | 0.326 ± 0.002 | 12.298 ± 0.148 | 0.314 ± 0.002 | 0.179 ± 0.003 | 0.379 ± 0.003 | 0.255 ± 0.002 | 0.460 ± 0.004 |
| TimeGEN | 0.353 ± 0.000 | 15.047 ± 0.095 | 0.332 ± 0.000 | 0.177 ± 0.000 | 0.405 ± 0.000 | 0.292 ± 0.000 | 0.474 ± 0.000 |
| Moirai-Large | 0.520 ± 0.006 | 12.873 ± 0.263 | 0.596 ± 0.009 | 0.140 ± 0.001 | 0.431 ± 0.002 | 0.499 ± 0.007 | 0.438 ± 0.011 |
| **STATISTICAL MODELS\*** | | | | | | | |
| ARIMA | 0.475 ± 0.006 | 12.721 ± 0.167 | 0.557 ± 0.009 | 0.200 ± 0.007 | 0.350 ± 0.003 | 0.375 ± 0.006 | 0.440 ± 0.011 |
| ETS | 0.530 ± 0.009 | 15.001 ± 0.198 | 0.639 ± 0.014 | 0.362 ± 0.014 | 0.315 ± 0.006 | 0.402 ± 0.010 | 0.508 ± 0.017 |
| Exp-Smoothing | 0.605 ± 0.013 | 15.689 ± 0.152 | 0.702 ± 0.020 | 0.493 ± 0.016 | 0.397 ± 0.006 | 0.480 ± 0.015 | 0.827 ± 0.060 |

pendix E).

**Constraint satisfaction:** The RCRPS penalizes constraint violations according to a task-specific function $v_{\mathbf{C}}$ whose value is positive for any trajectory that violates the constraints. Concrete examples are given in Appendix E.4.

Given an inferred forecast distribution $\widetilde{\mathbf{X}}_F$ and a ground truth $\mathbf{x}_F$, the scoring rule is defined as:

$$\text{RCRPS}(\widetilde{\mathbf{X}}_F, \mathbf{x}_F) := \alpha \cdot \left[ \frac{1}{2|\mathcal{I}|} \sum_{i \in \mathcal{I}} \text{CRPS}(\widetilde{X}_i, x_i) + \right.$$

$$\left. \frac{1}{2|\neg\mathcal{I}|} \sum_{i \in \neg\mathcal{I}} \text{CRPS}(\widetilde{X}_i, x_i) + \beta \cdot \text{CRPS}(v_{\mathbf{C}}(\widetilde{\mathbf{X}}_F), 0) \right],$$

where the terms respectively account for the CRPS inside the RoI, the CRPS outside of the RoI, and the constraint violation penalty. The last term, which is inspired by the threshold-weighted CRPS of Gneiting & Ranjan (2011), vanishes when all constraints are satisfied. The $\alpha$ term is a task-dependent normalization factor to make the RCRPS scale-independent, which enables fair RCRPS aggregation

across tasks; its calculation is described in Appendix E.1. Finally, $\beta$ is a scaling factor that controls the impact of constraint violation on the score; we use $\beta = 10$ in our experiments. We refer the reader to Appendix E for additional details.

## 5. Experiments and Results

In this section, we define our evaluation protocol (Sec. 5.1) and outline the models that we evaluate on CiK (Sec. 5.2). We then present results on the benchmark (Sec. 5.3), along with an analysis of factors affecting model performance. Finally, we look at areas for improvement by analyzing forecasting errors (Sec. 5.4) and inference cost (Sec. 5.5).

### 5.1. Evaluation Protocol

Each task in CiK has many unique instances arising from the selection of time series and windows in the associated numerical data, as well as minor variations in natural language context. We deterministically sample five instances of each task in order to make the evaluation reproducible and affordable. For every instance, we generate 25 independent

forecasts per model for evaluation. Many of the tasks in the benchmark share similarities due to sharing the same data sources or using variants of the same context. Therefore, we identify clusters of similar tasks and design a weighting scheme such that each cluster has equal total weight in our aggregate score (see Appendix A.5 for more details).

## 5.2. Methods

We evaluate a wide variety of models including methods based on LLMs, state-of-the-art numerical time series foundation models (TSFMs) and classical statistical forecasting methods. Since CiK is meant to be an evaluation benchmark, it does not have a corresponding training set. We therefore only evaluate models that can produce forecasts directly based on the history of a series. This includes LLMs and TSFMs that support zero-shot inference, and traditional statistical models that can be fit directly to the history of a given instance. We outline these methods below and refer the reader to Appendix D for additional details.

**LLM-based Forecasters:** We consider two prompt-based approaches. We propose "DIRECT PROMPT", a simple approach where we instruct the model to directly output a forecast as a structured output for all of the required timestamps (see Appendix D.1 for more details). We also evaluate LLM Processes (LLMP; Requeima et al. (2024)), a method which autoregressively prompts the LLM multiple times to output a forecast. For each of these, we evaluate a variety of LLMs with diverse architectures and sizes, such as `GPT-4o`, `GPT-4o-mini` (Achiam et al., 2023), `Qwen-2.5-{0.5B, 1.5B, 7B}` (Yang et al., 2024), `Mixtral-8x7B` (Jiang et al., 2024), `Llama-3-{8B, 70B}` (Dubey et al., 2024), `Llama-3.1-405B` (Dubey et al., 2024).[1] We also evaluate multimodal forecasting models, `ChatTime` (Wang et al., 2025) (zero-shot), `UniTime` (Liu et al., 2024c) and `Time-LLM` (ETTh1) (Jin et al., 2024) (trained according to their respective authors' guidelines) (details in Appendix D.4). For all of these approaches, inference is performed zero-shot on the benchmark and we compare their performance with and without the natural language context.

**Quantitative Forecasting Models:** We also evaluate a number of models that process only numerical data, but cannot integrate natural language. We evaluate `Exponential Smoothing` (Gardner Jr. (1985)), `ETS` (Hyndman et al., 2008), and `ARIMA` (Box et al., 2015), three simple, but time-tested statistical approaches. We also evaluate four state-of-the-art TSFMs: `Lag-Llama` (Rasul et al., 2023), `Chronos` (Ansari et al., 2024), `Moirai` (Woo et al., 2024) and `TimeGEN` (Garza et al., 2023). We fit `Exponential Smoothing`, `ETS` and `ARIMA` to the individual numerical history of each

---

[1]For LLMP, we do not consider `Llama-3.1-405B` and `GPT` models as LLMP requires loading model weights into memory, which is infeasible due to resource limitations and confidentiality of the respective models (see Appendix D.2.2 for more details).

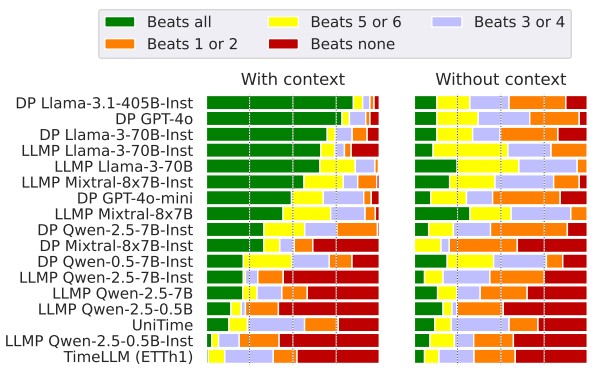

*Figure 5.* Proportion of tasks for which LLM-based methods outperform the 7 quantitative forecasting methods (see Sec. 5.2). A method is considered to outperform another on a task if its average RCPRS is lower on said task. Results are shown for variants that use (left) and do not use (right) the natural language context. A full green bar would indicate that the method is better on all tasks, whereas a full red bar would indicate that it is worse everywhere. Tasks are weighted according to Sec. 5.1.

task instance, which is the same input that the TSFMs process to produce forecasts zero-shot.

## 5.3. Results on CiK

Tab. 1 shows our main results. At a high level, we observe that the best-performing methods combine pretrained LLMs with prompting strategies like DIRECT PROMPT and LLMP, with a bias towards the largest models. In terms of RCRPS, `Llama-3.1-405B-Inst` (DIRECT PROMPT) significantly outperforms all of its counterparts. `GPT-4o` (DIRECT PROMPT) performs worse with respect to RCRPS, but compares favorably in terms of average rank. This discrepancy is due to strong failures on some of the tasks, which we discuss in Sec. 5.4. Other models like `Llama-3-70B` (LLMP), `Mixtral-8x7B-Inst` (LLMP) and `Mixtral-8x7B` (LLMP) are on par with `GPT-4o` (DIRECT PROMPT), `Llama-3-70B-Inst` (DIRECT PROMPT) and `Qwen-2.5-7B-Inst` (DIRECT PROMPT) in terms of RCRPS. Interestingly, all of these methods outperform `UniTime` and `Time-LLM`, which also rely on LLMs (GPT-2 & LLaMA-7B). We discuss this gap in Appendix D.4. Finally, as emphasized in Figure 5, we observe that the best-performing LLM methods significantly outperform purely quantitative models. In what follows, we examine various aspects of these results (and refer to Appendix C for additional results).

**Explaining the performance of LLM-based approaches:** The strong performance of LLM-based methods could be due to two factors: (i) properly leveraging the natural language context and (ii) being more proficient at numerical forecasting. We thus aim to disentangle their contributions.

On the one hand, Figure 6 shows clear evidence that methods with access to the context improve their forecasts. For

example, `Llama-3.1-405B-Inst` (DIRECT PROMPT) improves by 67.1% with context. We find these differences statistically significant across many of the models (see Appendix C.6). This is reflected in the quality of the example forecasts in Appendix C.11, where we observe clear improvements in regions of interest, as well as improved constraint satisfaction. Other models show lesser improvements and, in some cases, even a degradation in performance. Our analysis in Sec. 5.4 shows that this can be explained either by the context being ignored, or by significant failures in using context, worsening overall performance.

On the other hand, Figure 5 (right) shows that LLM-based forecasters, when evaluated *without context*, no longer dominate the quantitative forecasting models. However, some LLM-based forecasters remain surprisingly competitive. For instance, multiple `Llama-3` (LLMP) models outperform at least 5 of the quantitative models on the majority of tasks. The extended results in Appendix C.1 further substantiate this. In contrast, models such as `Llama-3.1-405B-Inst` (DIRECT PROMPT) and `GPT-4o` (DIRECT PROMPT) show significantly weaker forecasting performance without context. This suggests that such models are especially preferable in cases where context is available. This is also reflected in their aggregate scores without context (in Tab. 3).

**Comparing** LLMP **and** DIRECT PROMPT: Figure 5 (right) shows that without context, LLMP models exhibit stronger numerical forecasting performance than DIRECT PROMPT models. This advantage likely stems from LLMP 's closer alignment with the forecasting task: LLMP simply prompts the LLM to autoregressively predict the next value in the time series, a task well suited to base models with no instruction tuning. In contrast, DIRECT PROMPT requires output forecasts to be structured, which relatively complicates the task.

This line of reasoning leads us to another observation on the impact of instruction tuning; as reflected in Tab. 1 and Figure 5, instruction tuning appears to generally degrade LLMP performance. `Llama-3` models show a twofold decrease in performance after instruction tuning, a behavior previously observed by Gruver et al. (2024). Interestingly, instruction tuning does not degrade the performance of `Mixtral-8x7B`. Finally, whereas LLMP mostly suffers from instruction tuning, DIRECT PROMPT requires forecasts to be produced in a specific structure, a skill that instruction-tuned models might be better at (see Appendix D.1.1 for details).

**No Method Excels Across All Context Types:** Tab. 1 shows that some methods can effectively produce better forecasts with the provided contextual information. However, no single method is the best across context types. `Llama-3.1-405B-Inst` (DIRECT PROMPT), the top-performing method on average, outperforms its counterparts on only 3 out of 5 context types. This finding indicates that the

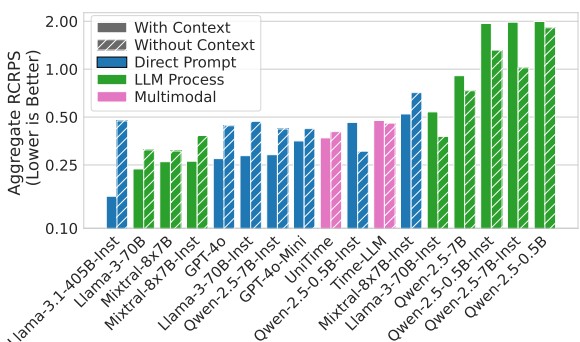

*Figure 6.* RCRPS with and without context (log scale, lower is better). Full bars show performance with context; striped bars show performance without context. In general, larger models outperform smaller models and benefit much more from context. DIRECT PROMPT models all improve with context, other than `Qwen-2.5-0.5B-Instruct`. For LLMP, larger models benefit from context, but smaller models fail to do so and perform worse in general.

benchmark remains unsolved, leaving significant room for advancements from the research community.

### 5.4. Error Analysis

Foundation models are known to make mistakes or return factually inaccurate information (Bommasani et al., 2021). We find that models occasionally return forecasts that miss the ground truth by a large margin. We use the term *significant failure* to denote a forecast that over or undershoots by at least 500% of the range of the ground truth; we clip the RCRPS of such instances to 5 to avoid them disproportionately skewing the aggregate score. Despite this, such significant failures impact the results in Tab. 1: `GPT-4o` with DIRECT PROMPT, while emerging as a top-performer in most tasks (as reflected in its average rank), has a significantly higher aggregate RCRPS than models ranked worse, such as `Mixtral-8x7B` with LLMP. As an example, `GPT-4o` with DIRECT PROMPT fails significantly in a task with a context involving scientific notation (see Figure 25 and more examples in Appendix C.12). Notably, while a model may generally achieve a high win rate, a few significant failures can dominate its aggregate performance, as observed for `GPT-4o` with DIRECT PROMPT. In the case of `Qwen-2.5-0.5B-Inst` with DIRECT PROMPT, this leads to an aggregate RCRPS that is worse with context than without. We analyze this in detail in Appendix C.10. These findings underscore the need for future work to develop more robust models that can handle context effectively while avoiding significant failures.

### 5.5. Inference Cost

A key practical aspect for forecasting applications is the inference time of models and their associated cost. Figure 7 shows that, while `Llama-3.1-405B-Instruct` performs the best on average, it comes at the cost of a significantly higher parameter count than the quantitative forecasters.

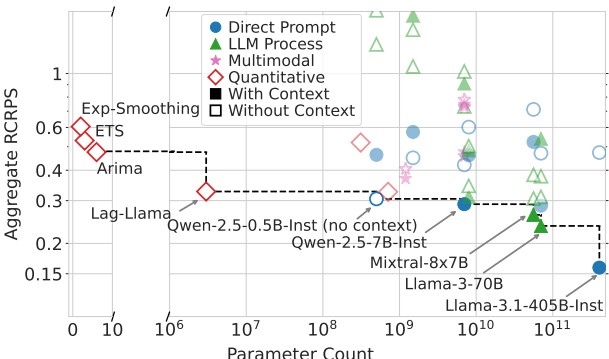

*Figure 7.* Comparison of average RCRPS (per Tab. 1), vs. the parameter count of each method(lower is better for both). The GPT family, as well as TimeGEN, are left out as there is no information on them about parameter count. The dashed line illustrates the Pareto front: models above and to the right of this line are dominated. Quantitative forecasters dominate the low-parameter regime, while LLM-based methods such as `Qwen-2.5-7B-Inst` (DIRECT PROMPT), `Llama-3-70B` (LLMP) and `Llama-3.1-405B-Inst` (DIRECT PROMPT) offer superior performance for a higher parameter count.

This emphasizes that while LLMs can be powerful context-aware forecasters, they come with steep computational costs, which highlights the need for efficient models that balance both accuracy and resource demands. Of note is also that many LLM-based forecasters are pareto-dominated by quantitative forecasters such as `Lag-Llama` and `Chronos`. This suggests that, beyond the ability to process text, a careful choice of prompting strategy and LLM is crucial for Pareto efficiency. Further, due to their high parameter count, the LLMs have inference times that are orders of magnitude longer than quantitative forecasters, which are far more efficient for sustained usage (see Figure 13 for a comparison). LLMs require significant computational power, making them unsuitable for real-world practical forecasting at scale where speed and cost matter. However, additional research could improve the cost-effectiveness of context-aided forecasting models to match that of traditional models.

## 6. Related Work

We review two streams of related work: (i) works that introduce related benchmarks and datasets, and (ii) works that repurpose LLMs for forecasting.

**Benchmarks and Datasets** Merrill et al. (2024) present a benchmark designed to evaluate LLMs' ability to reason about time series, with context-aided forecasting as one assessed capability. They focus on purely synthetic time series, which may not accurately reflect real-world dynamics, whereas our benchmark is based primarily on real-world data. Further, their evaluation is limited to point forecasting metrics, which do not measure the quality of the full forecast distribution. In contrast, we adopt prob-abilistic forecasting metrics, e.g., the continuous ranked probability score (CRPS; *c.f.* Gneiting & Raftery, 2007), to assess the quality of entire forecast distributions. Other related datasets include Time-MMD (Liu et al., 2024a), which integrates text extracted from reports and web searches, TGTSF (Xu et al., 2024), which incorporates information such as weather reports and news articles, SysCaps (Emami et al., 2024), which includes LLM-generated descriptions of building energy consumption systems, TS-Insights (Zhang et al., 2023), which includes LLM-generated descriptions of trends and seasonalities, and Dual-Forecaster (Wu et al., 2025) where time series datasets are captioned with trend and seasonality information. Several works (Sawhney et al., 2021; Liu et al., 2024b; Wang et al., 2024b;a; Kim et al., 2024) automatically construct datasets of paired textual and numerical information. The key distinction between these works and ours lies in the importance of the textual information: while in the above works, the text is not guaranteed to be essential for high-quality forecasts, in CiK, all tasks are handcrafted to ensure that accurate forecasts *cannot be achieved* without relying on the provided natural language context, thereby making it a high-quality evaluation benchmark for context-aided forecasting.

**Repurposing LLMs for Forecasting** A natural approach to context-aided forecasting is to build methods based on LLMs. Xue & Salim (2023) showed that forecasting could be framed as a question-answering problem. Subsequently, Gruver et al. (2024) and Requeima et al. (2024) showed that LLMs could generate accurate forecasts with sequence completion, and that textual side-information could be used to influence forecasts. However, their analysis is limited to illustrative examples rather than a comprehensive evaluation. Some works have explored the ability of LLMs to reason about time series (Chow et al., 2024; Kong et al., 2025; Aksu et al., 2024; Potosnak et al., 2024; Ye et al., 2024). Other approaches have incorporated time series into pretrained LLMs (Jin et al., 2024; Liu et al., 2024c; Zhang et al., 2024) by introducing special tokens used to represent patched time series patterns, or modifying their encoders to account for time series data (Jia et al., 2024). While these methods show promising results, their evaluations primarily rely on datasets where the contextual information is not guaranteed to improve forecasts over numerical data alone. As a result, it remains unclear whether their performance is due by accurate numerical forecasting or by effectively incorporating context; this shortcoming motivates our investigation into this question.

## 7. Discussion

In this work, we propose the Context is Key (CiK) benchmark: a collection of forecasting tasks that require processing historical data with essential natural language context. We evaluate a range of models on CiK, including our

proposed LLM prompting method, DIRECT PROMPT, which achieves the best aggregate performance. We analyze and discuss the failure modes of these models, and our findings underscore the critical role of contextual information in improving forecasts, while also revealing both the unexpected strengths and notable limitations of the investigated LLM-based forecasters.

**Limitations:** While our benchmark provides valuable insights into the integration of contextual information in time series forecasting, it is important to acknowledge its limitations. Our study limits itself to the natural language modality for context, and excludes multivariate time series scenarios. Although we deliberately designed the tasks to assess how well forecasting models can integrate contextual information, our benchmark does not evaluate whether models can leverage latent relationships that might elude human observation. While we have taken steps to mitigate memorization concerns, as discussed in Sec. 3.1, achieving absolute certainty in this regard is challenging without strictly held-out data. Finally, the performance of dataset-specific methods such as ChatTime (Wang et al., 2024a), UniTime (Liu et al., 2024c) and Time-LLM (Jin et al., 2024) may improve in the presence of a dataset-specific training set.

**Future work:** There are several promising avenues for future work. Extensions to CiK could include multivariate forecasting tasks, or tasks that incorporate other modalities such as images, databases or spatiotemporal data. Tasks that deliberately challenge context length limitations, probe specific weaknesses of language models or include domain-specific expert knowledge would also be valuable additions. More research is also needed to better understand what drives the catastrophic failures that LLMs exhibit. The analysis of catastrophic failures could benefit from searching for systematic failure patterns, or failures associated with specific linguistic patterns. Training datasets for context-aided forecasting would enable a better evaluation of dataset-specific methods. In fact, methods to improve the automatic generation of large, high-quality datasets for context-aided forecasting could complement CiK. Furthermore, this motivates research into developing more accurate and efficient multimodal forecasting models, which our benchmark is well-positioned to support. Other avenues of research include exploring different input/output structures for forecasting with LLMs, finetuning specialized LLMs for context-aided forecasting, allowing models to scale test-time computation, and compressing the context to reduce the required amount of computation. Lastly, as models become more robust, they could be integrated into agentic systems with conversational interfaces, allowing forecasts to be augmented with human expertise and automatically retrieved information. Such advancements would represent a significant step toward automating and democratizing access to powerful forecasting tools.

## Impact Statement

This paper's goal is to advance the development of forecasting methods which can leverage contextual information, which can increase the likelihood that such methods will be adopted by various organizations. Therefore, this paper shares the positive and negative impacts that such methods will have. Such methods could increase forecasting accuracy and efficiency, increasing the ability to foresee future courses and plan in consequence. A secondary effect is an increased democratization of access to high-quality forecasts. On the flip side, this democratization could increase reliance on automated methods for decision-making purposes.

## Acknowledgements

The authors are grateful to Andrei Dinin, Christian Hudon, Ethan Honey, Gabrielle Gauthier Melançon, Ghazwa Darwiche, Kiarash Mohammadi, Léo Boisvert, Loic Mandine, Megh Vipul Thakkar, Oluwanifemi Isaac Bamgbose, Orlando Marquez, Raymond Li, Thibault Le Sellier de Chezelles and Thomas Lai for participating in the human study on the relevance of context. The authors are grateful to Midan Kim, Torsten Scholak, Mohammad Reza Samsami, Oussama Boussif and Can Chen for their valuable feedback and suggestions. This research was supported by Mitacs Accelerate Grants and enabled by compute resources provided by ServiceNow Research and the Frontier supercomputer. The latter resources were awarded through the Frontier DD allocation and INCITE 2023 program for the project "Scalable Foundation Models for Transferable Generalist AI" and were supplied by the Oak Ridge Leadership Computing Facility at the Oak Ridge National Laboratory, with support from the Office of Science of the U.S. Department of Energy.

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

# Appendix

## Table of Contents

## A. Additional Details on the Benchmark

### A.1. Data Sources

We list here the domains and the respective sources of time series data we use in the various tasks in the CiK benchmark. We also show the number of tasks that use each source's data and list any memorization mitigation strategies used for each dataset.

- **Traffic** (11 tasks):

  - **Traffic occupancy rate**: We use traffic occupancy rate (%) data from the California Performance Measurement System (PeMS) (Chen et al., 2001), with frequency hourly. This dataset contains a total of 446 series.
    * As this is a live dataset (updated frequently), we use data from 2024 (i.e. data after the cutoff dates of LLMs used) and do not apply any memorization mitigation strategy.

- **Climatology** (12 tasks):

  - **Solar irradiance and cloud cover data** (9 tasks): We use solar irradiance and cloud cover data for the Americas in 2022 (Sengupta et al., 2018), with frequency either 10 minutes or hourly. We extract a subset of 45 series from this dataset for the benchmark.
    * To mitigate memorization, we shift the dates by one day ahead.
  - **Solar photovoltaic power production** (3 tasks): Time series reflecting solar power production in Alabama during 2006 (Godahewa et al., 2021), with a frequency 10 minutes. This dataset contains a total of 137 series, but our tasks only use a single aggregated series generated from them.
    * To mitigate memorization, we add gaussian noise to the data with a standard deviation of 3% of the standard deviation of the data in each respective sampled window.

- **Public Safety** (26 tasks):

  - **Fire Department Intervention Logs**: Logs of number of interventions carried out by the Montreal Fire Department due to the occurence of various kinds of incidents (such as trash fires, field fires, nautical accidents, bike accidents) (Ville de Montréal, 2020). The data was processed from a raw log and aggregated to monthly frequency. This dataset contains a total of 48 series.
    * Due to it being processed, we do not apply any special memorization mitigation strategy on top.

- **Mechanics** (3 tasks):

  - **Causal Chambers**: Experimental data collected from the wind tunnel physical system from Gamella et al. (2024), released in April 2024. We make use of the `load_in`, `pressure_downwind`, `pressure_ambient` and `speed_in` series (downsampling them to 1s frequency) to build out-of-distribution forecasting tasks where the target values can be inferred from the driver variate provided as covariate and the description of the physical system given in the context. We select a subset of 17 series from this dataset for the benchmark.
    * Since the data is released in 2024 and after the cutoff dates of the LLMs used, we do not apply any memorization mitigation technique to transform the data.

*Table 2.* Summary of Transformations Applied to Tasks Per Domain

| Domain | Number of tasks | Transformations | |
|--------|-----------------|------------|----------------|
| | | Date shift | Gaussian noise |
| Public Safety | 26 | None | None |
| Traffic | 11 | None | None |
| Mechanics | 3 | None | None |
| Economics | 3 | None | None |
| Synthetic | 3 | None | None |
| **Total** | **46** | **None** | **None** |
| Climatology | 12 | 9 | 3 |
| Energy | 7 | None | 7 |
| Retail | 6 | None | 6 |
| **Total** | **25** | **9** | **16** |

- **Economics** (3 tasks):

  - **FRED**: American unemployment data at the state and county levels, from the Federal Reserve Bank of St. Louis (U.S. Bureau of Labor Statistics, 2024), with frequency monthly. We extract a subset of 1769 series from this dataset for the benchmark.
    * As this is a live dataset (updated frequently), we use data from 2024 (i.e. data after the cutoff dates of LLMs used) and do not apply any memorization mitigation strategy.

- **Retail** (6 tasks):

  - **NN5 ATM cash withdrawals**: The NN5 dataset of ATM cash withdrawals in the UK from the Monash Time Series Forecasting Repository (Godahewa et al., 2021), with frequency daily. This dataset contains a total of 111 series.
    * To mitigate memorization, we add gaussian noise to the data with a standard deviation of 3% of the standard deviation of the data in each respective sampled window.

- **Energy** (7 tasks):

  - **Electricity consumption**: Electricity usage from 2012 to 2014 from the Monash Time Series Forecasting Repository (Godahewa et al., 2021), with frequency daily. This dataset contains a total of 321 series.
    * To mitigate memorization, we add gaussian noise to the data with a standard deviation of 3% of the standard deviation of the data in each respective sampled window.

- **Synthetic Data** (3 tasks): We employ a bivariate setup where the parent variable is drawn from a categorical distribution, and the child variable is generated using a continuous linear Structural Vector Autoregressive (SVAR) model with Gaussian noise, with a lag of 3 and a noise scale of 0.1.

  - Since this data is synthetic, we do not apply any mitigation technique on top of data to mitigate memorization. Since our models assume a timestamp, we use dates from 2025, and a frequency of daily when we input this data to our models.

Depending on the task and the context used in the task, appropriate history and prediction lengths are used in the task.

A summary of the number of tasks with either types of memorization strategy (shifting the dates by one day, or adding gaussian noise) is presented in Tab. 2.

### A.2. Task Creation Process

All tasks were manually designed, from scratch, by the authors of this work without resorting to external annotators, crowdsourcing, or LLMs. We used the following procedure to create the tasks in the benchmark.

**Task Diversity:** First, we identified high-quality sources of public time series data from various application domains (Sec. 3.1). Special care was taken to find data sources that are continuously updated to facilitate future benchmark updates. Second, we established the categorization for types of context (Sec. 3.2). Third, we posited various reasoning capabilities that one could potentially use to infer non-trivial information about numerical forecasts from contextual information (e.g., using common sense, making analogies to covariates, etc.; Appendix A.3). The task ideation process that followed aimed to ensure sufficient coverage of these three aspects.

**Task Ideation:** With this framework established, all authors contributed to the ideation of new tasks. In summary, the process consisted of:

1. Selecting a data source

2. Implementing a time series window selection strategy (e.g., short or long history)

3. Brainstorming about the nature of contextual information that could help achieve better forecasts (e.g., information about the past) and the capabilities that might potentially serve to apply it to the forecasting problem.

4. Writing code to verbalize the context (e.g., calculating statistics of the series beyond the observed numerical history, creating a template to render such statistics as text, etc.), and

5. Finally, if required, writing code to modify the time series data to reflect the context (e.g., introducing some spikes in future values).

**Peer Review:** Then, the tasks were peer-reviewed by a committee composed of all other authors (each with time series research experience). The creator of each task was not allowed to participate in the review. The review ensured that the contextual information was of high quality, that it undoubtedly enabled a better forecast, and that the context types used in the task were tagged correctly. If a task was deemed of not high enough quality, it was either returned for revisions or excluded from the benchmark.

**Code availability:** The code for all tasks is available here: `https://github.com/ServiceNow/context-is-key-forecasting`. An example task can be found here: `https://github.com/ServiceNow/context-is-key-forecasting/blob/main/cik_benchmark/tasks/montreal_fire/short_history.py`, where the time series window selection occurs from L94-112 and the context generation occurs from L114-158.

## A.3. Model Capabilities

As mentioned in Appendix A.2, we designed the tasks with consideration of the capabilities a model might potentially use to incorporate contextual information into its forecasts. All tasks in CiK are tagged with such capabilities. However, these tags are inherently subjective and not intended as formal attributions. Rather, they serve as broad categories to help readers identify examples of interest within the benchmark. These are as follows:

**Common-Sense** (24 Tasks): Using direct instructions available in the context. Instructions could express constraints to be satisfied, or the exact expected effect of an event, for example.

**Retrieval**: Retrieving facts from memory or context.

• **Retrieval from memory** (35 Tasks): Retrieving from memory facts that enable interpretation of the context, such as relevant physical constants or quantitative laws.
• **Retrieval from context** (25 Tasks): Retrieving relevant information from context and distinguishing it from irrelevant information.

**Reasoning**: Reasoning about information in context or memory.

• **Analogical Reasoning** (6 tasks): Making analogies between entities or events, for instance, applying knowledge from a past event that is similar to an upcoming one.
• **Mathematical Reasoning** (32 tasks): Performing calculations over the context, e.g. solving an equation.
• **Deductive Reasoning** (39 tasks): Inferring new facts not explicitly mentioned in the context, e.g. inferring from the context that certain values are logically impossible to occur.
• **Causal Reasoning** (22 tasks): Deriving or using causal information from the context to reason about actions (such as interventions).

**Example:** To illustrate the rationale, we provide the following example. To solve the task in Figure 4, one could *retrieve from memory* that Montreal experiences snowfall and cold weather during the winter months. It could then infer that trash fires are less likely to occur during this period through *deductive reasoning*. This chain of thought reveals a seasonal pattern that is not apparent in the short numerical history. Additionally, through *causal reasoning*, it is apparent that, despite a strong association between field fires and trash fires, the intervention described in ⑤ is unlikely to reduce the frequency of the latter. Failure to recognize this distinction would lead to inaccurate forecasts.

### A.4. Validating the Relevance of the Context

To evaluate the relevance of the contextual information for tasks in CiK, we query both human and LLM annotators on the relevance of the context. Our findings confirm that the context is relevant for all tasks (see Figure 8).

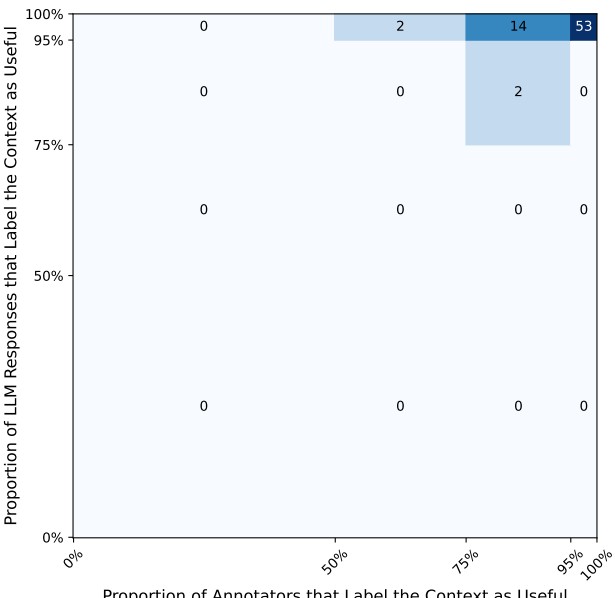

*Figure 8.* Ratings of the relevance of the context for both human annotations (x-axis) and LLM annotations (y-axis). There are 5 ratings per task for the LLM, and between 4 and 10 ratings per task for the human annotators. Each cell represents the number of tasks that correspond to a given pair of (human, LLM) ratings for the relevance of the context. For example, 53 (top right) of the 71 tasks have over 95% of annotators tagging the context as useful, as well as more than 95% of the LLM annotations tagging the context as useful. All tasks are considered relevant by either the LLM or the human annotators. Furthermore, the vast majority of tasks are considered relevant across more than 95% of ratings .

**Human Evaluation of the Relevance of Context** To ensure that the context used in the tasks is relevant to the tasks, we ask 11 human annotators to evaluate the relevance of the context across 5 seeds. After a brief presentation of two example tasks based on examples from (Hyndman & Athanasopoulos, 2018), we ask the annotators whether the context enables better forecasts. Annotators are instructed that tasks are designed for the purpose of context-aided forecasting, and that we are seeking to identify tasks for which the context is not useful.

The results of this study can be found in Figure 9: the vast majority of tasks are always annotated as useful, while annotators disagree on a small minority of tasks, such as `FullCausalContextImplicitEquationBivarLinSVAR` (visualized at `https://servicenow.github.io/context-is-key-forecasting/v0/FullCausalContextImplicitEquationBivarLinSVAR.html`) likely due to the highly statistical nature of such task(s). Overall, $94.7\%$ of annotations report that the context is useful.

**An LLM-based Critique of the Relevance of Context** To further validate the quality of the tasks, we build an LLM-based critique by prompting `GPT-4o` with the historical and future numerical data, as well as the context, and asking it to assess

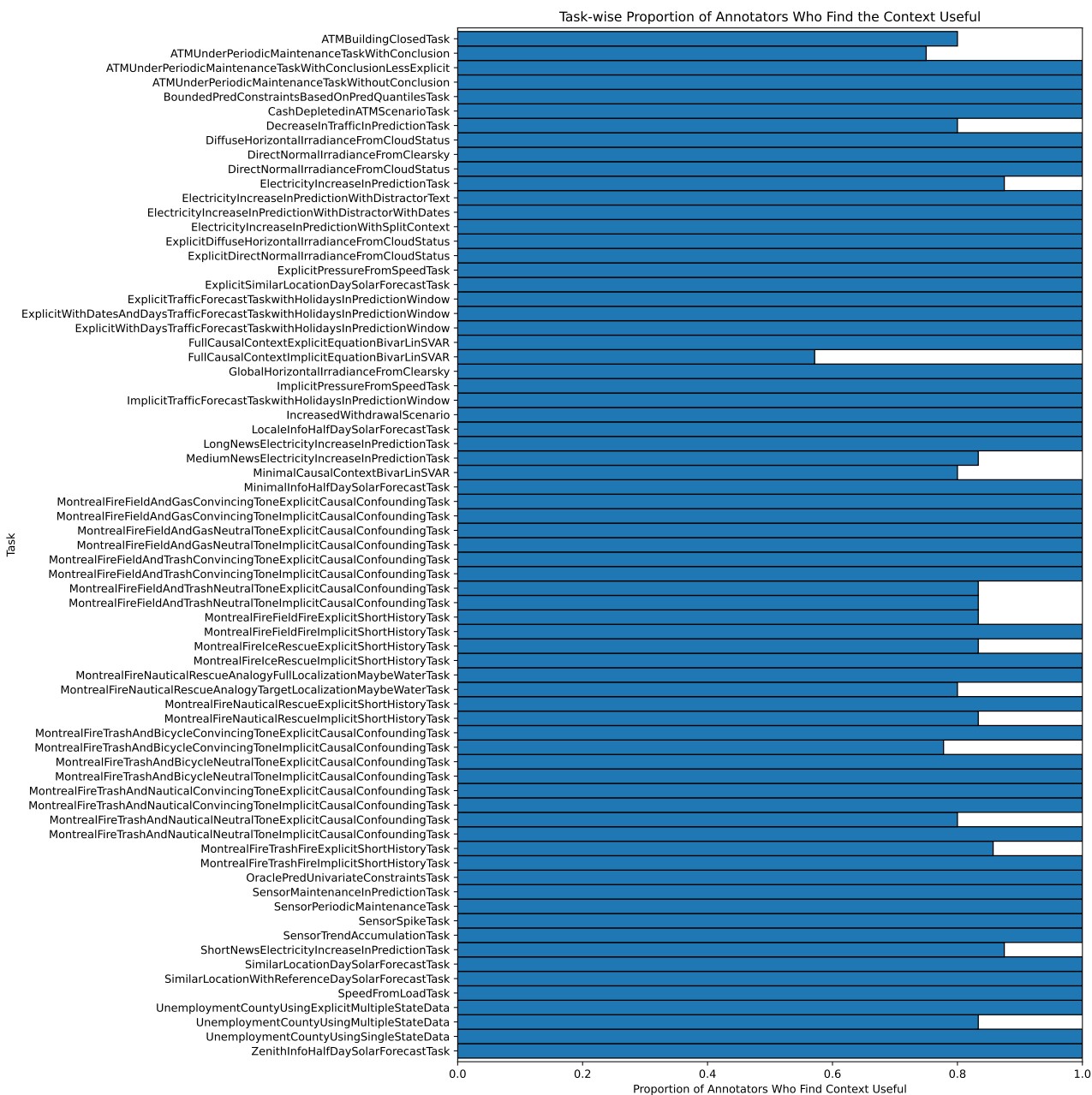

*Figure 9.* The task-wise proportion of annotators (n=11) who tag the context as useful. The overall rate of tasks tagged as useful across all annotators is 94.7%.

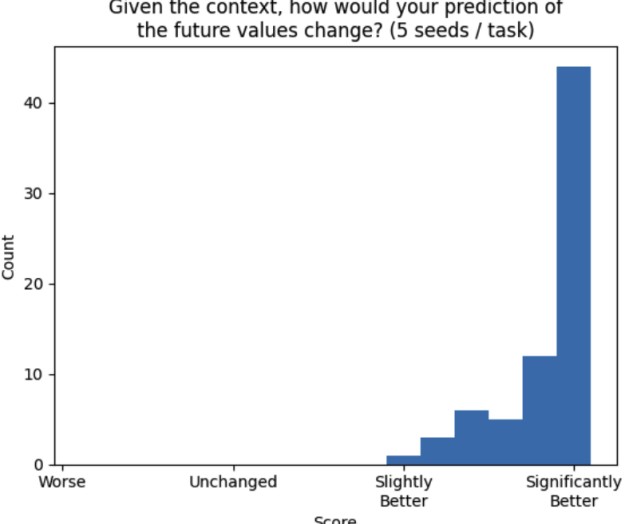

*Figure 10.* A histogram of results from the LLM-based critique of the relevance of context. Given the historical data, the future data and the associated context of tasks, GPT-4o is asked to assess whether its predictions would be "significantly better", "slightly better", "unchanged", or "worse" (see Appendix A.4 for the details). The context in all tasks is considered as enabling better forecasts, with the majority of tasks having context that enable "significantly better" forecasts.

whether its estimation of future values would be "significantly better", "slightly better", "unchanged", or "worse" when the context is provided compared to when it is not provided. Note that this experiment was ran after the benchmark was created, as an analysis tool to further validate the quality of the tasks.

We run this critique on 5 instances of each of the 71 tasks and report results in Figure 10. [2] All tasks are assessed as enabling better forecasts when given context, with the majority of tasks assessed as having contexts that enable "significantly better" forecasts. The prompt used in the critique is in Figure 11.

### A.5. Weighting scheme for tasks

To take full advantage of the available data, we create multiple tasks using each data source, by varying the specific contextual information we provide to the models. Since we do not want our aggregate results to be dominated by the few datasets for which there are a larger number of tasks, we weight the contribution of each task to the various aggregated results.

To define the weight of each task, we first group the tasks in clusters. These clusters are primarily defined based on the original data source used to create the tasks. However, when tasks are fundamentally different, due to not testing the same capabilities, we put them in different clusters despite them using the same data source. For example, for tasks created using the Solar irradiance and cloud cover data, all of which ask models to forecast the irradiance, the tasks form three distinct clusters: one for tasks asking models to do forecast with very short history (less than a day), one for tasks giving the cloud cover as covariate, and the final one for tasks where the models are given a tight upper bound on the possible irradiance. Once we define these clusters, we simply give equal weight to each cluster, and equal weight to each task inside each cluster.

### A.6. Standard errors and average ranks

To get the standard errors shown in Tab. 1, we first compute the standard error for tasks using the method described in Appendix E.5. We then aggregate them according to each task weight, by assuming that errors for each are independent and thus using the formula for the variance of a weighted sum of independent variables.

To take into consideration the uncertainty we have for the scores, we compute average ranks through a simple simulation. In

---

[2]For Figure 8, we consider both "significantly better" and "slightly better" as meaning the context is useful.

```
            "

You are a critic whose role is to evaluate the quality of tasks in the "context is key" time series forecasting
benchmark.

"Context is Key" (CiK) is a time series forecasting benchmark that pairs numerical data with diverse types of
carefully crafted textual context, requiring models to integrate both modalities to arrive at accurate
predictions.

Here is a task to evaluate.

<history>
((history))
</history>

<context>
    <background>
        ((background))
    </background>
    <scenario>
        ((scenario))
    </scenario>
    <constraints>
        ((constraints))
    </constraints>
</context>
<future>
((future))
</future>

Assume the following two scenarios:
1) You are given only the numerical data in <history> and have no additional information about the nature of the
time series. You must ignore the <context> section completely.

2) You are given the <context> section in addition to the numerical data in <history>.

Now, assume you had to estimate the probability distribution of the <future> values given the information
available in each scenario. How would the quality of your estimation change in scenario 2 compared to
scenario 1?

First show your reasoning in <reason></reason> tags, then answer in <answer></answer> tags with either
"significantly better", "slightly better", "unchanged", "worse" (no other responses are allowed).
```

*Figure 11.* The prompt used to query the LLM critique.

this simulation, we replace the RCRPS for each task and model pair by an independent Gaussian variable of mean equals to the one we measured, and of standard deviation equals to the standard error. We then draw from this distribution and compute the weighted average ranks for each model. The results shown in Tab. 1 are the mean and standard deviation measured from 10,000 repetitions of this simulation.

### A.7. Task lengths

Figure 12 provides an overview of the distribution of the lengths of the natural language context, numerical history and target (prediction horizon) for a set of five instances for each task in the CiK benchmark.

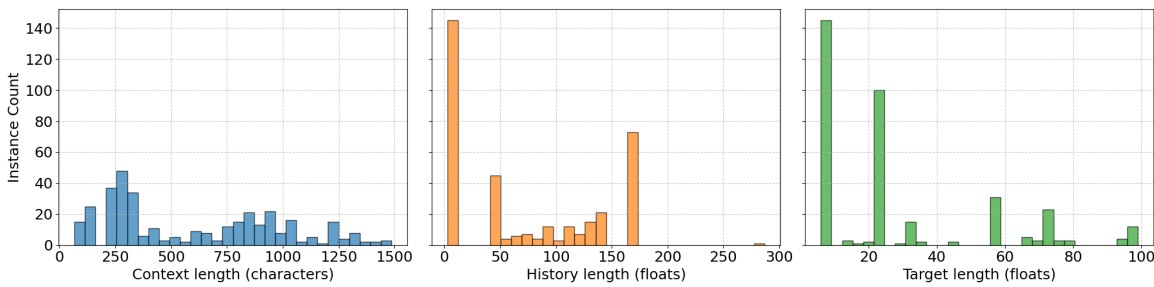

*Figure 12.* Histograms depicting the distribution of lengths for the context, numerical history and target length of a set of five instances for each task in CiK. We measure the length of the natural language context in characters, and the numerical sequences in floats.

## B. Examples of tasks from the benchmark

In this section, we feature multiple examples from the benchmark to exemplify exactly what a task is, what context types represent (Sec. 3.2), and how we tag these tasks with descriptive capabilities (Appendix A.3). To visualize all tasks in the benchmark, we refer the reader to `https://servicenow.github.io/context-is-key-forecasting/v0/`.

### B.1. Task: Constrained Predictions

> **Domain:** Traffic
> **Context types:** Future information
>
> **Context:** "Suppose that in the forecast, the values are bounded above by 11.88, the values are bounded below by 7.06."

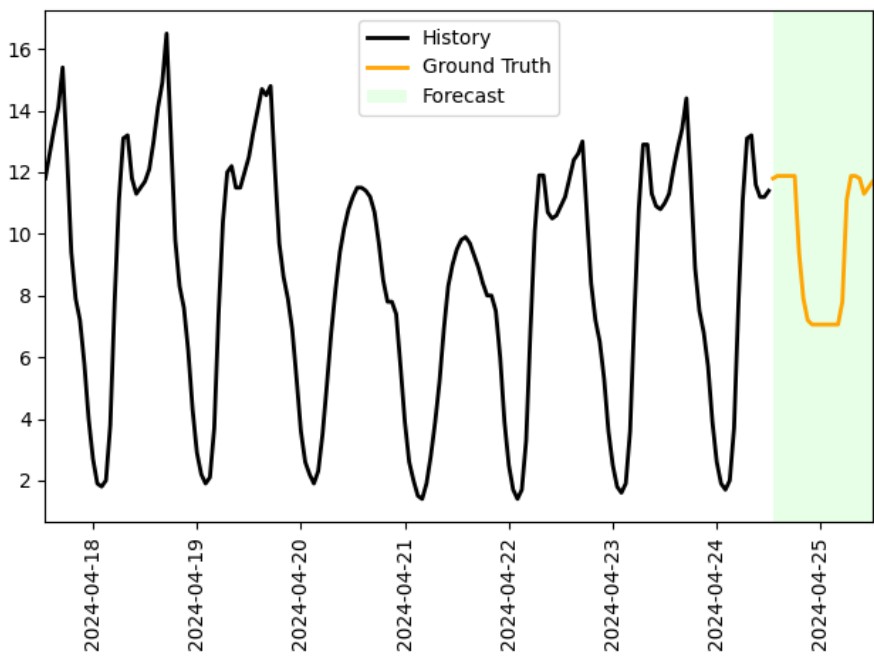

This task, which we refer to as "Bounded Prediction Constraint Based On Prediction Quantiles", is a forecasting task where we modify the forecast horizon (in green in the plot) by bounding one or both of its extremes according to its unmodified ground truth's quantile values. We verbalize these bounds in the context, and the model is expected to interpret and respect them.

Since we draw this series from the PeMS dataset (Chen et al., 2001), we tag its domain as "Traffic". The context directly

refers to the future, hence the context source is tagged as "Future information".

Since the context contains constraints, the Region of Interest CRPS metric that we introduce (Sec. 4) heavily penalizes forecasts that exceed these constraints: models that do not incorporate the information about bounds in the context, such as quantitative forecasting models, would not be able to predict the ground truth (orange line) because its lower bound is much higher than that of the history. In this case, the region of interest for the metric is the entire forecast horizon because the context applies everywhere. Although statistical forecasters may pick up on the seasonality present in the history (black line), they would obtain worse scores than models capable of processing the context and adjusting the lower bound of their predictions.

### B.2. Task: Electrical Consumption Increase

**Domain:** Energy
**Context types:** Future information, Covariate information

**Context:** "This is the electricity consumption recorded in Kilowatt (kW) in city A. A heatwave struck the city, which began on 2012-10-09 18:00:00 and lasted for approximately 3 hours, saw temperatures soar to unprecedented levels. According to the city's electricity provider, power consumption during the peak of the heatwave reached approximately 5 times the typical usage for this time of year."

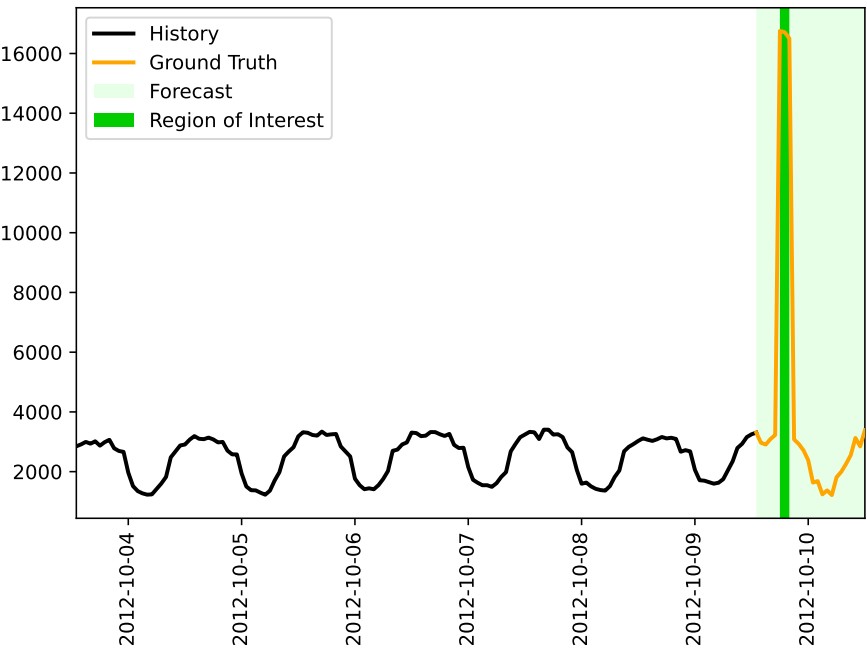

The "Short News Electricity Increase" task introduces a large shock in the forecast horizon that is only referred to in the context. Hence, the model must interpret the context appropriately to forecast the spike.

Since this series represents electricity consumption (Sec. 3.1), we tag it a coming from the "Energy" domain. The context types for this task are twofold: the first context source is "Future information", which represents knowledge of the five-fold increase in typical usage during the shock. The second source of context, "Covariate information", represents the occurrence of a heatwave, which coincides with the timing and duration of the shock. The model must therefore interpret both the information on the magnitude of the shock from the future information, as well as the timing and duration of the sock from the covariate information. Together, these pieces of information enable an accurate forecast despite the lack of information about the shock in the task's numerical history.

In this task, we also see a "Region of Interest" (RoI), characterized by a darker region of the forecast horizon. This RoI represents the region of the forecast horizon for which the context is relevant, i.e. the period during which the increased power consumption occurred. As detailed in Sec. 4, this region of interest is taking into account in the RCRPS metric.

**B.3. Task: ATM Maintenance**

---

**Domain:** Retail

**Context types:** Intemporal information, Covariate information

**Context:** "This is the number of cash withdrawals from an automated teller machine (ATM) in an arbitrary location in England. The ATM was under maintenance for 7 days, periodically every 14 days, starting from 1996-11-30 00:00:00. Assume that the ATM will not be in maintenance in the future."

---

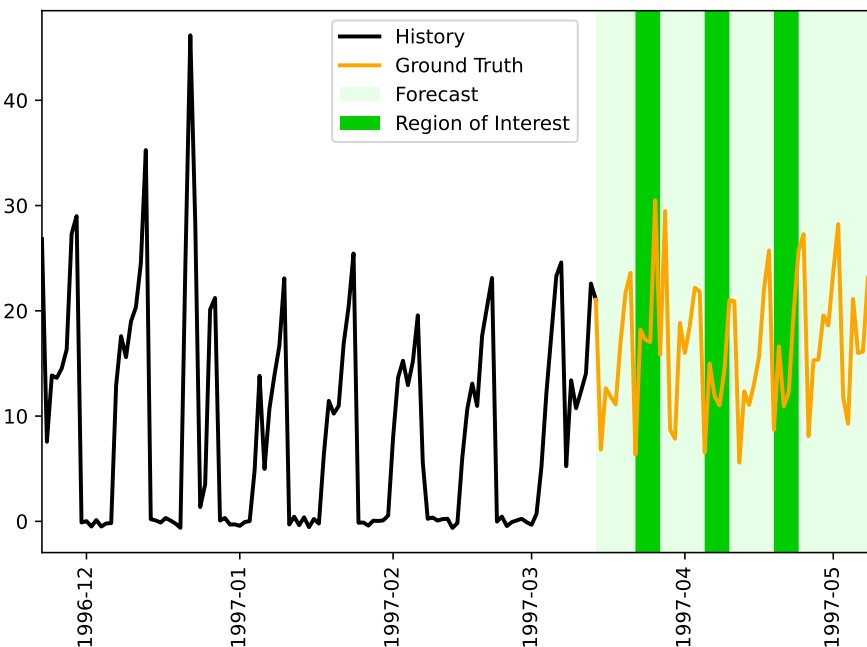

The "Automated Teller Machine (ATM) Under Period Maintenance" task represents the history of withdrawals from an ATM that undergoes regular maintenance. This maintenance introduces a periodic, easily forecastable signal into the history. However, the context explicitly states that the forecast should assume the ATM will not be in maintenance during the forecast. Therefore, forecasting models are expected to ignore this signal.

Since this series represents ATM withdrawals, we tag it as "Retail". The context includes information such as the location of the ATM, and therefore provides "Intemporal information". As the maintenance frequency and duration is also described, the context types include "Covariate information".

The RoI represents when the maintenance periods would have occurred in the forecast horizon, which is likely where forecasting models that do not leverage the context will forecast 0. While a quantitative forecasting model would find such a signal irresistible, context-aware models should avoid repeating the pattern in the forecast.

We also note that the series is not quite 0 during the maintenance periods. This is a consequence of using one of our memorization mitigation schemes (Appendix A.1, paragraph "Memorization mitigation").

**B.4. Task: Montreal Fire High Season**

**Domain:** Public Safety

**Context types:** Intemporal information, Historical information

**Context:** "The Montreal Fire Department is in charge of responding to various kind of public safety incidents. This is the number of field fire incidents responded to by Montreal firefighters in the borough of Rivière-des-Prairies-Pointe-aux-Trembles. In other years, the yearly average number of incidents was 106 with the busiest month being June."

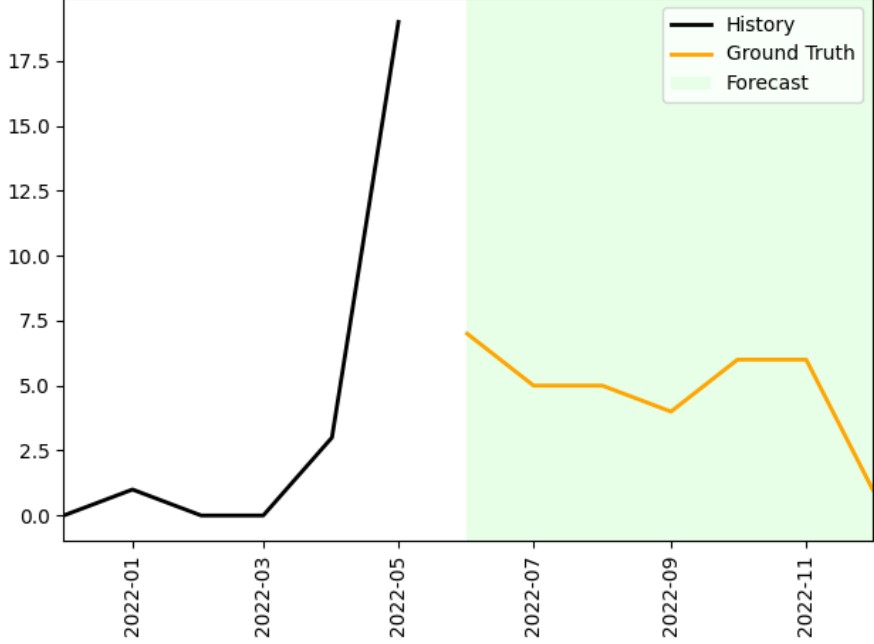

The "Montreal Field Fire With Explicit Short History" task requires predicting the number of field fire incidents during the summer, so we tag it as being part of the "Public Safety" domain.

The context contains information from two different sources: it contains "Intemporal information", such as the location and nature of the incidents. However, it also contains "Historical information", which verbalizes statistics about past values of the series, beyond the numerical data. That is, the yearly average number of incidents, along with the knowledge that June is the month with the most incidents.

## B.5. Task: Solar Prediction

**Domain:** Climatology
**Context types:** Intemporal information

**Context:** "This series estimates the power production for a given day of a new solar power plant located in the state of Georgia, which has a climate similar to Alabama's."

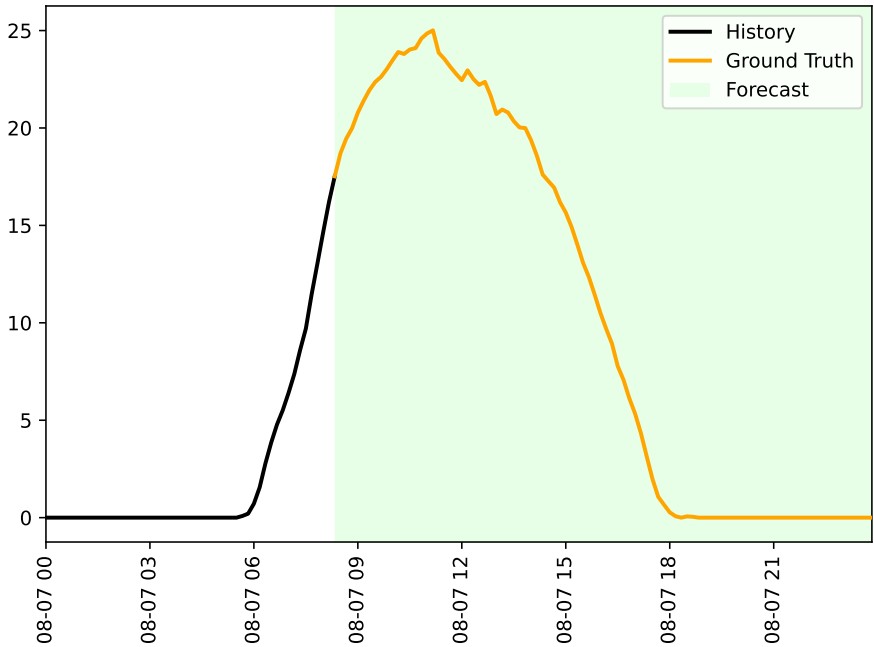

The "Explicit Similar Location and Day Solar Forecast" task requires forecasting the power production of a solar power plant based on a very short history and information about the similarity between its climate and that of an adjacent location. We therefore tag the domain of this series as "Climatology".

Without the "Intemporal information" that the context provides, accurate forecasts of the parabola-like shape of the ground truth are unlikely: the history contains very few defining characteristics, which makes it interchangeable with that of many potential processes and therefore many possible forecasts.

**B.6. Task: Speed From Load**

---

**Domain:** Mechanics

**Context types:** Causal information, Intemporal information, Covariate information

**Context:** "The wind tunnel is a chamber with one controllable fan that pushes air through it. We can control the load of the fan (corresponding to the duty cycle of the pulse-width-modulation signal) and measure its speed (in revolutions per minute). The fan is designed so its steady-state speed scales broadly linearly with the load. Unless completely powered off, the fan never operates below a certain speed, corresponding to a minimum effective load between 0.1 and 0.2. The task is to forecast the speed of the fan. The load is between 0 and 1. At full load (=1), the fan turns at a maximum speed of 3000 rpm. The load is set to: 0.0 until 05:47:09, 0.1 from 05:47:09 until 05:47:29, 0.0 from 05:47:29 until 05:48:01, 0.2 from 05:48:01 until 05:48:27, 0.1 from 05:48:27 until 05:48:49, 0.0 from 05:48:49 until 05:49:00."

---

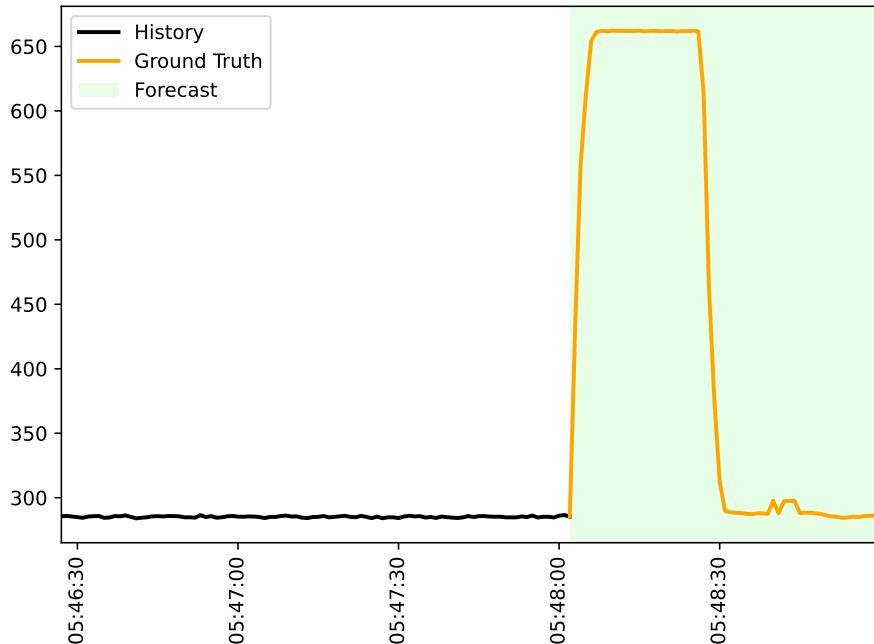

The "Speed From Load" task combines many different context types and capabilities to produce a forecast of the revolutions per minute (RPM) of a fan in a wind tunnel based on its load. This task, based on the Causal Chambers dataset (Gamella et al., 2024), is therefore tagged as part of the "Mechanics" domain.

As the plot shows, producing an accurate forecast of the ground truth (orange line) from the numerical history alone (black line) is essentially impossible. However, the context of the task is quite rich: it provides "Intemporal information" on the nature of the task, such as the limits of the load and of the fan, "Covariate information" that describes the load during the history and future, as well as "Causal information" on the control that the load exerts on the fan, as well as the proportionality of their relationship. With this information, it is possible to forecast the series nearly perfectly, excepting some noise.

## C. Additional Results

The full unaggregated results from our benchmark can be found at the following location: `https://github.com/ServiceNow/context-is-key-forecasting/blob/main/results/results_complete.csv`.

## C.1. Extended results on all models

*Table 3.* Results of all models in the CiK benchmark aggregated over all tasks. The first column shows the RCRPS averaged over all tasks. The second column shows the rank of each method w.r.t. other baselines, averaged over all tasks. All averages are weighted according to the scheme described in Sec. 5.1 and accompanied by standard errors. Lower is better and the best means are in bold.

| Model | Average RCRPS | Average Rank |
|---|---|---|
| **With Context** | | |
| DIRECT PROMPT (ours) | | |
| Llama-3.1-405B-Inst | **0.159 ± 0.008** | **7.962 ± 0.591** |
| Llama-3-70B-Inst | 0.286 ± 0.004 | 14.806 ± 0.200 |
| Llama-3-8B-Inst | 0.461 ± 0.008 | 28.020 ± 0.579 |
| Mixtral-8x7B-Inst | 0.523 ± 0.023 | 33.066 ± 0.419 |
| Qwen-2.5-7B-Inst | 0.290 ± 0.003 | 22.840 ± 0.731 |
| Qwen-2.5-1.5B-Inst | 0.575 ± 0.014 | 32.643 ± 0.860 |
| Qwen-2.5-0.5B-Inst | 0.463 ± 0.012 | 27.216 ± 0.530 |
| GPT-4o | 0.274 ± 0.010 | 8.632 ± 0.441 |
| GPT-4o-mini | 0.354 ± 0.022 | 17.574 ± 0.498 |
| LLMP | | |
| Llama-3-70B-Inst | 0.539 ± 0.013 | 18.039 ± 0.569 |
| Llama-3-70B | 0.236 ± 0.006 | 12.383 ± 0.727 |
| Llama-3-8B-Inst | 0.483 ± 0.010 | 18.597 ± 0.471 |
| Llama-3-8B | 0.311 ± 0.023 | 18.640 ± 1.042 |
| Mixtral-8x7B-Inst | 0.264 ± 0.004 | 16.078 ± 0.668 |
| Mixtral-8x7B | 0.262 ± 0.008 | 16.296 ± 0.516 |
| Qwen-2.5-7B-Inst | 1.974 ± 0.027 | 45.227 ± 0.747 |
| Qwen-2.5-7B | 0.910 ± 0.037 | 38.133 ± 1.049 |
| Qwen-2.5-1.5B-Inst | 2.158 ± 0.027 | 50.647 ± 0.865 |
| Qwen-2.5-1.5B | 1.731 ± 0.036 | 45.116 ± 0.525 |
| Qwen-2.5-0.5B-Inst | 1.937 ± 0.024 | 50.480 ± 0.611 |
| Qwen-2.5-0.5B | 1.995 ± 0.024 | 48.511 ± 0.823 |
| Multimodal Models | | |
| UniTime | 0.370 ± 0.001 | 35.459 ± 0.155 |
| Time-LLM (ETTh1) | 0.476 ± 0.001 | 44.086 ± 0.149 |
| ChatTime-Base | 0.735 ± 0.002 | 39.033 ± 0.312 |
| ChatTime-Chat | 0.747 ± 0.005 | 34.182 ± 0.389 |
| **Without Context** | | |
| DIRECT PROMPT (ours) | | |
| Llama-3.1-405B-Inst | 0.473 ± 0.005 | 34.332 ± 0.292 |
| Llama-3-70B-Inst | 0.470 ± 0.008 | 35.151 ± 0.361 |
| Llama-3-8B-Inst | 0.602 ± 0.006 | 41.872 ± 0.406 |
| Mixtral-8x7B-Inst | 0.712 ± 0.021 | 46.805 ± 0.384 |
| Qwen-2.5-7B-Inst | 0.421 ± 0.022 | 37.162 ± 0.539 |
| Qwen-2.5-1.5B-Inst | 0.450 ± 0.007 | 36.314 ± 0.621 |
| Qwen-2.5-0.5B-Inst | 0.305 ± 0.007 | 26.817 ± 0.351 |
| GPT-4o | 0.441 ± 0.008 | 31.511 ± 0.380 |
| GPT-4o-mini | 0.423 ± 0.006 | 35.716 ± 0.278 |
| LLMP | | |
| Llama-3-70B-Inst | 0.378 ± 0.004 | 26.036 ± 0.477 |
| Llama-3-70B | 0.311 ± 0.006 | 21.819 ± 0.472 |
| Llama-3-8B-Inst | 0.503 ± 0.009 | 31.156 ± 0.435 |
| Llama-3-8B | 0.345 ± 0.003 | 25.063 ± 0.379 |
| Mixtral-8x7B-Inst | 0.383 ± 0.015 | 24.595 ± 0.471 |
| Mixtral-8x7B | 0.306 ± 0.007 | 22.565 ± 0.514 |
| Qwen-2.5-7B-Inst | 1.020 ± 0.026 | 39.936 ± 1.236 |
| Qwen-2.5-7B | 0.732 ± 0.030 | 38.092 ± 1.194 |
| Qwen-2.5-1.5B-Inst | 1.515 ± 0.033 | 47.068 ± 1.080 |
| Qwen-2.5-1.5B | 1.070 ± 0.028 | 41.194 ± 1.057 |
| Qwen-2.5-0.5B-Inst | 1.318 ± 0.037 | 44.391 ± 0.759 |
| Qwen-2.5-0.5B | 1.821 ± 0.027 | 47.759 ± 0.706 |
| Multimodal Models | | |
| UniTime | 0.405 ± 0.002 | 37.247 ± 0.178 |
| Time-LLM (ETTh1) | 0.458 ± 0.002 | 43.019 ± 0.162 |
| ChatTime-Base | 0.725 ± 0.002 | 38.771 ± 0.351 |
| ChatTime-Chat | 0.781 ± 0.015 | 35.260 ± 0.446 |
| TS Foundation Models | | |
| Lag-Llama | 0.327 ± 0.004 | 30.441 ± 0.810 |
| Chronos-Tiny | 0.328 ± 0.001 | 27.487 ± 0.442 |
| Chronos-Mini | 0.341 ± 0.001 | 28.895 ± 0.421 |
| Chronos-Small | 0.328 ± 0.002 | 26.523 ± 0.368 |
| Chronos-Base | 0.672 ± 0.003 | 30.601 ± 0.379 |
| Chronos-Large | 0.326 ± 0.002 | 25.602 ± 0.399 |
| TimeGEN | 0.353 ± 0.000 | 35.928 ± 0.167 |
| Moirai-Small | 0.565 ± 0.031 | 36.037 ± 0.426 |
| Moirai-Base | 0.624 ± 0.013 | 35.267 ± 0.408 |
| Moirai-Large | 0.520 ± 0.006 | 28.641 ± 0.864 |
| Statistical Models | | |
| ARIMA | 0.475 ± 0.006 | 27.041 ± 0.482 |
| ETS | 0.530 ± 0.009 | 33.781 ± 0.632 |
| Exp-Smoothing | 0.605 ± 0.013 | 36.424 ± 0.350 |

Tab. 3 provides the extended results with all models evaluated on the CiK benchmark, aggregated over all tasks.

## C.2. Full results partitioned by types of context

*Table 4.* Results of all models on the CiK benchmark. Starting from the left, the first column shows the RCRPS averaged over all tasks. The second column shows the rank of each method w.r.t. other baselines, averaged over all tasks. The remaining columns show the average RCRPS stratified by types of context. All averages are weighted according to the scheme described in Sec. 5.1 and accompanied by standard errors. Lower is better and the best averages are in bold.

| Model | Average RCRPS | Average Rank | Intemporal Information | Historical Information | Future Information | Covariate Information | Causal Information |
|---|---|---|---|---|---|---|---|
| **With Context** | | | | | | | |
| DIRECT PROMPT (ours) | | | | | | | |
| Llama-3.1-405B-Inst | **0.159 ± 0.008** | **7.967 ± 0.587** | **0.174 ± 0.010** | 0.146 ± 0.001 | **0.075 ± 0.005** | **0.164 ± 0.010** | 0.398 ± 0.045 |
| Llama-3-70B-Inst | 0.286 ± 0.004 | 14.806 ± 0.201 | 0.336 ± 0.006 | 0.180 ± 0.003 | 0.194 ± 0.006 | 0.228 ± 0.004 | 0.629 ± 0.019 |
| Llama-3-8B-Inst | 0.461 ± 0.008 | 28.013 ± 0.588 | 0.572 ± 0.011 | 0.313 ± 0.008 | 0.253 ± 0.017 | 0.262 ± 0.003 | 0.531 ± 0.005 |
| Mixtral-8x7B-Inst | 0.523 ± 0.023 | 33.068 ± 0.415 | 0.723 ± 0.037 | 0.236 ± 0.002 | 0.241 ± 0.001 | 0.359 ± 0.028 | 0.875 ± 0.128 |
| Qwen-2.5-7B-Inst | 0.290 ± 0.003 | 22.842 ± 0.733 | 0.290 ± 0.004 | 0.176 ± 0.003 | 0.287 ± 0.007 | 0.240 ± 0.002 | 0.525 ± 0.003 |
| Qwen-2.5-1.5B-Inst | 0.575 ± 0.014 | 32.642 ± 0.864 | 0.684 ± 0.023 | 0.284 ± 0.006 | 0.370 ± 0.010 | 0.450 ± 0.004 | 1.270 ± 0.009 |
| Qwen-2.5-0.5B-Inst | 0.463 ± 0.012 | 27.218 ± 0.530 | 0.609 ± 0.019 | 0.165 ± 0.004 | 0.218 ± 0.012 | 0.476 ± 0.015 | 0.429 ± 0.006 |
| GPT-4o | 0.274 ± 0.010 | 8.631 ± 0.443 | 0.218 ± 0.007 | **0.118 ± 0.001** | 0.121 ± 0.001 | 0.250 ± 0.011 | 0.858 ± 0.053 |
| GPT-4o-mini | 0.354 ± 0.022 | 17.565 ± 0.506 | 0.475 ± 0.035 | 0.139 ± 0.002 | 0.143 ± 0.002 | 0.341 ± 0.028 | 0.644 ± 0.128 |
| LLMP | | | | | | | |
| Llama-3-70B-Inst | 0.539 ± 0.013 | 18.034 ± 0.571 | 0.438 ± 0.017 | 0.516 ± 0.028 | 0.847 ± 0.024 | 0.546 ± 0.016 | **0.392 ± 0.028** |
| Llama-3-70B | 0.236 ± 0.006 | 12.382 ± 0.726 | 0.212 ± 0.005 | 0.121 ± 0.008 | 0.299 ± 0.017 | 0.193 ± 0.004 | 0.360 ± 0.011 |
| Llama-3-8B-Inst | 0.483 ± 0.010 | 18.597 ± 0.482 | 0.476 ± 0.013 | 0.161 ± 0.006 | 0.326 ± 0.003 | 0.304 ± 0.008 | 0.878 ± 0.035 |
| Llama-3-8B | 0.311 ± 0.023 | 18.647 ± 1.030 | 0.332 ± 0.035 | 0.123 ± 0.004 | 0.271 ± 0.010 | 0.288 ± 0.029 | 0.739 ± 0.134 |
| Mixtral-8x7B-Inst | 0.264 ± 0.004 | 16.087 ± 0.667 | 0.242 ± 0.007 | 0.173 ± 0.004 | 0.324 ± 0.005 | 0.219 ± 0.005 | 0.437 ± 0.007 |
| Mixtral-8x7B | 0.262 ± 0.008 | 16.282 ± 0.523 | 0.250 ± 0.008 | 0.119 ± 0.003 | 0.310 ± 0.019 | 0.229 ± 0.006 | 0.457 ± 0.011 |
| Qwen-2.5-7B-Inst | 1.974 ± 0.027 | 45.235 ± 0.742 | 2.509 ± 0.044 | 2.857 ± 0.056 | 1.653 ± 0.008 | 1.702 ± 0.035 | 1.333 ± 0.144 |
| Qwen-2.5-7B | 0.910 ± 0.037 | 38.144 ± 1.039 | 1.149 ± 0.047 | 1.002 ± 0.053 | 0.601 ± 0.071 | 0.639 ± 0.047 | 0.928 ± 0.129 |
| Qwen-2.5-1.5B-Inst | 2.158 ± 0.027 | 50.652 ± 0.866 | 2.614 ± 0.041 | 1.672 ± 0.055 | 1.413 ± 0.029 | 2.057 ± 0.033 | 2.448 ± 0.128 |
| Qwen-2.5-1.5B | 1.731 ± 0.036 | 45.108 ± 0.519 | 2.337 ± 0.049 | 2.982 ± 0.052 | 0.942 ± 0.065 | 1.435 ± 0.046 | 1.304 ± 0.129 |
| Qwen-2.5-0.5B-Inst | 1.937 ± 0.024 | 50.493 ± 0.602 | 2.444 ± 0.038 | 1.960 ± 0.063 | 1.443 ± 0.010 | 1.805 ± 0.030 | 1.199 ± 0.129 |
| Qwen-2.5-0.5B | 1.995 ± 0.024 | 48.499 ± 0.834 | 2.546 ± 0.039 | 2.083 ± 0.052 | 1.579 ± 0.015 | 1.821 ± 0.030 | 1.225 ± 0.128 |
| Multimodal Models | | | | | | | |
| UniTime | 0.370 ± 0.001 | 35.456 ± 0.152 | 0.457 ± 0.002 | 0.155 ± 0.000 | 0.194 ± 0.003 | 0.395 ± 0.001 | 0.423 ± 0.001 |
| Time-LLM (ETTh1) | 0.476 ± 0.001 | 44.087 ± 0.148 | 0.518 ± 0.002 | 0.183 ± 0.000 | 0.403 ± 0.002 | 0.441 ± 0.001 | 0.482 ± 0.001 |
| ChatTime-Base | 0.735 ± 0.002 | 39.037 ± 0.311 | 0.663 ± 0.002 | 0.181 ± 0.001 | 0.374 ± 0.003 | 0.794 ± 0.002 | 2.727 ± 0.003 |
| ChatTime-Chat | 0.747 ± 0.005 | 34.186 ± 0.391 | 0.693 ± 0.007 | 0.405 ± 0.038 | 0.347 ± 0.007 | 0.832 ± 0.006 | 2.971 ± 0.018 |
| **Without Context** | | | | | | | |
| DIRECT PROMPT (ours) | | | | | | | |
| Llama-3.1-405B-Inst | 0.473 ± 0.005 | 34.334 ± 0.296 | 0.527 ± 0.007 | 0.713 ± 0.014 | 0.392 ± 0.003 | 0.320 ± 0.002 | 0.587 ± 0.005 |
| Llama-3-70B-Inst | 0.470 ± 0.008 | 35.150 ± 0.356 | 0.532 ± 0.013 | 0.676 ± 0.018 | 0.389 ± 0.003 | 0.317 ± 0.002 | 0.615 ± 0.005 |
| Llama-3-8B-Inst | 0.602 ± 0.006 | 41.874 ± 0.404 | 0.748 ± 0.009 | 0.679 ± 0.015 | 0.345 ± 0.008 | 0.335 ± 0.003 | 0.604 ± 0.004 |
| Mixtral-8x7B-Inst | 0.712 ± 0.021 | 46.807 ± 0.389 | 0.906 ± 0.035 | 0.758 ± 0.015 | 0.400 ± 0.001 | 0.485 ± 0.028 | 0.893 ± 0.128 |
| Qwen-2.5-7B-Inst | 0.421 ± 0.022 | 37.158 ± 0.547 | 0.479 ± 0.035 | 0.515 ± 0.017 | 0.322 ± 0.008 | 0.357 ± 0.028 | 0.830 ± 0.128 |
| Qwen-2.5-1.5B-Inst | 0.450 ± 0.007 | 36.312 ± 0.615 | 0.494 ± 0.011 | 0.324 ± 0.007 | 0.368 ± 0.008 | 0.315 ± 0.006 | 0.498 ± 0.009 |
| Qwen-2.5-0.5B-Inst | 0.305 ± 0.007 | 26.817 ± 0.353 | 0.341 ± 0.007 | 0.185 ± 0.004 | 0.236 ± 0.016 | 0.255 ± 0.005 | 0.396 ± 0.002 |
| GPT-4o | 0.441 ± 0.008 | 31.505 ± 0.387 | 0.492 ± 0.013 | 0.280 ± 0.007 | 0.376 ± 0.002 | 0.276 ± 0.001 | 0.504 ± 0.002 |
| GPT-4o-mini | 0.423 ± 0.006 | 35.711 ± 0.273 | 0.480 ± 0.009 | 0.391 ± 0.007 | 0.335 ± 0.004 | 0.280 ± 0.001 | 0.531 ± 0.003 |
| LLMP | | | | | | | |
| Llama-3-70B-Inst | 0.378 ± 0.004 | 26.030 ± 0.469 | 0.405 ± 0.006 | 0.186 ± 0.004 | 0.353 ± 0.004 | 0.253 ± 0.002 | 0.481 ± 0.004 |
| Llama-3-70B | 0.311 ± 0.006 | 21.812 ± 0.470 | 0.311 ± 0.004 | 0.142 ± 0.004 | 0.321 ± 0.018 | 0.245 ± 0.002 | 0.479 ± 0.006 |
| Llama-3-8B-Inst | 0.503 ± 0.009 | 31.147 ± 0.437 | 0.598 ± 0.014 | 0.262 ± 0.009 | 0.365 ± 0.004 | 0.266 ± 0.002 | 0.510 ± 0.001 |
| Llama-3-8B | 0.345 ± 0.003 | 25.063 ± 0.375 | 0.387 ± 0.004 | 0.162 ± 0.006 | 0.271 ± 0.007 | 0.250 ± 0.001 | 0.491 ± 0.002 |
| Mixtral-8x7B-Inst | 0.383 ± 0.015 | 24.587 ± 0.471 | 0.420 ± 0.024 | 0.162 ± 0.008 | 0.340 ± 0.004 | 0.349 ± 0.019 | 0.470 ± 0.005 |
| Mixtral-8x7B | 0.306 ± 0.007 | 22.567 ± 0.516 | 0.295 ± 0.004 | 0.150 ± 0.004 | 0.336 ± 0.021 | 0.242 ± 0.001 | 0.489 ± 0.003 |
| Qwen-2.5-7B-Inst | 1.020 ± 0.026 | 39.951 ± 1.232 | 1.435 ± 0.047 | 0.889 ± 0.032 | 0.376 ± 0.018 | 0.812 ± 0.032 | 0.810 ± 0.128 |
| Qwen-2.5-7B | 0.732 ± 0.030 | 38.091 ± 1.180 | 0.923 ± 0.045 | 0.403 ± 0.034 | 0.441 ± 0.029 | 0.545 ± 0.034 | 0.792 ± 0.128 |
| Qwen-2.5-1.5B-Inst | 1.515 ± 0.033 | 47.085 ± 1.063 | 2.108 ± 0.047 | 0.607 ± 0.038 | 0.971 ± 0.050 | 1.300 ± 0.041 | 0.926 ± 0.128 |
| Qwen-2.5-1.5B | 1.070 ± 0.028 | 41.199 ± 1.063 | 1.296 ± 0.044 | 0.272 ± 0.019 | 0.650 ± 0.011 | 0.855 ± 0.036 | 0.785 ± 0.128 |
| Qwen-2.5-0.5B-Inst | 1.515 ± 0.033 | 47.085 ± 1.063 | 2.108 ± 0.047 | 0.607 ± 0.038 | 0.971 ± 0.050 | 1.300 ± 0.041 | 0.926 ± 0.128 |
| Qwen-2.5-0.5B | 1.821 ± 0.027 | 47.768 ± 0.703 | 2.252 ± 0.042 | 1.480 ± 0.054 | 1.484 ± 0.024 | 1.642 ± 0.034 | 1.004 ± 0.129 |
| Multimodal Models | | | | | | | |
| UniTime | 0.405 ± 0.002 | 37.250 ± 0.178 | 0.460 ± 0.003 | 0.178 ± 0.001 | 0.330 ± 0.003 | 0.384 ± 0.002 | 0.443 ± 0.003 |
| Time-LLM (ETTh1) | 0.458 ± 0.002 | 43.016 ± 0.164 | 0.487 ± 0.002 | 0.174 ± 0.000 | 0.406 ± 0.004 | 0.419 ± 0.001 | 0.465 ± 0.001 |
| ChatTime-Base | 0.725 ± 0.002 | 38.762 ± 0.353 | 0.658 ± 0.003 | 0.171 ± 0.001 | 0.367 ± 0.003 | 0.783 ± 0.003 | 2.719 ± 0.005 |
| ChatTime-Chat | 0.781 ± 0.015 | 35.267 ± 0.436 | 0.741 ± 0.024 | 0.160 ± 0.001 | 0.425 ± 0.035 | 0.791 ± 0.001 | 2.882 ± 0.000 |
| TS Foundation Models | | | | | | | |
| Lag-Llama | 0.327 ± 0.004 | 30.446 ± 0.818 | 0.330 ± 0.005 | 0.167 ± 0.005 | 0.292 ± 0.009 | 0.294 ± 0.004 | 0.495 ± 0.014 |
| Chronos-Tiny | 0.328 ± 0.001 | 27.495 ± 0.440 | 0.302 ± 0.002 | 0.163 ± 0.002 | 0.393 ± 0.002 | 0.264 ± 0.002 | 0.486 ± 0.003 |
| Chronos-Mini | 0.341 ± 0.001 | 28.892 ± 0.425 | 0.318 ± 0.002 | 0.171 ± 0.003 | 0.407 ± 0.002 | 0.272 ± 0.002 | 0.481 ± 0.004 |
| Chronos-Small | 0.328 ± 0.002 | 26.528 ± 0.371 | 0.308 ± 0.002 | 0.179 ± 0.002 | 0.393 ± 0.003 | 0.257 ± 0.002 | 0.453 ± 0.007 |
| Chronos-Base | 0.672 ± 0.003 | 30.592 ± 0.377 | 0.570 ± 0.002 | 0.211 ± 0.005 | 0.392 ± 0.002 | 0.697 ± 0.003 | 2.481 ± 0.013 |
| Chronos-Large | 0.326 ± 0.002 | 25.600 ± 0.401 | 0.314 ± 0.002 | 0.179 ± 0.003 | 0.379 ± 0.003 | 0.255 ± 0.002 | 0.460 ± 0.004 |
| TimeGEN | 0.353 ± 0.000 | 35.925 ± 0.168 | 0.332 ± 0.000 | 0.177 ± 0.000 | 0.405 ± 0.000 | 0.292 ± 0.000 | 0.474 ± 0.000 |
| Moirai-small | 0.565 ± 0.031 | 36.048 ± 0.438 | 0.662 ± 0.050 | 0.195 ± 0.010 | 0.434 ± 0.002 | 0.558 ± 0.040 | 0.464 ± 0.013 |
| Moirai-base | 0.624 ± 0.013 | 35.261 ± 0.410 | 0.629 ± 0.021 | 0.172 ± 0.002 | 0.399 ± 0.004 | 0.630 ± 0.017 | 0.486 ± 0.015 |
| Moirai-large | 0.520 ± 0.006 | 28.636 ± 0.870 | 0.596 ± 0.009 | 0.140 ± 0.001 | 0.431 ± 0.002 | 0.499 ± 0.002 | 0.438 ± 0.011 |
| Statistical Models | | | | | | | |
| ARIMA | 0.475 ± 0.006 | 27.039 ± 0.483 | 0.557 ± 0.009 | 0.200 ± 0.007 | 0.350 ± 0.003 | 0.375 ± 0.006 | 0.440 ± 0.011 |
| ETS | 0.530 ± 0.009 | 33.786 ± 0.635 | 0.639 ± 0.014 | 0.362 ± 0.014 | 0.315 ± 0.006 | 0.402 ± 0.010 | 0.508 ± 0.017 |
| Exp-Smoothing | 0.605 ± 0.013 | 36.426 ± 0.349 | 0.702 ± 0.020 | 0.493 ± 0.016 | 0.397 ± 0.006 | 0.480 ± 0.015 | 0.827 ± 0.060 |

Tab. 4 provides the results of all tested models, partitioned by the types of context.

## C.3. Full results partitioned by model capabilities

We provide an additional view of the results of all models in Tab. 5, partitioned by model capabilities.

*Table 5.* Results of all models on the CiK benchmark. Starting from the left, the first column shows the RCRPS averaged over all tasks. The second column shows the rank of each method w.r.t. other baselines, averaged over all tasks. The remaining columns show the average RCRPS stratified by model capabilities. All averages are weighted according to the scheme described in Sec. 5.1 and accompanied by standard errors. Lower is better and the best averages are in bold.

| Model | Average RCRPS | Average Rank | Common-Sense | Retrieval From Context | Retrieval From Memory | Reasoning Deductive | Reasoning Analogical | Reasoning Mathematical | Reasoning Causal |
|---|---|---|---|---|---|---|---|---|---|
| **With Context** | | | | | | | | | |
| DIRECT PROMPT (ours) | | | | | | | | | |
| Llama-3.1-405B-Inst | **0.159 ± 0.008** | **7.971 ± 0.585** | **0.140 ± 0.013** | 0.109 ± 0.002 | **0.191 ± 0.006** | 0.132 ± 0.001 | **0.167 ± 0.008** | 0.316 ± 0.028 | 0.376 ± 0.039 |
| Llama-3-70B-Inst | 0.286 ± 0.004 | 14.802 ± 0.203 | 0.323 ± 0.008 | 0.122 ± 0.003 | 0.408 ± 0.012 | 0.168 ± 0.002 | 0.492 ± 0.019 | 0.473 ± 0.012 | 0.577 ± 0.017 |
| Llama-3-8B-Inst | 0.461 ± 0.008 | 28.016 ± 0.584 | 0.323 ± 0.010 | 0.174 ± 0.003 | 0.849 ± 0.021 | 0.407 ± 0.014 | 1.245 ± 0.039 | 0.437 ± 0.004 | 0.494 ± 0.004 |
| Mixtral-8x7B-Inst | 0.523 ± 0.023 | 33.069 ± 0.413 | 0.433 ± 0.043 | 0.204 ± 0.000 | 0.864 ± 0.029 | 0.426 ± 0.024 | 1.245 ± 0.006 | 0.644 ± 0.080 | 0.789 ± 0.112 |
| Qwen-2.5-7B-Inst | 0.290 ± 0.003 | 22.852 ± 0.734 | 0.343 ± 0.005 | 0.127 ± 0.002 | 0.324 ± 0.008 | 0.205 ± 0.005 | 0.281 ± 0.014 | 0.409 ± 0.002 | 0.480 ± 0.002 |
| Qwen-2.5-1.5B-Inst | 0.575 ± 0.014 | 32.631 ± 0.861 | 0.610 ± 0.007 | 0.214 ± 0.004 | 0.988 ± 0.049 | 0.344 ± 0.009 | 1.077 ± 0.122 | 0.896 ± 0.006 | 1.151 ± 0.008 |
| Qwen-2.5-0.5B-Inst | 0.463 ± 0.012 | 27.221 ± 0.533 | 0.267 ± 0.008 | 1.029 ± 0.055 | 0.744 ± 0.039 | 0.244 ± 0.007 | 2.043 ± 0.104 | 0.330 ± 0.004 | 0.392 ± 0.005 |
| GPT-4o | 0.274 ± 0.010 | 8.640 ± 0.436 | 0.179 ± 0.004 | **0.087 ± 0.003** | 0.519 ± 0.029 | **0.110 ± 0.006** | 0.447 ± 0.029 | 0.590 ± 0.033 | 0.769 ± 0.046 |
| GPT-4o-mini | 0.354 ± 0.022 | 17.573 ± 0.505 | 0.296 ± 0.043 | 0.419 ± 0.014 | 0.471 ± 0.012 | 0.219 ± 0.005 | 1.024 ± 0.033 | 0.475 ± 0.080 | 0.578 ± 0.112 |
| LLMP | | | | | | | | | |
| Llama-3-70B-Inst | 0.539 ± 0.013 | 18.042 ± 0.572 | 0.641 ± 0.018 | 0.284 ± 0.015 | 0.392 ± 0.014 | 0.495 ± 0.025 | 0.312 ± 0.019 | 0.453 ± 0.020 | 0.495 ± 0.028 |
| Llama-3-70B | 0.236 ± 0.006 | 12.377 ± 0.723 | 0.309 ± 0.011 | 0.126 ± 0.009 | 0.217 ± 0.007 | 0.132 ± 0.003 | 0.241 ± 0.019 | 0.294 ± 0.008 | **0.329 ± 0.010** |
| Llama-3-8B-Inst | 0.483 ± 0.010 | 18.585 ± 0.477 | 0.345 ± 0.002 | 0.138 ± 0.004 | 0.910 ± 0.030 | 0.242 ± 0.008 | 1.278 ± 0.069 | 0.617 ± 0.022 | 0.787 ± 0.030 |
| Llama-3-8B | 0.311 ± 0.023 | 18.634 ± 1.039 | 0.403 ± 0.043 | 0.124 ± 0.003 | 0.280 ± 0.026 | 0.177 ± 0.014 | 0.267 ± 0.015 | 0.530 ± 0.084 | 0.661 ± 0.117 |
| Mixtral-8x7B-Inst | 0.264 ± 0.004 | 16.078 ± 0.666 | 0.344 ± 0.004 | 0.127 ± 0.003 | 0.224 ± 0.005 | 0.179 ± 0.010 | 0.173 ± 0.009 | 0.348 ± 0.005 | 0.405 ± 0.007 |
| Mixtral-8x7B | 0.262 ± 0.008 | 16.302 ± 0.523 | 0.348 ± 0.012 | 0.146 ± 0.022 | 0.230 ± 0.016 | 0.153 ± 0.002 | 0.230 ± 0.041 | 0.354 ± 0.007 | 0.414 ± 0.009 |
| Qwen-2.5-7B-Inst | 1.974 ± 0.027 | 45.233 ± 0.739 | 1.816 ± 0.048 | 1.022 ± 0.054 | 2.215 ± 0.046 | 2.758 ± 0.024 | 1.723 ± 0.092 | 2.025 ± 0.093 | 1.607 ± 0.127 |
| Qwen-2.5-7B | 0.910 ± 0.037 | 38.157 ± 1.041 | 0.691 ± 0.063 | 0.794 ± 0.083 | 1.558 ± 0.062 | 0.893 ± 0.028 | 2.328 ± 0.153 | 0.878 ± 0.084 | 0.881 ± 0.113 |
| Qwen-2.5-1.5B-Inst | 2.158 ± 0.027 | 50.654 ± 0.863 | 2.056 ± 0.046 | 1.566 ± 0.033 | 2.671 ± 0.038 | 2.165 ± 0.035 | 3.635 ± 0.053 | 2.480 ± 0.085 | 2.323 ± 0.113 |
| Qwen-2.5-1.5B | 1.731 ± 0.036 | 45.118 ± 0.528 | 1.343 ± 0.061 | 1.737 ± 0.074 | 2.594 ± 0.042 | 2.256 ± 0.042 | 3.275 ± 0.132 | 2.036 ± 0.083 | 1.526 ± 0.114 |
| Qwen-2.5-0.5B-Inst | 1.937 ± 0.024 | 50.482 ± 0.612 | 1.740 ± 0.043 | 1.800 ± 0.021 | 2.193 ± 0.025 | 2.305 ± 0.028 | 3.439 ± 0.004 | 1.685 ± 0.084 | 1.398 ± 0.114 |
| Qwen-2.5-0.5B | 1.995 ± 0.024 | 48.507 ± 0.840 | 1.829 ± 0.045 | 0.950 ± 0.025 | 1.967 ± 0.020 | 2.809 ± 0.023 | 1.804 ± 0.036 | 1.695 ± 0.085 | 1.443 ± 0.113 |
| Multimodal Models | | | | | | | | | |
| UniTime | 0.370 ± 0.001 | 35.453 ± 0.152 | 0.267 ± 0.002 | 0.179 ± 0.001 | 0.321 ± 0.001 | 0.511 ± 0.003 | 0.337 ± 0.001 | 0.333 ± 0.001 | 0.385 ± 0.001 |
| Time-LLM (ETTh1) | 0.476 ± 0.001 | 44.084 ± 0.150 | 0.448 ± 0.002 | 0.192 ± 0.000 | 0.373 ± 0.000 | 0.538 ± 0.003 | 0.397 ± 0.001 | 0.382 ± 0.001 | 0.440 ± 0.001 |
| ChatTime-Base | 0.735 ± 0.002 | 39.033 ± 0.312 | 0.843 ± 0.002 | 0.216 ± 0.002 | 1.099 ± 0.002 | 0.263 ± 0.004 | 0.374 ± 0.004 | 1.788 ± 0.002 | 2.407 ± 0.002 |
| ChatTime-Chat | 0.747 ± 0.005 | 34.182 ± 0.389 | 0.825 ± 0.004 | 0.299 ± 0.020 | 1.198 ± 0.015 | 0.305 ± 0.012 | 0.277 ± 0.003 | 2.015 ± 0.016 | 2.691 ± 0.022 |
| **Without Context** | | | | | | | | | |
| DIRECT PROMPT (ours) | | | | | | | | | |
| Llama-3.1-405B-Inst | 0.473 ± 0.005 | 34.336 ± 0.294 | 0.393 ± 0.002 | 0.325 ± 0.006 | 0.752 ± 0.015 | 0.494 ± 0.009 | 0.720 ± 0.027 | 0.594 ± 0.006 | 0.617 ± 0.006 |
| Llama-3-70B-Inst | 0.470 ± 0.008 | 35.143 ± 0.357 | 0.404 ± 0.002 | 0.304 ± 0.007 | 0.717 ± 0.015 | 0.488 ± 0.022 | 0.694 ± 0.024 | 0.606 ± 0.007 | 0.631 ± 0.007 |
| Llama-3-8B-Inst | 0.602 ± 0.006 | 41.873 ± 0.398 | 0.390 ± 0.005 | 0.322 ± 0.004 | 1.123 ± 0.018 | 0.643 ± 0.012 | 1.446 ± 0.035 | 0.581 ± 0.005 | 0.617 ± 0.005 |
| Mixtral-8x7B-Inst | 0.712 ± 0.021 | 46.809 ± 0.376 | 0.624 ± 0.043 | 0.324 ± 0.006 | 1.053 ± 0.007 | 0.783 ± 0.005 | 1.237 ± 0.007 | 0.858 ± 0.080 | 0.872 ± 0.112 |
| Qwen-2.5-7B-Inst | 0.421 ± 0.022 | 37.154 ± 0.546 | 0.447 ± 0.043 | 0.259 ± 0.008 | 0.505 ± 0.010 | 0.375 ± 0.007 | 0.411 ± 0.016 | 0.692 ± 0.080 | 0.792 ± 0.112 |
| Qwen-2.5-1.5B-Inst | 0.450 ± 0.007 | 36.308 ± 0.619 | 0.377 ± 0.006 | 0.232 ± 0.004 | 0.661 ± 0.018 | 0.387 ± 0.013 | 0.939 ± 0.039 | 0.423 ± 0.007 | 0.476 ± 0.008 |
| Qwen-2.5-0.5B-Inst | 0.305 ± 0.007 | 26.819 ± 0.351 | 0.267 ± 0.010 | 0.162 ± 0.001 | 0.384 ± 0.008 | 0.300 ± 0.011 | 0.440 ± 0.016 | **0.315 ± 0.002** | 0.367 ± 0.002 |
| GPT-4o | 0.441 ± 0.002 | 31.507 ± 0.385 | 0.381 ± 0.002 | 0.179 ± 0.002 | 0.692 ± 0.028 | 0.357 ± 0.007 | 0.953 ± 0.067 | 0.422 ± 0.003 | 0.471 ± 0.002 |
| GPT-4o-mini | 0.423 ± 0.006 | 35.715 ± 0.275 | 0.359 ± 0.003 | 0.214 ± 0.002 | 0.649 ± 0.019 | 0.391 ± 0.008 | 0.771 ± 0.041 | 0.461 ± 0.003 | 0.511 ± 0.003 |
| LLMP | | | | | | | | | |
| Llama-3-70B-Inst | 0.378 ± 0.004 | 26.031 ± 0.474 | 0.368 ± 0.003 | 0.150 ± 0.003 | 0.513 ± 0.012 | 0.292 ± 0.006 | 0.668 ± 0.025 | 0.384 ± 0.003 | 0.440 ± 0.003 |
| Llama-3-70B | 0.311 ± 0.006 | 21.810 ± 0.464 | 0.349 ± 0.011 | 0.141 ± 0.002 | 0.351 ± 0.008 | 0.215 ± 0.004 | 0.395 ± 0.017 | 0.372 ± 0.004 | 0.434 ± 0.005 |
| Llama-3-8B-Inst | 0.503 ± 0.009 | 31.156 ± 0.438 | 0.385 ± 0.003 | 0.159 ± 0.002 | 0.914 ± 0.030 | 0.431 ± 0.010 | 1.271 ± 0.070 | 0.424 ± 0.003 | 0.467 ± 0.001 |
| Llama-3-8B | 0.345 ± 0.003 | 25.067 ± 0.381 | 0.326 ± 0.004 | 0.150 ± 0.002 | 0.497 ± 0.009 | 0.266 ± 0.005 | 0.640 ± 0.017 | 0.381 ± 0.002 | 0.444 ± 0.002 |
| Mixtral-8x7B-Inst | 0.383 ± 0.015 | 24.582 ± 0.471 | 0.357 ± 0.003 | 0.550 ± 0.072 | 0.459 ± 0.047 | 0.211 ± 0.008 | 1.027 ± 0.133 | 0.371 ± 0.004 | 0.428 ± 0.004 |
| Mixtral-8x7B | 0.306 ± 0.007 | 22.560 ± 0.512 | 0.360 ± 0.013 | 0.146 ± 0.002 | 0.327 ± 0.008 | 0.202 ± 0.005 | 0.340 ± 0.016 | 0.382 ± 0.002 | 0.445 ± 0.003 |
| Qwen-2.5-7B-Inst | 1.020 ± 0.026 | 39.942 ± 1.233 | 0.521 ± 0.045 | 1.157 ± 0.022 | 1.634 ± 0.017 | 1.061 ± 0.034 | 3.319 ± 0.037 | 0.853 ± 0.081 | 0.769 ± 0.112 |
| Qwen-2.5-7B | 0.732 ± 0.030 | 38.109 ± 1.185 | 0.649 ± 0.048 | 0.359 ± 0.036 | 0.974 ± 0.037 | 0.751 ± 0.039 | 1.433 ± 0.096 | 0.728 ± 0.084 | 0.730 ± 0.112 |
| Qwen-2.5-1.5B-Inst | 1.515 ± 0.033 | 47.071 ± 1.066 | 1.316 ± 0.055 | 1.159 ± 0.057 | 1.802 ± 0.031 | 1.652 ± 0.042 | 3.383 ± 0.106 | 1.108 ± 0.082 | 0.848 ± 0.112 |
| Qwen-2.5-1.5B | 1.070 ± 0.028 | 41.187 ± 1.053 | 1.005 ± 0.048 | 0.287 ± 0.026 | 1.339 ± 0.023 | 1.264 ± 0.040 | 1.798 ± 0.047 | 0.771 ± 0.086 | 0.720 ± 0.112 |
| Qwen-2.5-0.5B-Inst | 1.318 ± 0.037 | 44.393 ± 0.750 | 1.464 ± 0.064 | 0.239 ± 0.031 | 1.192 ± 0.019 | 1.433 ± 0.047 | 1.675 ± 0.072 | 0.930 ± 0.082 | 0.743 ± 0.112 |
| Qwen-2.5-0.5B | 1.821 ± 0.027 | 47.763 ± 0.715 | 1.705 ± 0.045 | 0.572 ± 0.040 | 1.722 ± 0.033 | 2.498 ± 0.036 | 1.671 ± 0.064 | 1.492 ± 0.083 | 1.113 ± 0.114 |
| Multimodal Models | | | | | | | | | |
| UniTime | 0.405 ± 0.002 | 37.248 ± 0.177 | 0.361 ± 0.002 | 0.166 ± 0.001 | 0.319 ± 0.001 | 0.496 ± 0.005 | 0.314 ± 0.002 | 0.352 ± 0.002 | 0.409 ± 0.003 |
| Time-LLM (ETTh1) | 0.458 ± 0.002 | 43.014 ± 0.164 | 0.440 ± 0.003 | 0.191 ± 0.000 | 0.371 ± 0.000 | 0.499 ± 0.002 | 0.399 ± 0.001 | 0.368 ± 0.001 | 0.424 ± 0.001 |
| ChatTime-Base | 0.725 ± 0.002 | 38.771 ± 0.351 | 0.837 ± 0.002 | 0.205 ± 0.003 | 1.090 ± 0.004 | 0.250 ± 0.004 | 0.365 ± 0.006 | 1.779 ± 0.003 | 2.398 ± 0.004 |
| ChatTime-Chat | 0.781 ± 0.015 | 35.260 ± 0.446 | 0.865 ± 0.022 | 0.165 ± 0.002 | 1.217 ± 0.036 | 0.282 ± 0.020 | 0.445 ± 0.064 | 1.896 ± 0.001 | 2.536 ± 0.000 |
| TS Foundation Models | | | | | | | | | |
| Lag-Llama | 0.327 ± 0.004 | 30.451 ± 0.819 | 0.353 ± 0.007 | 0.181 ± 0.003 | 0.324 ± 0.003 | 0.269 ± 0.006 | 0.342 ± 0.006 | 0.386 ± 0.009 | 0.449 ± 0.012 |
| Chronos-Tiny | 0.328 ± 0.001 | 27.487 ± 0.441 | 0.400 ± 0.002 | 0.144 ± 0.003 | 0.297 ± 0.002 | 0.229 ± 0.002 | 0.290 ± 0.005 | 0.382 ± 0.002 | 0.440 ± 0.003 |
| Chronos-Mini | 0.341 ± 0.001 | 28.893 ± 0.428 | 0.412 ± 0.002 | 0.147 ± 0.002 | 0.302 ± 0.002 | 0.248 ± 0.002 | 0.305 ± 0.004 | 0.378 ± 0.003 | 0.436 ± 0.004 |
| Chronos-Small | 0.328 ± 0.002 | 26.524 ± 0.372 | 0.388 ± 0.003 | 0.144 ± 0.002 | 0.287 ± 0.002 | 0.248 ± 0.003 | 0.290 ± 0.003 | 0.358 ± 0.005 | 0.412 ± 0.006 |
| Chronos-Base | 0.672 ± 0.003 | 30.601 ± 0.375 | 0.702 ± 0.002 | 0.143 ± 0.002 | 1.023 ± 0.006 | 0.261 ± 0.003 | 0.299 ± 0.004 | 1.643 ± 0.003 | 2.187 ± 0.012 |
| Chronos-Large | 0.326 ± 0.002 | 25.602 ± 0.399 | 0.385 ± 0.002 | 0.138 ± 0.002 | 0.288 ± 0.002 | 0.249 ± 0.002 | 0.295 ± 0.003 | 0.362 ± 0.003 | 0.417 ± 0.004 |
| TimeGEN | 0.353 ± 0.000 | 35.924 ± 0.167 | 0.401 ± 0.000 | 0.176 ± 0.000 | 0.308 ± 0.000 | 0.278 ± 0.000 | 0.324 ± 0.000 | 0.377 ± 0.000 | 0.431 ± 0.000 |
| Moirai-small | 0.565 ± 0.031 | 36.038 ± 0.438 | 0.429 ± 0.005 | 0.671 ± 0.146 | 0.468 ± 0.076 | 0.566 ± 0.017 | 1.204 ± 0.271 | 0.376 ± 0.009 | 0.426 ± 0.012 |
| Moirai-base | 0.624 ± 0.013 | 35.263 ± 0.407 | 0.410 ± 0.006 | 0.600 ± 0.053 | 0.680 ± 0.028 | 0.690 ± 0.019 | 1.147 ± 0.099 | 0.375 ± 0.010 | 0.441 ± 0.013 |
| Moirai-large | 0.520 ± 0.006 | 28.635 ± 0.862 | 0.414 ± 0.004 | 0.155 ± 0.004 | 0.260 ± 0.003 | 0.751 ± 0.015 | 0.276 ± 0.008 | 0.337 ± 0.007 | 0.397 ± 0.010 |
| Statistical Models | | | | | | | | | |
| ARIMA | 0.475 ± 0.006 | 27.047 ± 0.485 | 0.395 ± 0.005 | 0.160 ± 0.002 | 0.517 ± 0.012 | 0.513 ± 0.012 | 0.706 ± 0.026 | 0.354 ± 0.007 | 0.403 ± 0.010 |
| ETS | 0.530 ± 0.009 | 33.786 ± 0.635 | 0.418 ± 0.009 | 0.228 ± 0.010 | 0.682 ± 0.018 | 0.577 ± 0.019 | 0.855 ± 0.035 | 0.453 ± 0.012 | 0.479 ± 0.015 |
| Exp-Smoothing | 0.605 ± 0.013 | 36.425 ± 0.346 | 0.569 ± 0.021 | 0.334 ± 0.013 | 0.743 ± 0.018 | 0.563 ± 0.016 | 0.899 ± 0.035 | 0.673 ± 0.038 | 0.782 ± 0.053 |

## C.4. Inference Time

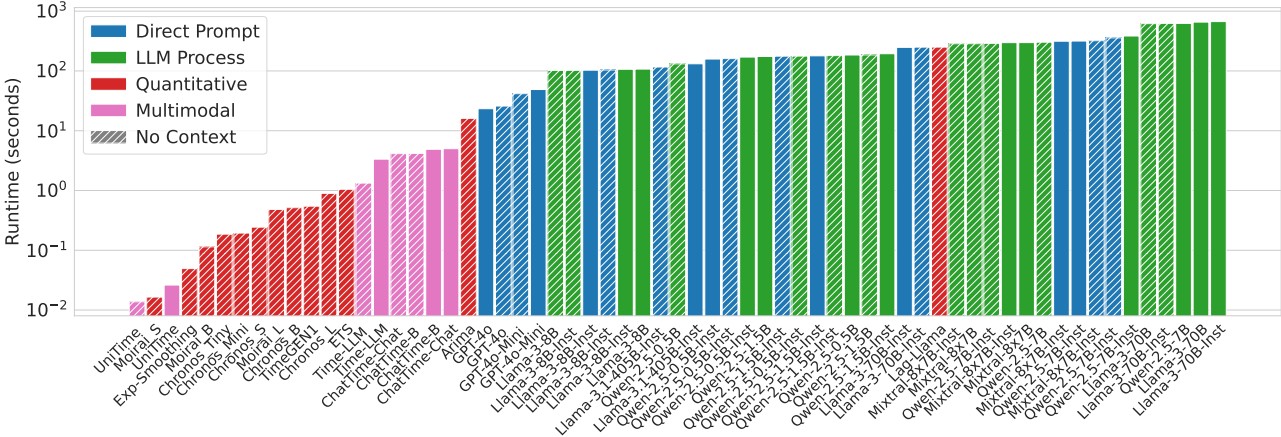

*Figure 13.* Inference time in seconds, for all baselines, averaged over all tasks. Several quantitative methods are much faster on average than LLM-based methods. However, there are significant differences in inference time between the LLM-based forecasters: for the `Llama` models, LLMP takes about an order of magnitude more time to run on average than DIRECT PROMPT.

Figure 13 provides the inference time of all tested models on the benchmark. Note that these values have not been normalized based on the computing resources made available to each model during inference; please refer to Appendix D for information about how much compute resources were allocated to each of them.

## C.5. Significant failures per model

We observe that in a few instances in the benchmark, some models tend to obtain significantly worse performance when evaluated with context. In our evaluation, we term all instances where the RCRPS value of a model is greater than 5, as significant failures of the model on those instances. We found 5 as a suitable value for analyzing such failures, as it intuitively represents the value a forecast would get if the distance between the forecast and the ground-truth was 5 times bigger than the range of the ground-truth for the task. When we aggregate the RCRPS of instances in the benchmark (such as in Tab. 1), we cap the RCRPS of such significant failures to 5, to avoid outliers with a much higher RCRPS affecting the aggregate score. In Tab. 6, we show the number of such instances in our evaluation of the benchmark where we found models to have significant failures (out of a total of 355 evaluated instances). Interestingly, some models such as DIRECT PROMPT with `Llama-3.1-405B-Instruct` and LLMP with `Llama-3-70B` and `Llama-3-8B` are more robust to such significant failures, and do not incur such failures. On the other hand, models such as `Qwen` family of models (that are notably significantly smaller than the rest) with LLMP achieve the most significant failures, followed by `Llama-3-70B-Instruct` and `Llama-3-8B-Instruct` with LLMP. We postulate that this is because of models misinterpreting context. It is still an open question as to how to increase the robustness of models to prevent or reduce such significant failures. We visualize such significant failures in Appendix C.12.

*Table 6.* Number of instances with significant failures in models that support context

| Model | Number of instances with significant failures |
|---|---|
| **With Context** | |
| DIRECT PROMPT (ours) | |
| Llama-3.1-405B-Inst | 0 |
| Llama-3-70B-Inst | 1 |
| Llama-3-8B-Inst | 2 |
| Mixtral-8x7B-Inst | 7 |
| Qwen-2.5-7B-Inst | 1 |
| Qwen-2.5-1.5B-Inst | 10 |
| Qwen-2.5-0.5B-Inst | 7 |
| GPT-4o | 5 |
| GPT-4o-mini | 0 |
| LLMP | |
| Llama-3-70B-Inst | 18 |
| Llama-3-70B | 0 |
| Llama-3-8B-Inst | 12 |
| Llama-3-8B | 0 |
| Mixtral-8x7B-Inst | 1 |
| Mixtral-8x7B | 0 |
| Qwen-2.5-7B-Inst | 107 |
| Qwen-2.5-7B | 27 |
| Qwen-2.5-1.5B-Inst | 93 |
| Qwen-2.5-1.5B | 99 |
| Qwen-2.5-0.5B-Inst | 100 |
| Qwen-2.5-0.5B | 109 |
| Multimodal Models | |
| UniTime | 0 |
| Time-LLM (ETTh1) | 2 |
| ChatTime-Base | 21 |
| ChatTime-Chat | 23 |
| **Without Context** | |
| DIRECT PROMPT (ours) | |
| Llama-3.1-405B-Inst | 9 |
| Llama-3-70B-Inst | 4 |
| Llama-3-8B-Inst | 8 |
| Mixtral-8x7B-Inst | 14 |
| Qwen-2.5-7B-Inst | 3 |
| Qwen-2.5-1.5B-Inst | 5 |
| Qwen-2.5-0.5B-Inst | 0 |
| GPT-4o | 2 |
| GPT-4o-mini | 1 |
| LLMP | |
| Llama-3-70B-Inst | 1 |
| Llama-3-70B | 0 |
| Llama-3-8B-Inst | 9 |
| Llama-3-8B | 3 |
| Mixtral-8x7B-Inst | 3 |
| Mixtral-8x7B | 0 |
| Qwen-2.5-7B-Inst | 35 |
| Qwen-2.5-7B | 15 |
| Qwen-2.5-1.5B-Inst | 52 |
| Qwen-2.5-1.5B | 36 |
| Qwen-2.5-0.5B-Inst | 42 |
| Qwen-2.5-0.5B | 74 |
| Multimodal Models | |
| UniTime | 1 |
| Time-LLM (ETTh1) | 1 |
| ChatTime-Base | 21 |
| ChatTime-Chat | 23 |
| TS Foundation Models | |
| Lag-Llama | 1 |
| Chronos-Tiny | 2 |
| Chronos-Mini | 2 |
| Chronos-Small | 1 |
| Chronos-Base | 18 |
| Chronos-Large | 1 |
| TimeGEN | 2 |
| Moirai-Small | 3 |
| Moirai-Base | 8 |
| Moirai-Large | 7 |
| Statistical Models | |
| ARIMA | 2 |
| ETS | 1 |
| Exp-Smoothing | 5 |

## C.6. Testing the Statistical Significance of the Relevance of Context

Table 7. $p$-value of the one-sided paired $t$-test between the RCPRS values with and without context for models who can use it. Since this test is done on the unweighted RCRPS values, the average RCRPS presented in this table are also unweighted.

| Model | Average RCRPS With context | Average RCRPS Without context | $p$-value |
|---|---|---|---|
| DIRECT PROMPT - Llama-3.1-405B-Inst | $0.165 \pm 0.005$ | $0.544 \pm 0.007$ | $6.92 \times 10^{-13}$ |
| LLMP - Llama-3-70B | $0.191 \pm 0.004$ | $0.249 \pm 0.004$ | $1.85 \times 10^{-9}$ |
| LLMP - Mixtral-8x7B | $0.202 \pm 0.005$ | $0.245 \pm 0.004$ | $9.17 \times 10^{-8}$ |
| LLMP - Llama-3-8B | $0.214 \pm 0.007$ | $0.283 \pm 0.003$ | $4.66 \times 10^{-4}$ |
| LLMP - Mixtral-8x7B-Inst | $0.223 \pm 0.002$ | $0.290 \pm 0.009$ | $0.002$ |
| DIRECT PROMPT - Qwen-2.5-7B-Inst | $0.244 \pm 0.003$ | $0.403 \pm 0.009$ | $7.99 \times 10^{-8}$ |
| DIRECT PROMPT - Llama-3-70B-Inst | $0.246 \pm 0.003$ | $0.529 \pm 0.010$ | $1.07 \times 10^{-10}$ |
| DIRECT PROMPT - GPT-4o-mini | $0.250 \pm 0.003$ | $0.403 \pm 0.005$ | $2.85 \times 10^{-8}$ |
| DIRECT PROMPT - GPT-4o | $0.252 \pm 0.010$ | $0.387 \pm 0.007$ | $6.21 \times 10^{-4}$ |
| UniTime | $0.290 \pm 0.001$ | $0.321 \pm 0.001$ | $0.016$ |
| DIRECT PROMPT - Qwen-2.5-0.5B-Inst | $0.343 \pm 0.011$ | $0.258 \pm 0.004$ | $0.987$ |
| Time-LLM (ETTh1) | $0.378 \pm 0.001$ | $0.364 \pm 0.001$ | $1 - 8.08 \times 10^{-7}$ |
| DIRECT PROMPT - Mixtral-8x7B-Inst | $0.413 \pm 0.007$ | $0.699 \pm 0.006$ | $1.88 \times 10^{-6}$ |
| LLMP - Llama-3-8B-Inst | $0.413 \pm 0.009$ | $0.432 \pm 0.010$ | $0.287$ |
| DIRECT PROMPT - Llama-3-8B-Inst | $0.416 \pm 0.007$ | $0.631 \pm 0.007$ | $3.31 \times 10^{-10}$ |
| DIRECT PROMPT - Qwen-2.5-1.5B-Inst | $0.481 \pm 0.016$ | $0.406 \pm 0.006$ | $0.975$ |
| ChatTime-Chat | $0.557 \pm 0.001$ | $0.554 \pm 0.001$ | $0.086$ |
| ChatTime-Base | $0.568 \pm 0.001$ | $0.556 \pm 0.002$ | $1 - 1.35 \times 10^{-4}$ |
| LLMP - Llama-3-70B-Inst | $0.579 \pm 0.019$ | $0.313 \pm 0.003$ | $1 - 1.93 \times 10^{-5}$ |
| LLMP - Qwen-2.5-7B | $0.909 \pm 0.025$ | $0.618 \pm 0.018$ | $1 - 8.48 \times 10^{-6}$ |
| LLMP - Qwen-2.5-1.5B-Inst | $2.038 \pm 0.025$ | $1.181 \pm 0.022$ | $1$ |
| LLMP - Qwen-2.5-0.5B-Inst | $2.067 \pm 0.025$ | $1.047 \pm 0.017$ | $1$ |
| LLMP - Qwen-2.5-0.5B | $2.144 \pm 0.021$ | $1.766 \pm 0.025$ | $1 - 4.83 \times 10^{-8}$ |
| LLMP - Qwen-2.5-1.5B | $2.162 \pm 0.028$ | $0.861 \pm 0.016$ | $1$ |
| LLMP - Qwen-2.5-7B-Inst | $2.275 \pm 0.025$ | $0.895 \pm 0.014$ | $1$ |

To assess whether the lower average RCPRS using context than without using context we observe for the best performing model in our benchmark is statistically significant, we ran an analysis using the paired $t$-test. We used the paired $t$-test implemented in the `scipy` Python package as the `scipy.stats.ttest_rel` method, with `alternative="less"` to make the test one-sided. As can be seen in Tab. 7, for many models the improved RCPRS when using the context is statistically significant, with $p$-values lower than $\times 10^{-6}$ for 7 out of the 23 models under consideration. Furthermore, the best performing models are those for which the improvement is statistically significant, with the 9 best models all having $p$-values below 0.01.

## C.7. Cost of API-based models

Tab. 8 provides the cost incurred in evaluating GPT-4o (version gpt-4o-2024-05-13) and GPT-4o-mini (version gpt-4o-mini-2024-07-18) with the Direct Prompt method on CiK (as per the evaluation protocol used, described in Sec. 5.1).

*Table 8.* Costs ($CAD) of evaluating the `GPT-4o` family of models on CiK. "Total" represents the total cost of evaluating each model on the CiK benchmark. The "Per-instance average" and the "Per-instance median" are the average and median cost of running a single instance for a given task, in other words the average and median cost of generating 25 sample trajectories for a given example of a task. As a reminder, each task in CiK is evaluated over 5 instances in our evaluation protocol.

| Model | Total | Per-instance average | Per-instance median |
|---|---|---|---|
| GPT-4o | $143.83 | $0.288 | $0.170 |
| GPT-4o (no context) | $139.50 | $0.279 | $0.160 |
| GPT-4o-mini | $13.79 | $0.040 | $0.040 |
| GPT-4o-mini (no context) | $13.32 | $0.038 | $0.040 |

## C.8. Impact of Relevant and Irrelevant Information in Context

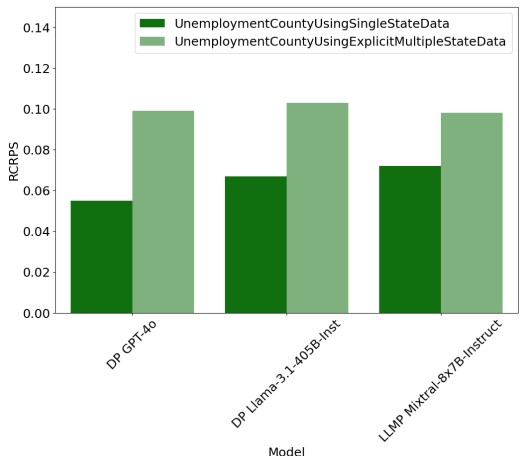

*Figure 14.* A comparison of RCRPS (lower is better) for two tasks on predicting the Unemployment Rate of a county. Both contain the context needed to solve the task. However, the `UnemploymentCountyUsingSingleStateData` task (dark green) is filtered to only contain the relevant context. Other the other hand, the `UnemploymentCountyUsingExpliciteMultipleStateData` task (light green) also contains other unrelated context. We visualize three models here, all of which perform better when the context only includes the most relevant information.

We study here if models perform better on context that has already been filtered to only contain relevant information. To assess this, we compare two tasks on predicting the Unemployment Rate of a county.

1. For the `UnemploymentCountyUsingSingleStateData` task, the context contains the unemployment rate of the state which the county belongs to, tagged with the name of the state. See https://servicenow.github.io/context-is-key-forecasting/v0/UnemploymentCountyUsingSingleStateData.html for a visualization.

2. In the `UnemploymentCountyUsingExpliciteMultipleStateData` task, in addition to the county's state unemployment rate, the context includes unemployment rates of 2 other randomly selected states, also tagged with state names. See https://servicenow.github.io/context-is-key-forecasting/v0/UnemploymentCountyUsingExplicitMultipleStateData.html for a visualization.

Results of three randomly picked models from the benchmark is visualized in Figure 14. We find that models perform much better when only the relevant state's data is provided, as opposed to the context also containing data from other states.

## C.9. Impact of Solely Irrelevant Information in Context

Many of our tasks include covariates in its context which are highly useful for the models to accurately predict the target time series. One question is: Do the LLM-based models perform well for such tasks due to correctly understanding that said covariates are helpful or because they blindly use the provided data without asking themselves if the data is actually relevant?

As a way to get some insight on this question, we took a task where the models have to forecast the unemployment data of an American county, given the unemployment data of the state the county is in (Task `UnemploymentCountyUsingSingleStateData`). We then modify this task by first trying to mislead the model by wrongly saying that the state-level data was from another state (without changing the data itself), then by giving the data from the other state (while explicitly telling the model that data is from said other state), before finally removing the state-level data altogether. The result for this experiment with 5 instances per task for DIRECT PROMPT - GPT-4o is shown in Tab. 9, while the forecasts for a single instance are shown in Figure 15. From these, we see that the model aggressively used data which is marked as being from an other state, even though if the data was actually from said other state, the performance would be closer to not having any state-level data. This shows that the model is liable to take any information provided as being useful, even though its usefulness is marginal.

*Table 9.* Ability of the DIRECT PROMPT - GPT-4o model to accurately predict the unemployment level of an American county, given various covariates. These results are averaged over 5 instances.

| Available data | RCPRS |
|---|---|
| Data from the correct state, accurately tagged | 0.0583 |
| Data from the correct state, inaccurately tagged | 0.0557 |
| Data from an incorrect state, accurately tagged | 0.1966 |
| No state-level data | 0.2630 |

## C.10. The effect of significant failures on the aggregate performance of models

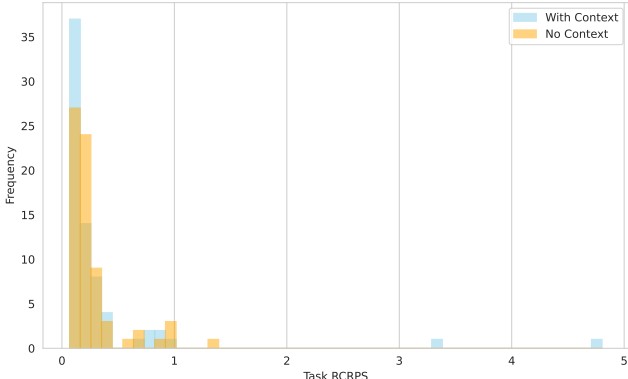

*Figure 16.* Histogram of the RCPRS (lower is better) of the DIRECT PROMPT - Qwen-2.5-0.5B-Instruct model on each task, with and without context. With context, the RCRPS is close to zero for a large number of tasks, but there is also a long tail of tasks with high RCRPS values, dominating and worsening the model's aggregate RCRPS.

As discussed in Sec. 5.4, in a few instances from the benchmark, some models return forecasts that miss the ground truth by a large margin, which we term significant failures (detailed in Appendix C.5). We analyse the effect of such significant failures on the results here. We use the DIRECT PROMPT - Mixtral 8x7B model as an example here, while the same phenomenon may apply to other models. In Figure 6, we can find that the aggregate RCRPS of DIRECT PROMPT - Mixtral 8x7B *worsens* when it uses context. However, in Figure 5 (left), the win rate of the model vs quantitative baselines *improves* when it uses context. These two figures show results that seem contradictory, but are in fact compatible: adding context improves the model's RCRPS for most tasks, but greatly worsens it for a minority of tasks where the model achieves significant failures.

To further illustrate this effect, we visualize the task-wise RCRPS of the DIRECT PROMPT - Mixtral-8x7B-Instruct model, both with and without context, in Figure 16. With context, the model gets an RCRPS close to zero in a large number of tasks. However, there is also a long tail of tasks with high RCRPS values with context, dominating and worsening the model's aggregate RCRPS.

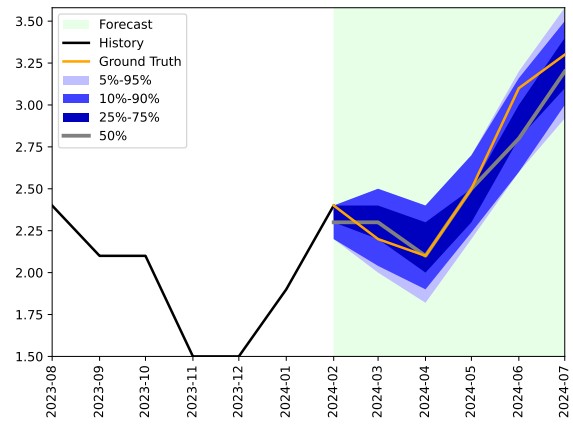

**(a)** *The task in our benchmark: the context contains the unemployment rate of the state the county is in, correctly tagged with the state name.*

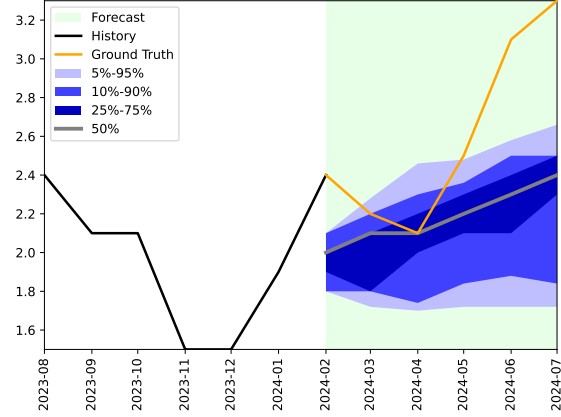

**(b)** *The context only mentions that this time series is an unemployment rate, and of which county it is. No state-level unemployement data is provided.*

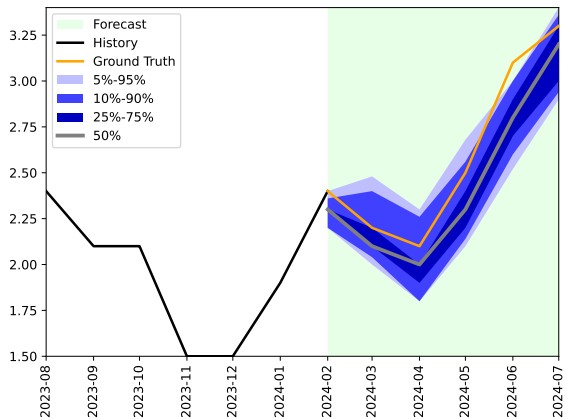

**(c)** *The state-level unemployment rate is incorrectly tagged as being from another state.*

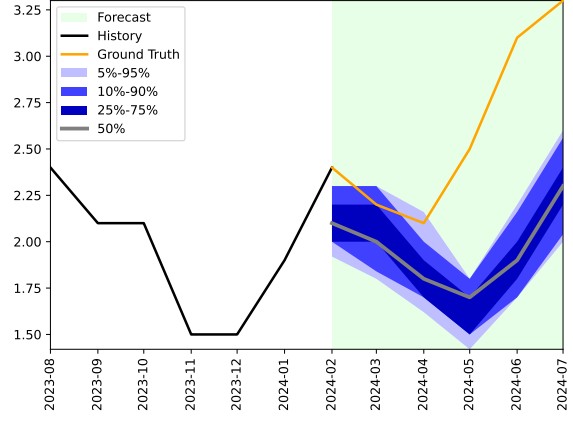

**(d)** *The context contains the unemployment rate of another state than the one the county is in, which is correctly tagged.*

*Figure 15.* Forecasts done by Direct Prompt - GPT-4o, with varying information in the context. The task is to forecast the unemployment rate of an American county.

## C.11. Visualizations of successful context-aware forecasts

**Context:** " This series represents the occupancy rate (%) captured by a highway sensor.
Consider that the meter will be offline for maintenance between 2024-04-11 13:00:00 and 2024-04-11 15:00:00, which results in zero readings. "

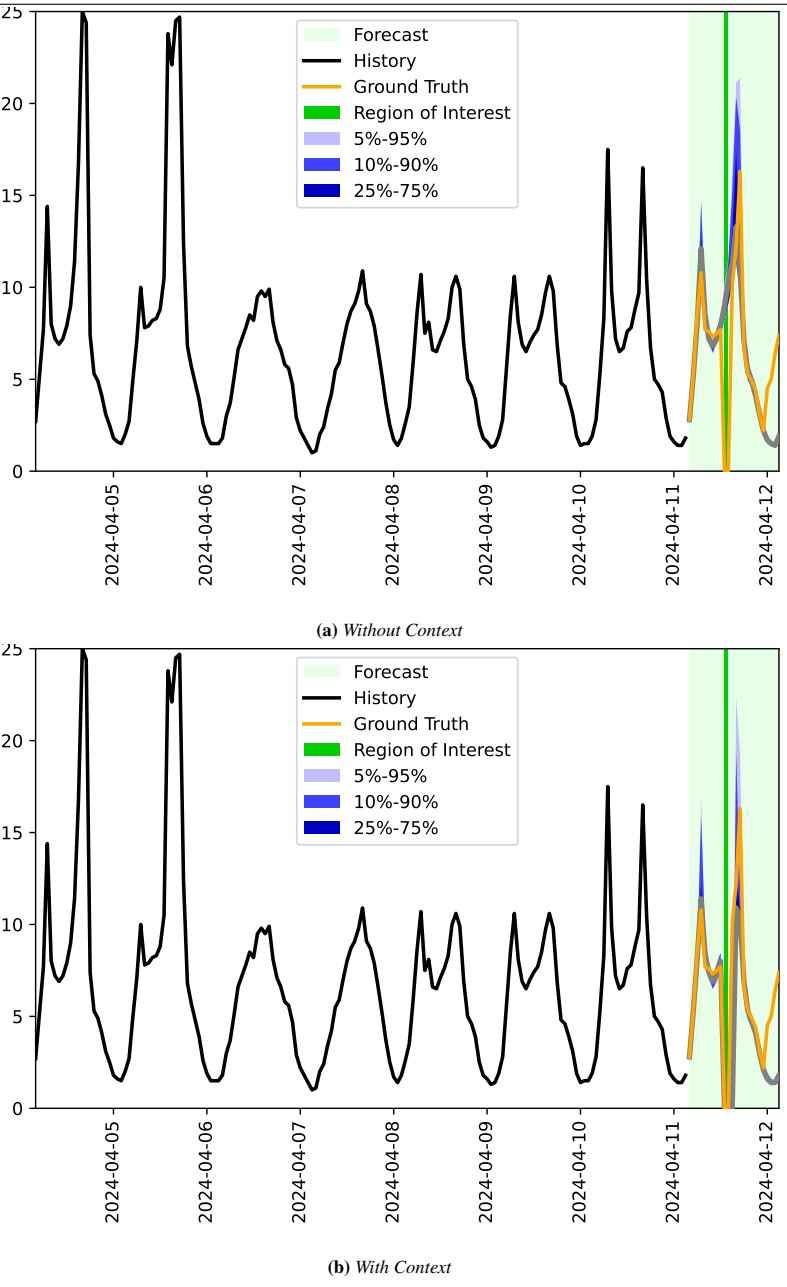

**(a)** *Without Context*

**(b)** *With Context*

*Figure 17.* Example of successful context-aware forecasting by DIRECT PROMPT with `Llama-3.1-405B-Instruct`

**Context:** " This series contains Diffuse Horizontal Irradiance for a location in Sinaloa, Mexico. The Diffuse Horizontal Irradiance is the total amount of sun energy (in Watts per squared meter) arriving indirectly on a horizontal surface, ignoring the direct sunlight. Even when there are no clouds to scatter the sun light, there will still be some Diffuse Horizontal Irradiance, since clouds are not the only cause of light scattering. When there are no clouds, the Diffuse Horizontal Irradiance is mostly a function of the position of the sun in the sky, with only small variations from factors such as water vapour and dust particles levels. If the cloud cover is light, the Diffuse Horizontal Irradiance will increase due to the increase scattering of sun light, but heavy cloud cover will decrease it due to some sun light no longer being able to reach the ground.

At the beginning of the series, the weather was cloudy.

At 2022-07-12 11:00:00, the weather became clear.

At 2022-07-12 19:00:00, the weather became cloudy.

At 2022-07-13 12:00:00, the weather became clear.

At 2022-07-13 13:00:00, the weather became cloudy.

At 2022-07-14 06:00:00, we expect that the weather will become clear.

At 2022-07-14 07:00:00, we expect that the weather will become cloudy.

At 2022-07-14 10:00:00, we expect that the weather will become clear.

At 2022-07-14 18:00:00, we expect that the weather will become cloudy. "

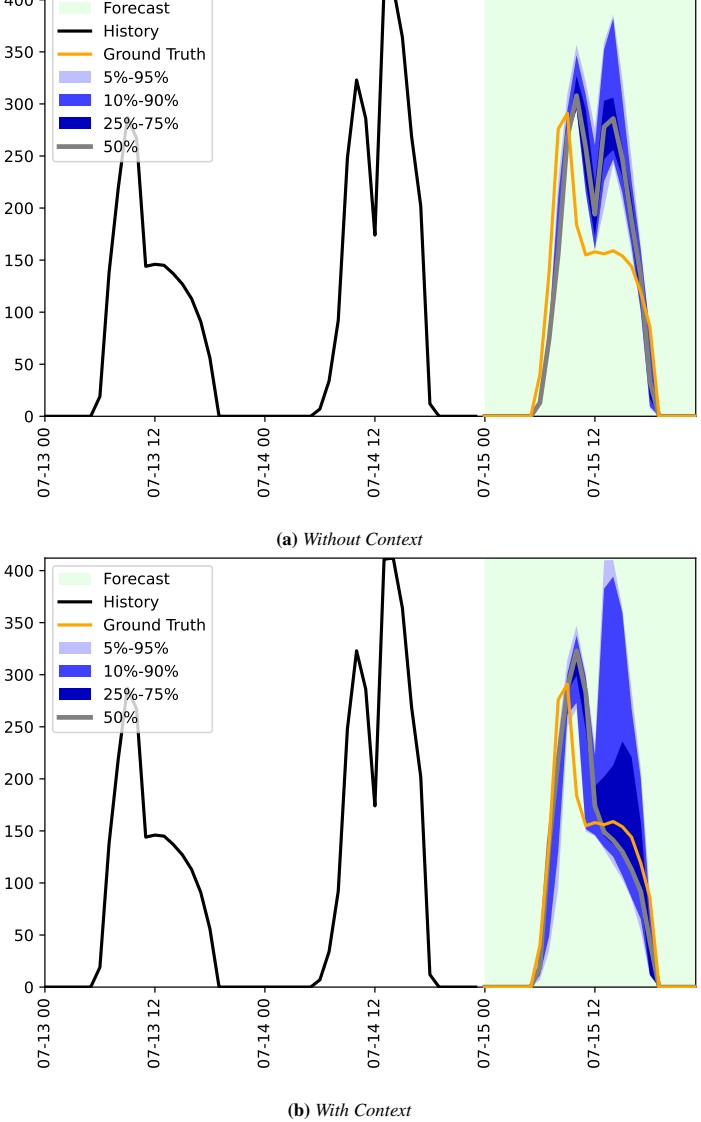

(a) *Without Context*

(b) *With Context*

*Figure 18.* Example of successful context-aware forecasting by DIRECT PROMPT with `Llama-3.1-405B-Instruct`

**Context:** " This is the number of cash withdrawals from an automated teller machine (ATM) in an arbitrary location in England.
Consider that the building which contains the ATM is closed from 1997-09-05 00:00:00, for 8 days. "

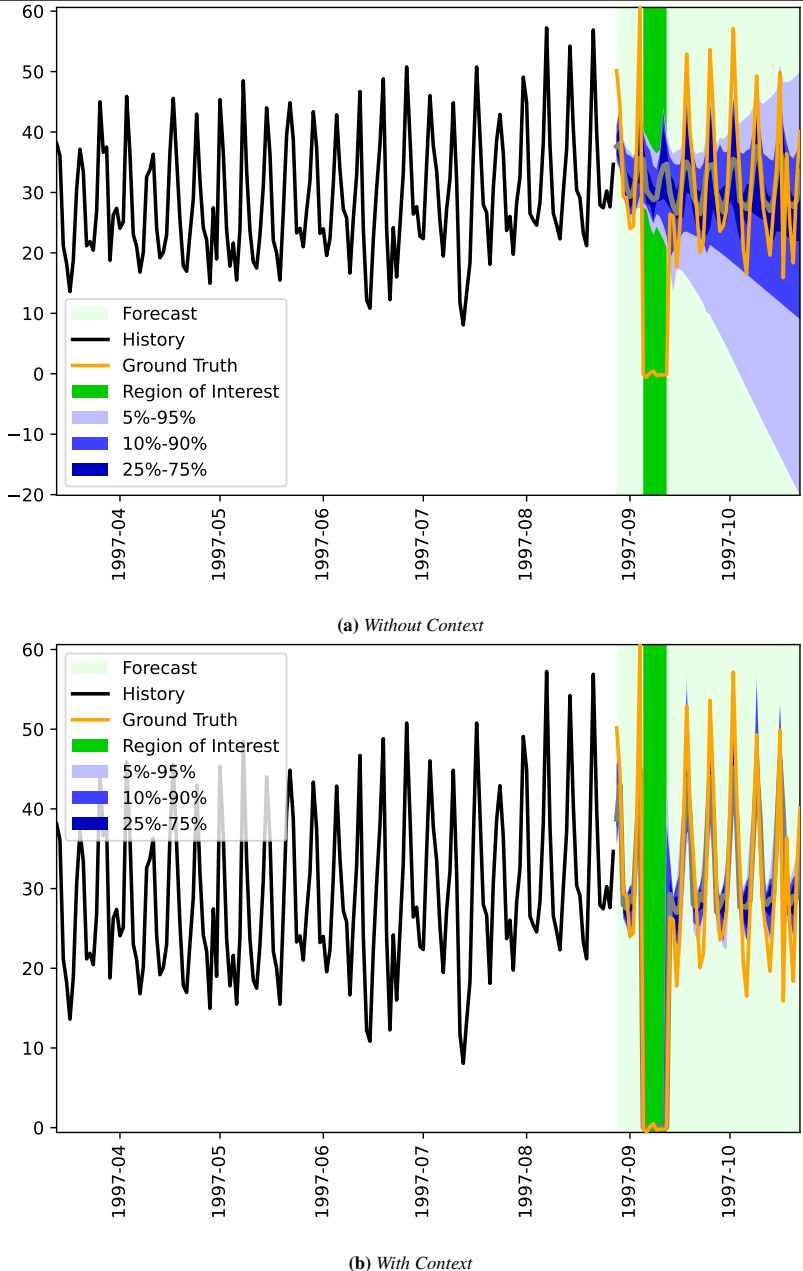

(a) *Without Context*

(b) *With Context*

*Figure 19.* Example of successful context-aware forecasts by DIRECT PROMPT with GPT-4o

**Context:** " The Montreal Fire Department is in charge of responding to various kind of public safety incidents. This is the number of field fire incidents responded to by Montreal firefighters in the Rivière-des-Prairies-Pointe-aux-Trembles borough. In other years, the yearly average number of incidents was 106 with the busiest month being June.

The Mayor is determined to completely eradicate this kind of incident. Fortunately, the city's public safety research group identified that field fires and trash fires tend to co-occur. When the amount of field fires increases, the amount of trash fires also tends to increase. The same holds when they decrease.

The Mayor has a plan: they will implement daily spraying of all piles of trash with water starting on 2022-06. "

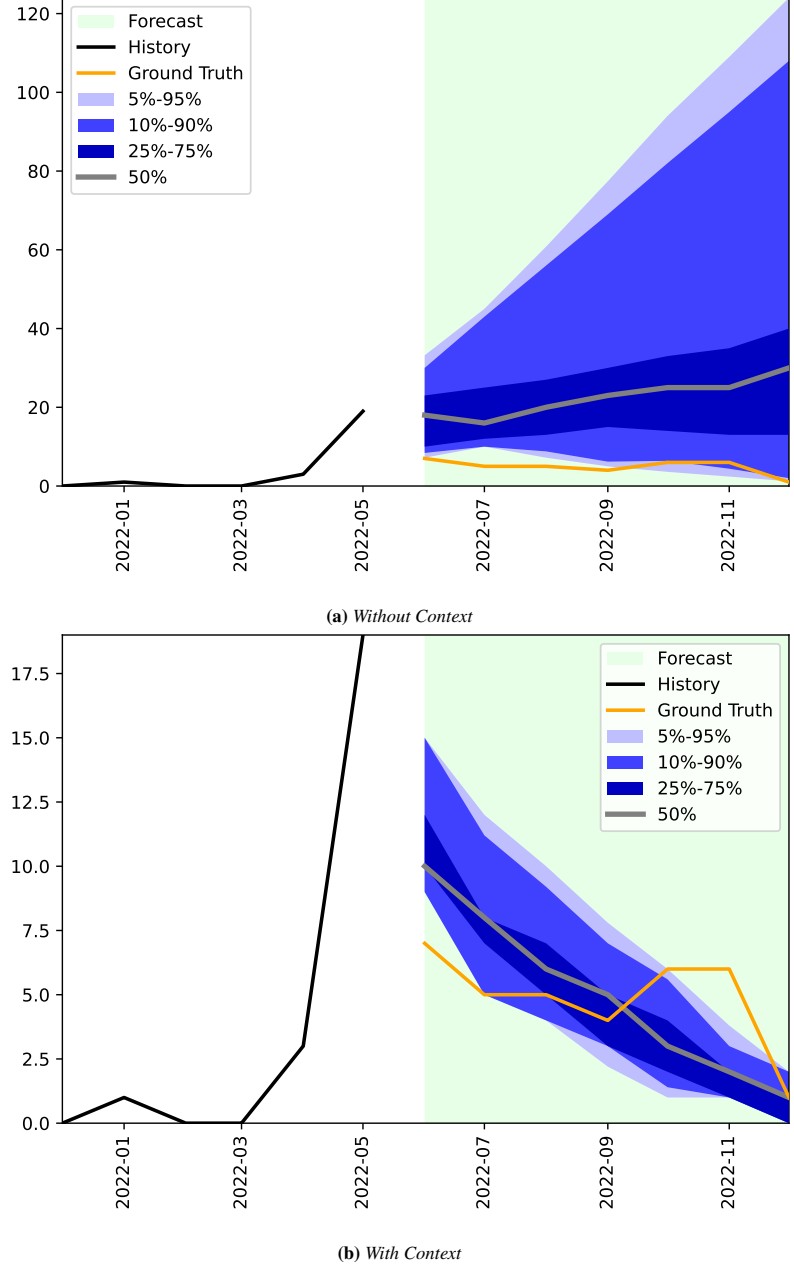

(a) *Without Context*

(b) *With Context*

*Figure 20.* Example of successful context-aware forecasts by DIRECT PROMPT with GPT-4o

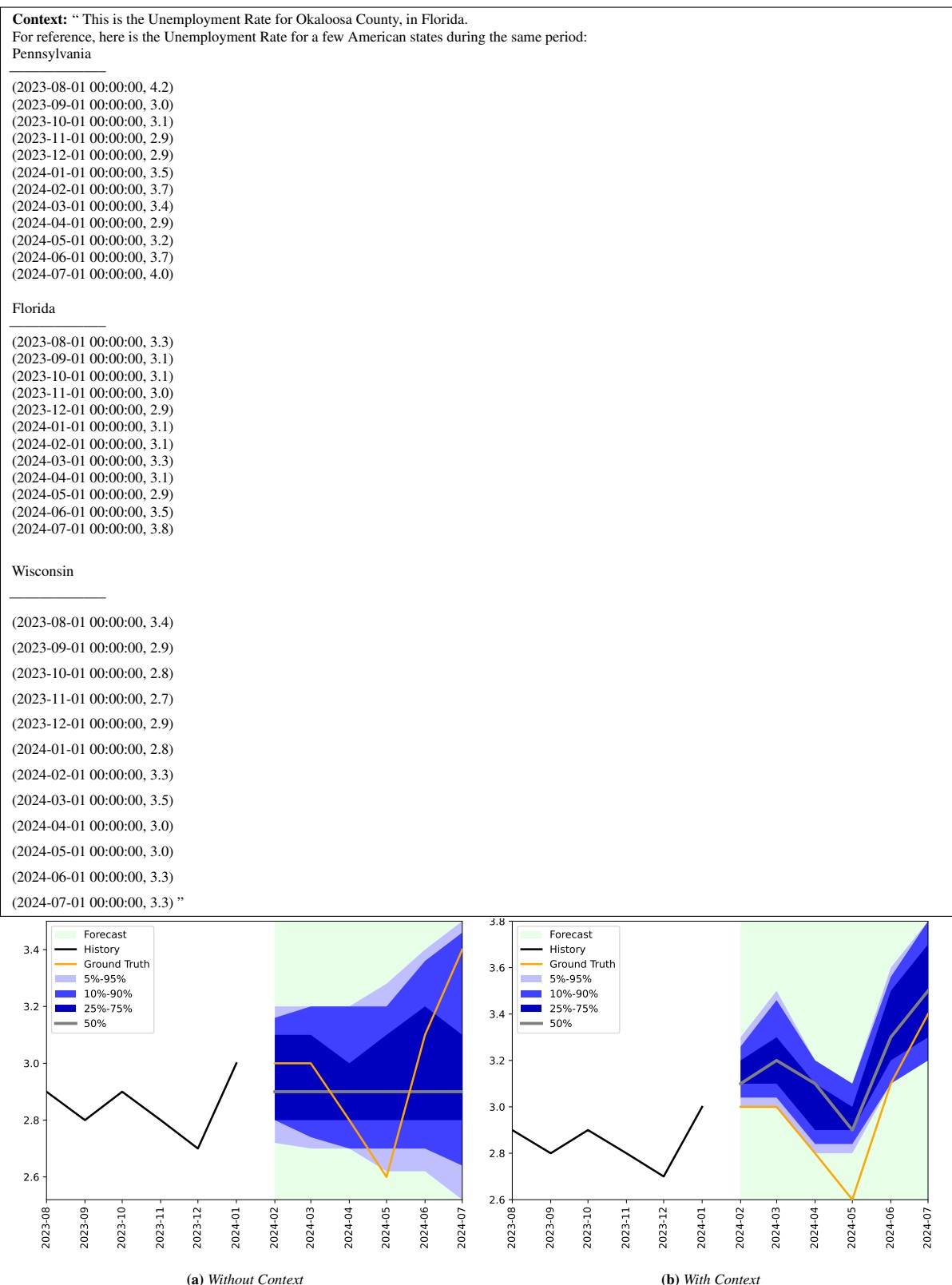

**Context:** " This is the Unemployment Rate for Okaloosa County, in Florida.
For reference, here is the Unemployment Rate for a few American states during the same period:
Pennsylvania
————————
(2023-08-01 00:00:00, 4.2)
(2023-09-01 00:00:00, 3.0)
(2023-10-01 00:00:00, 3.1)
(2023-11-01 00:00:00, 2.9)
(2023-12-01 00:00:00, 2.9)
(2024-01-01 00:00:00, 3.5)
(2024-02-01 00:00:00, 3.7)
(2024-03-01 00:00:00, 3.4)
(2024-04-01 00:00:00, 2.9)
(2024-05-01 00:00:00, 3.2)
(2024-06-01 00:00:00, 3.7)
(2024-07-01 00:00:00, 4.0)

Florida
————————
(2023-08-01 00:00:00, 3.3)
(2023-09-01 00:00:00, 3.1)
(2023-10-01 00:00:00, 3.1)
(2023-11-01 00:00:00, 3.0)
(2023-12-01 00:00:00, 2.9)
(2024-01-01 00:00:00, 3.1)
(2024-02-01 00:00:00, 3.1)
(2024-03-01 00:00:00, 3.3)
(2024-04-01 00:00:00, 3.1)
(2024-05-01 00:00:00, 2.9)
(2024-06-01 00:00:00, 3.5)
(2024-07-01 00:00:00, 3.8)

Wisconsin
————————
(2023-08-01 00:00:00, 3.4)
(2023-09-01 00:00:00, 2.9)
(2023-10-01 00:00:00, 2.8)
(2023-11-01 00:00:00, 2.7)
(2023-12-01 00:00:00, 2.9)
(2024-01-01 00:00:00, 2.8)
(2024-02-01 00:00:00, 3.3)
(2024-03-01 00:00:00, 3.5)
(2024-04-01 00:00:00, 3.0)
(2024-05-01 00:00:00, 3.0)
(2024-06-01 00:00:00, 3.3)
(2024-07-01 00:00:00, 3.3) "

**(a)** *Without Context*          **(b)** *With Context*

*Figure 21.* Example of successful context-aware forecasts by `LLMP` with `Mixtral-8x7B-Instruct`

**Context:** " Suppose that in the forecast, the values are bounded below by 0.80. "

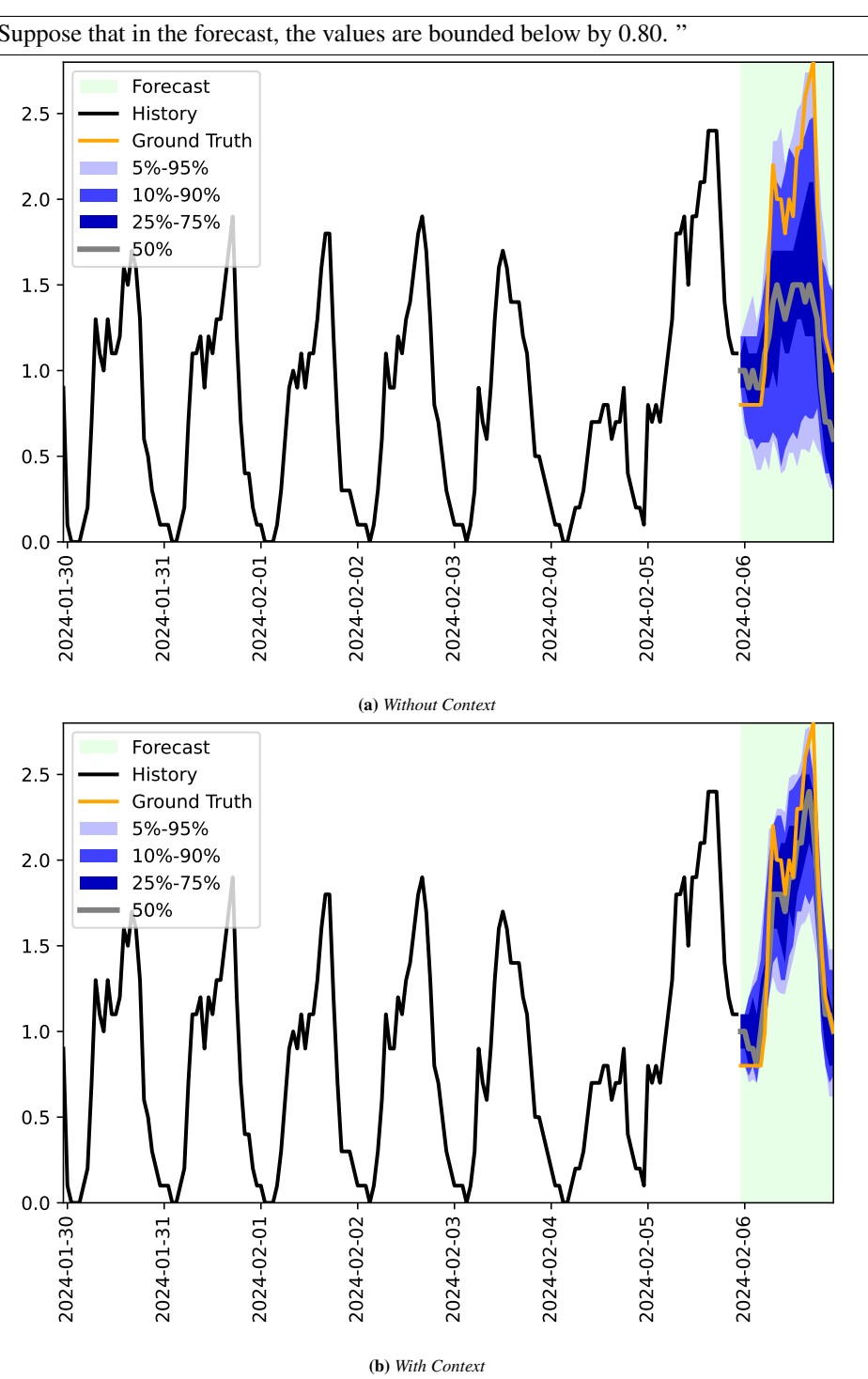

**(a)** *Without Context*

**(b)** *With Context*

*Figure 22.* Example of successful context-aware forecasts by LLMP with `Mixtral-8x7B-Instruct`

**Context:** " This series contains the amount of sunlight (in Watts per squared meter) arriving on a horizontal surface, for a location in Alaska, United States. "

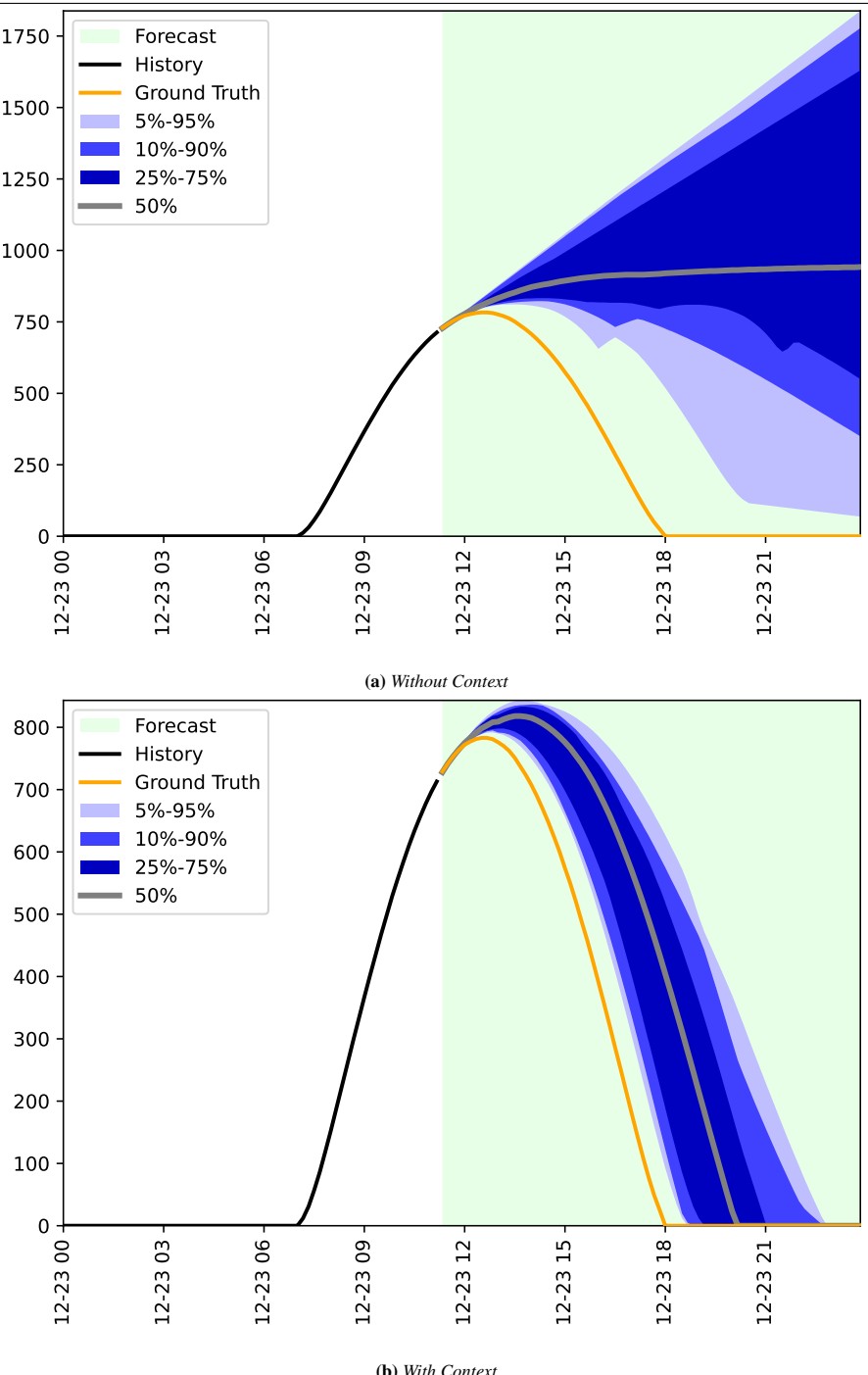

(a) *Without Context*

(b) *With Context*

*Figure 23.* Example of successful context-aware forecasts by LLMP with `Llama-3-70B`

**Context:** " The Montreal Fire Department is in charge of responding to various kind of public safety incidents. This series contains the number of field fire incidents responded to by the Montreal Fire Department in the Rosemont-La Petite-Patrie borough. On average, they respond to 58 incidents per year and the month with the most incidents was June.

The Mayor is determined to completely eradicate this kind of incident. Fortunately, the city's public safety research group, a team of highly qualified experts, identified that field fires and gas leaks tend to co-occur. When the amount of field fires increases, the amount of gas leaks also tends to increase. The same holds when they decrease.

The Mayor has a plan: they will implement a strict prohibition of using any form of combustible gas in the city starting on 2023-06. In a recent interview, they claimed, "This is a bulletproof plan, and I am certain it will immediately put an end to field fires." "

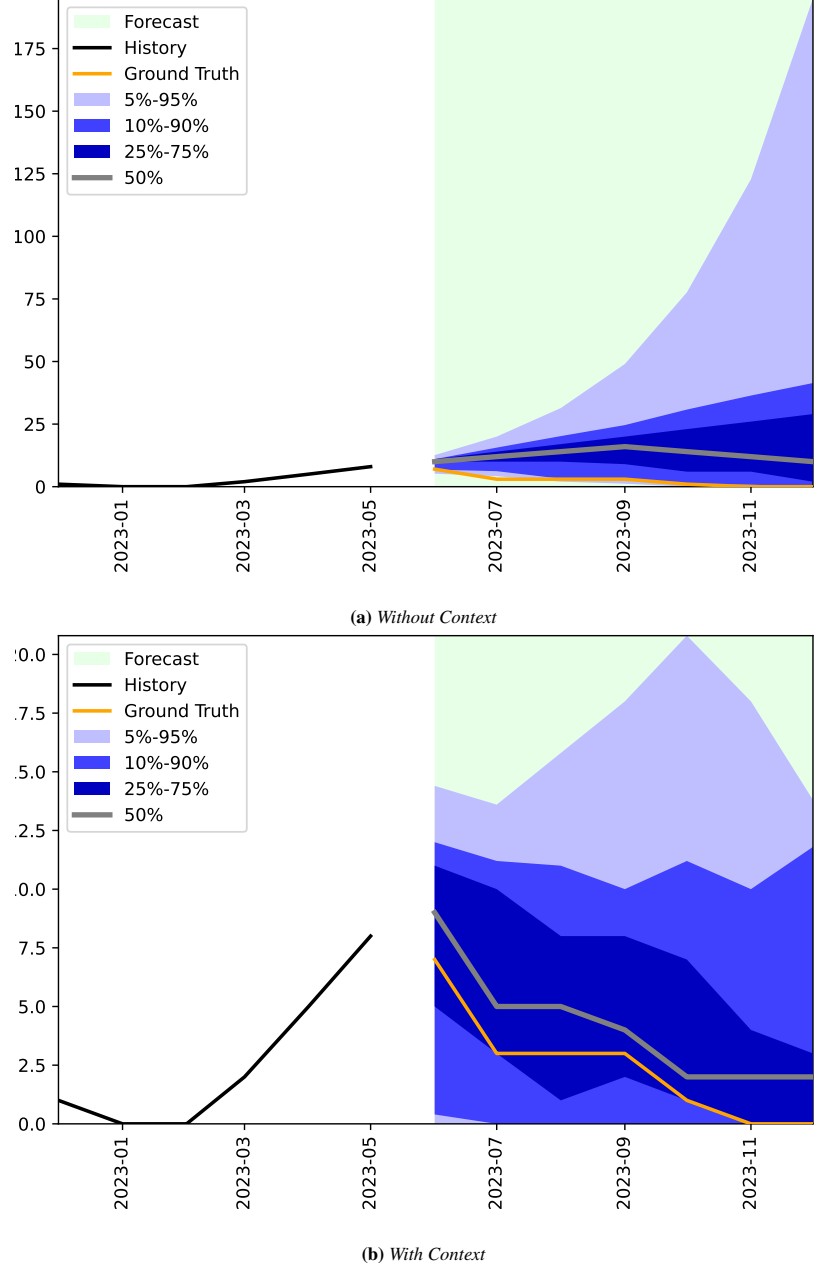

**(a)** *Without Context*

**(b)** *With Context*

*Figure 24.* Example of successful context-aware forecasts by LLMP with `Llama-3-70B`

## C.12. Visualizations of significant failures

**Context:** " Given are variables $X\_0$ and $X\_1$, where $X\_0$ is a covariate and $X\_1$ is the variable to forecast. Variables are generated from a linear Structural Vector Autoregressive (SVAR) model with additive gauss noise and a noise scale of 1.487e-03, with lag = 3.

The task is to forecast the value of the variable $X\_1$ at time t, given the values of the covariate $X\_0$ and the variable $X\_1$ itself at times t-1, ... t-3. For the first 128 days, the covariate $X\_0$ takes a value of 8 from 2024-02-21 to 2024-03-11, 12 from 2024-03-12 to 2024-05-06, 12 from 2024-05-07 to 2024-06-27. For the next 32 days, the covariate $X\_0$ takes a value of 30 from 2024-06-28 to 2024-07-13, 60 from 2024-07-14 to 2024-07-14, 60 from 2024-07-15 to 2024-07-29. Each day can be treated as a timestep for the forecasting task. The causal parents affect the child variables at different lags.

The causal parents for each variable is given below:

No parents for $X\_0$ at any lag.

Parents for $X\_1$ at lag 1: ['X\_0', 'X\_1'] affect the forecast variable as 0.527 * $X\_0$ + -0.895 * $X\_1$.

Parents for $X\_1$ at lag 2: ['X\_0', 'X\_1'] affect the forecast variable as 1.380 * $X\_0$ + -0.758 * $X\_1$.

Parents for $X\_1$ at lag 3: ['X\_0', 'X\_1'] affect the forecast variable as -0.661 * $X\_0$ + -0.793 * $X\_1$.
"

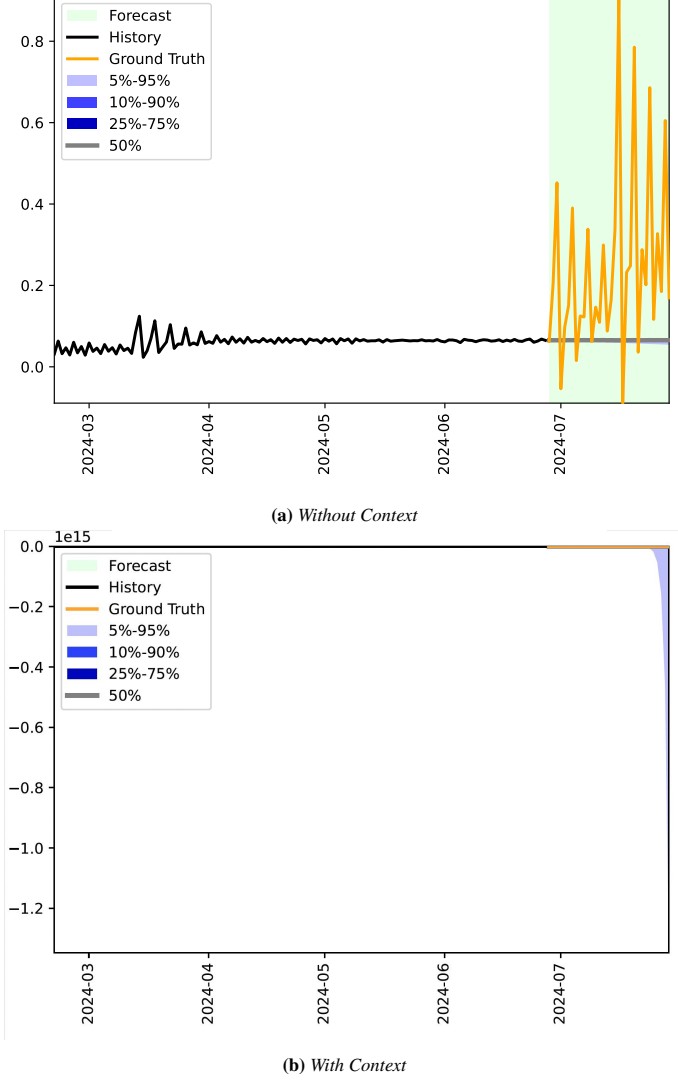

**(a)** *Without Context*

**(b)** *With Context*

*Figure 25.* Example to show a significant failure case of DIRECT PROMPT with GPT-4o where its performance worsens with context

**Context:** " This series contains the road occupancy rates on a freeway in the San Francisco Bay area. The days for which the forecast is required are Thursday 2024-07-04, Friday 2024-07-05, Saturday 2024-07-06. Note that 2024-07-04 is a holiday due to Independence Day. Note that traffic on this freeway typically reduces on holidays. "

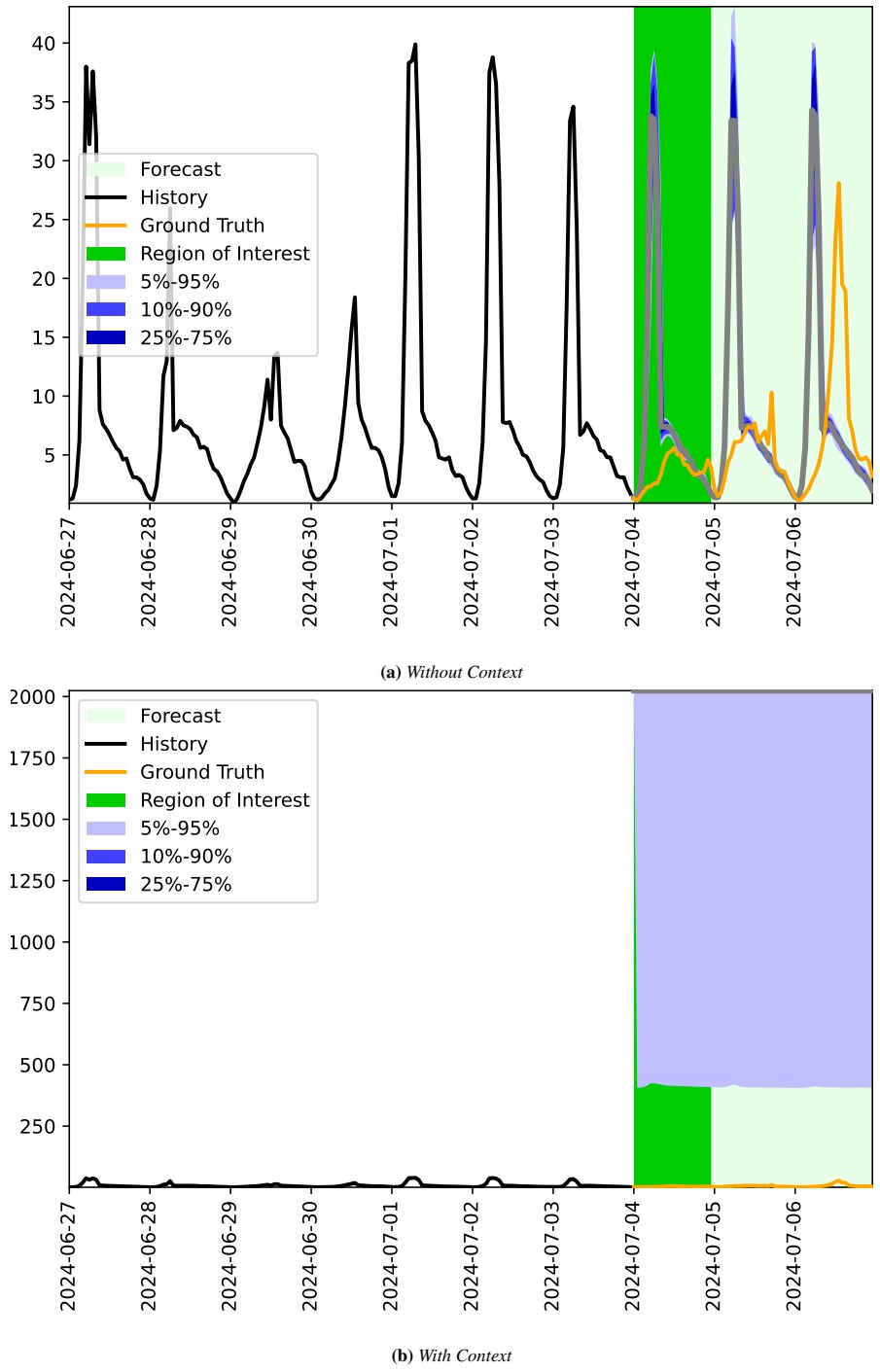

**(a)** *Without Context*

**(b)** *With Context*

*Figure 26.* Example to show a significant failure case of `LLMP` with `Llama-3-70B` where its performance worsens with context

**Context:** " This series represents the occupancy rate (%) captured by a highway sensor. The sensor had a calibration problem starting from 2024-04-20 13:00:00 which resulted in an additive trend in the series that increases by 0.0072 at every hour. At timestep 2024-04-24 13:00:00, the sensor was repaired and this additive trend will disappear. "

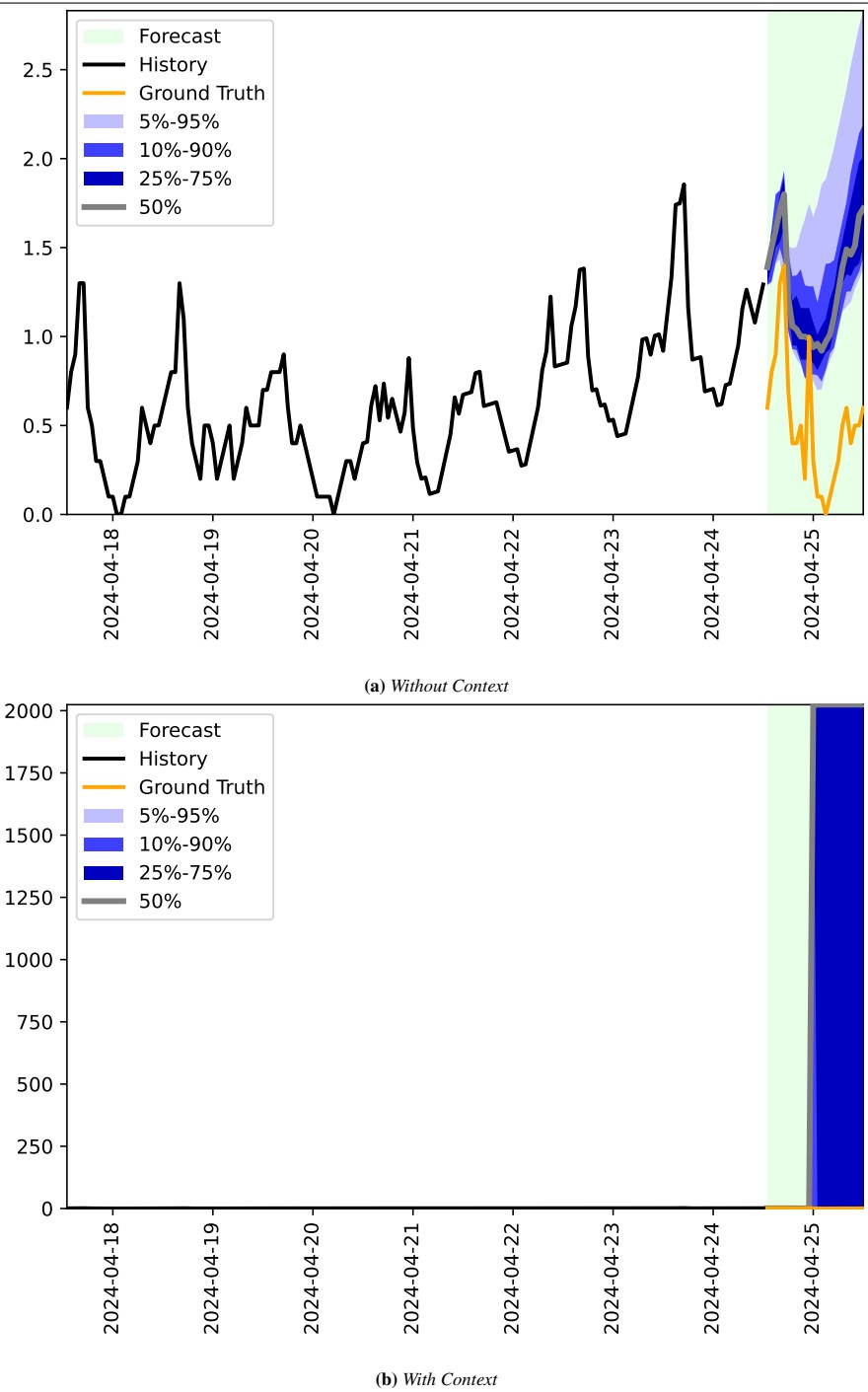

**(a)** *Without Context*

**(b)** *With Context*

*Figure 27.* Example to show a significant failure case of `LLMP` with `Llama-3-70B` where its performance worsens with context

**Context:** " The Montreal Fire Department is in charge of responding to various kind of public safety incidents. This series contains the number of field fire incidents responded to by the Montreal Fire Department in the L'Île-Bizard-Sainte-Geneviève borough. On average, they respond to 19 incidents per year with the busiest month being June.

The Mayor is determined to completely eradicate this kind of incident. Fortunately, the city's public safety research group, a team of highly qualified experts, identified that field fires and trash fires tend to co-occur. When the amount of field fires increases, the amount of trash fires also tends to increase. The same holds when they decrease.

The Mayor has a plan: they will implement daily spraying of all piles of trash with fire retardant foam starting on 2023-06. In a recent interview, they claimed, "This is a bulletproof plan, and I am certain it will immediately put an end to field fires." "

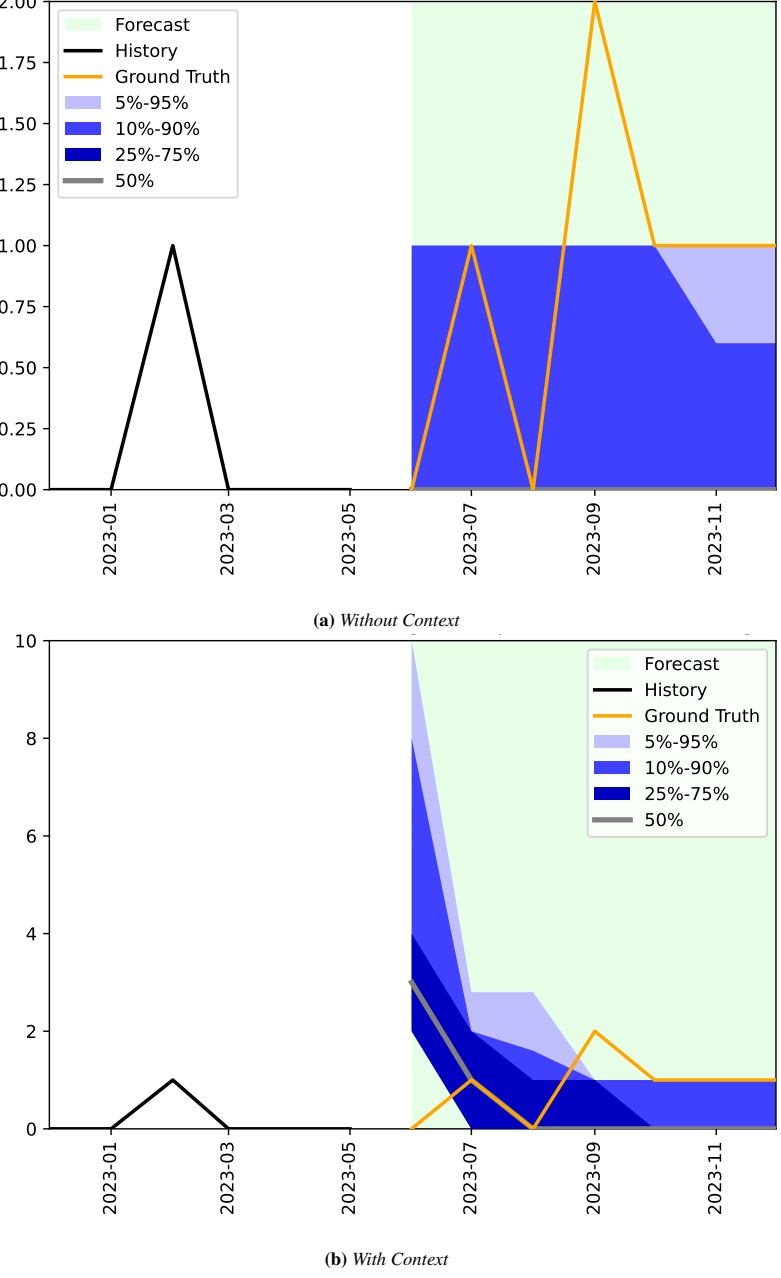

**(a)** *Without Context*

**(b)** *With Context*

*Figure 28.* Example to show a significant failure case of DIRECT PROMPT with `Llama-3-8B-Instruct` where it misinterprets the context

# D. Implementation Details of Models

## D.1. DIRECT PROMPT

### D.1.1. METHOD

For DIRECT PROMPT, we propose to use a simple prompt template that we describe below, where ((**context**)) is replaced with the context of the respective task, ((**history**)) is replaced with the historical values in the given format, and ((**pred_time**)) is replaced with the prediction timesteps. The prompted model is expected to output predictions in the given template style (i.e. within the given forecast tags, in the given format) for all prediction timesteps in the prompt. Notably, unlike LLMP which consists of predicting the single next digit in a loop, Direct Prompt expects models to forecast in a single pass in a highly structured format, which requires models to understand and adhere to the template.

```
"
I have a time series forecasting task for you.

Here is some context about the task. Make sure to factor in any background knowledge,
satisfy any constraints, and respect any scenarios.
<context>
((context))
</context>

Here is a historical time series in (timestamp, value) format:
<history>
((history))
</history>

Now please predict the value at the following timestamps: ((pred_time)).

Return the forecast in (timestamp, value) format in between <forecast> and </forecast> tags.
Do not include any other information (e.g., comments) in the forecast.

Example:
<history>
(t1, v1)
(t2, v2)
(t3, v3)
</history>
<forecast>
(t4, v4)
(t5, v5)
</forecast>
"
```

To constrain the output of the model to follow the specific structure, we use constrained decoding using the lm-format-enforcer tool (`https://github.com/noamgat/lm-format-enforcer`) and a regular expression that only allows models to output the values corresponding to the prediction timestamps. Without constrained decoding, we observe that models often produce samples that fail to adhere to the structure and are therefore rejected. Specifically, larger models (`Llama-3.1-405B-Instruct`, `GPT-4o` and `GPT-4o-mini`) can produce 25 valid forecasts with 1 to 3 trials. However with the smaller models (such as `Llama-3-70B-Instruct`, `Llama-3-8B-Instruct` and `Mixtral-8x7B-Instruct`), up to 10 trials can be required to obtain 25 valid forecasts.

Further, we found that without an explicit "Do not include any other information (e.g., comments) in the forecast.", models often included unwanted information along with the forecasts.

**Instruction-tuned models are more amenable for DIRECT PROMPT** DIRECT PROMPT requires forecasts to be produced in a specific structure. To generate structured outputs, models need to be steerable (Dubey et al., 2024), a capability that is typically elicited from base models with post-training methods such as instruction tuning (Wei et al., 2021). We observe this in our evaluations as we find that several base models, including `Llama-3-8B`, `Llama-3-70B`, `Mixtral-8x7B`, and even the biggest base model we tried, `Llama-3.1-405B`, are incapable of generating outputs adhering to the structure required for DIRECT PROMPT, despite increasing the number of retries to as high as 50 retries. With DIRECT PROMPT, these models often output irrelevant information, sometimes completing solely the context as a text completion task, and in other cases regurgitating forecasting datasets that they have memorized.

**Extensions of** DIRECT PROMPT     While very simple, such prompt templates can be powerful tools to understand how LLMs perform context-aided forecasting: as the prompt gives control over the structure and content of the output (particularly for instruction-tuned models), one may construct other, more involved template structures in the prompt. For instance, a prompt template could ask LLMs to explain the reasoning behind their (context-aided) forecasts, and more. We leave it to future work to understand how such prompt-based techniques can lead to more detailed evaluations and give us better insights into what the models are capable of.

### D.1.2. IMPLEMENTATION DETAILS

We used a single H100 GPU to run the DIRECT PROMPT approach for `Llama-3-8B-Instruct`, and 2 H100 GPUs for `Qwen-2.5-`{`0.5B-Instruct, 1.5-Instruct, 7B-Instruct`}, `Llama-3-70B-Instruct` and `Mixtral-8x7B-Instruct`. We queried `Llama-3.1-405b-Instruct` from an externally-hosted server running on 8 H100s. We use the OpenAI API to perform inference on the proprietary `GPT-4o` and `GPT-4o-mini` models. We provide the cost incurred in the inference of these models with the DIRECT PROMPT method in Appendix C.7.

### D.1.3. EXAMPLE PROMPT

A prompt used in an example task from the benchmark is given below.

```
 "
I have a time series forecasting task for you.

Here is some context about the task. Make sure to factor in any background knowledge,satisfy any constraints,
    and respect any scenarios.
<context>
Background: This is hourly traffic data.
Scenario: Suppose that there is an accident on the road and there is 40.0% of the usual traffic from 2024-04-24
    17:00:00 for 6 hours.
</context>

Here is a historical time series in (timestamp, value) format:
<history>
(2024-04-23 00:00:00, 0.1)(2024-04-23 01:00:00, 0)(2024-04-23 02:00:00, 0)(2024-04-23 03:00:00, 0)(2024-04-23
    04:00:00, 0.1)(2024-04-23 05:00:00, 0.2)(2024-04-23 06:00:00, 0.3)(2024-04-23 07:00:00, 0.5)(2024-04-23
    08:00:00, 0.5)(2024-04-23 09:00:00, 0.4)(2024-04-23 10:00:00, 0.5)(2024-04-23 11:00:00, 0.5)(2024-04-23
    12:00:00, 0.4)(2024-04-23 13:00:00, 0.6)(2024-04-23 14:00:00, 0.8)(2024-04-23 15:00:00, 1.2)(2024-04-23
    16:00:00, 1.2)(2024-04-23 17:00:00, 1.3)(2024-04-23 18:00:00, 0.6)(2024-04-23 19:00:00, 0.3)(2024-04-23
    20:00:00, 0.3)(2024-04-23 21:00:00, 0.3)(2024-04-23 22:00:00, 0.1)(2024-04-23 23:00:00, 0.1)(2024-04-24
    00:00:00, 0.1)(2024-04-24 01:00:00, 0)(2024-04-24 02:00:00, 0)(2024-04-24 03:00:00, 0.1)(2024-04-24
    04:00:00, 0.1)(2024-04-24 05:00:00, 0.2)(2024-04-24 06:00:00, 0.3)(2024-04-24 07:00:00, 0.5)(2024-04-24
    08:00:00, 0.6)(2024-04-24 09:00:00, 0.5)(2024-04-24 10:00:00, 0.4)(2024-04-24 11:00:00, 0.5)(2024-04-24
    12:00:00, 0.6)
</history>

Now please predict the value at the following timestamps: ['2024-04-24 13:00:00' '2024-04-24 14:00:00'
    '2024-04-24 15:00:00' '2024-04-24 16:00:00' '2024-04-24 17:00:00' '2024-04-24 18:00:00' '2024-04-24
    19:00:00' '2024-04-24 20:00:00' '2024-04-24 21:00:00' '2024-04-24 22:00:00' '2024-04-24 23:00:00'
    '2024-04-25 00:00:00' '2024-04-25 01:00:00' '2024-04-25 02:00:00' '2024-04-25 03:00:00' '2024-04-25
    04:00:00' '2024-04-25 05:00:00' '2024-04-25 06:00:00' '2024-04-25 07:00:00' '2024-04-25 08:00:00'
    '2024-04-25 09:00:00' '2024-04-25 10:00:00' '2024-04-25 11:00:00' '2024-04-25 12:00:00'].

Return the forecast in (timestamp, value) format in between <forecast> and </forecast> tags.Do not include any
    other information (e.g., comments) in the forecast.

Example:
<history>
(t1, v1)
(t2, v2)
(t3, v3)
</history>
<forecast>
(t4, v4)
(t5, v5)
</forecast>
 "
```

## D.2. LLMP

### D.2.1. METHOD

In this section we outline LLM-processes (LLMP; Requeima et al. (2024)), one of the prompt-based baselines evaluated in Sec. 5.3. Prompts are constructed by first providing textual information followed by the numerical history. The context may include background knowledge, a scenario description and task constraints, replaced by ((**background**)), ((**scenario**)) and ((**constraints**)), respectively, in the prompt template below. The numerical history (((**history**))) is provided by converting the numerical data to text where values are separated by commas (,) and tuples by newline characters (\n). The LLM then outputs the continuation of the string prompt, forecasing the the value for the next time index (((**next index**))). This forecast and the next time index is appended to the prompt allowing the LLM to autoregressively complete the entire forecast. Numerical samples are rejected if they do not adhere to a decimal representation format. See Requeima et al. (2024)) for full

details.

The following is the prompt template used to construct prompts for the LLMP baseline:

```
"
Forecast the future values of this time series, while considering the following background knowledge, scenario,
    and constraints.

Background knowledge:
((background))

Scenario:
((scenario))

Constraints:
((constraints))

((history))
((next index))
"
```

A prompt used in an example task from the benchmark is given below:

```
"
Forecast the future values of this time series, while considering the following background knowledge, scenario,
    and constraints.

Background knowledge:
This is hourly traffic data.

Scenario:
Suppose that there is an accident on the road and there is 40.0% of the usual traffic from 2024-04-24 17:00:00
    for 6 hours.

Constraints:

2024-04-23 00:00:00,0.1\n2024-04-23 01:00:00,0\n2024-04-23 02:00:00,0\n2024-04-23 03:00:00,0\n2024-04-23
    04:00:00,0.1\n2024-04-23 05:00:00,0.2\n2024-04-23 06:00:00,0.3\n2024-04-23 07:00:00,0.5\n2024-04-23
    08:00:00,0.5\n2024-04-23 09:00:00,0.4\n2024-04-23 10:00:00,0.5\n2024-04-23 11:00:00,0.5\n2024-04-23
    12:00:00,0.4\n2024-04-23 13:00:00,0.6\n2024-04-23 14:00:00,0.8\n2024-04-23 15:00:00,1.2\n2024-04-23
    16:00:00,1.2\n2024-04-23 17:00:00,1.3\n2024-04-23 18:00:00,0.6\n2024-04-23 19:00:00,0.3\n2024-04-23
    20:00:00,0.3\n2024-04-23 21:00:00,0.3\n2024-04-23 22:00:00,0.1\n2024-04-23 23:00:00,0.1\n2024-04-24
    00:00:00,0.1\n2024-04-24 01:00:00,0\n2024-04-24 02:00:00,0\n2024-04-24 03:00:00,0.1\n2024-04-24 04:00:00,0.1
    \n2024-04-24 05:00:00,0.2\n2024-04-24 06:00:00,0.3\n2024-04-24 07:00:00,0.5\n2024-04-24 08:00:00,0.6\n2024
    -04-24 09:00:00,0.5\n2024-04-24 10:00:00,0.4\n2024-04-24 11:00:00,0.5\n2024-04-24 12:00:00,0.6\n2024-04-24
    13:00:00,
"
```

### D.2.2. IMPLEMENTATION DETAILS

We used a single H100 GPU to run the LLMP approach for the following models: `Llama-3-8B`, and `Llama-3-8B-Instruct`. We used 2 H100 GPUs for the `Qwen-2.5` family of models, `Mixtral-8x7B`, and `Mixtral-8x7B-Instruct`, and used used 8 H100 GPUs for the following models: `Llama-3-70B`, and `Llama-3-70B-Instruct`.

Since the code of LLMP (https://github.com/requeima/llm_processes/) only supports using open-source models (such as those available in HuggingFace) and requires loading the weights into memory, it does not support experimenting with the `GPT-4o` and `GPT-4o-mini` models. Further, due to the memory requirements of LLMP, we were unable to experiment with the `Llama-3.1-405B` and `Llama-3.1-405B-Instruct` models that required more than 24 H100 GPUs in parallel to process a single instance from the benchmark, which exceeded our available resources.

### D.3. `ChatTime`

We evaluate the released ChatTime-Base (`https://huggingface.co/ChengsenWang/ChatTime-1-7B-Base`) and ChatTime-Chat (`https://huggingface.co/ChengsenWang/ChatTime-1-7B-Chat`) models zero-shot, as per the instructions in the authors' GitHub repository (`https://github.com/ForestsKing/ChatTime`).

### D.4. `UniTime` and `Time-LLM`

For multimodal models, we jointly train `UniTime` (Liu et al., 2024c) on its ensemble of datasets: ETTm1, ETTm2, ETTh1, ETTh2, Electricity, Weather, Exchange and Illness.

We also evaluate `Time-LLM` (Jin et al., 2024), another multimodal model built on top of the Llama architecture. We train `Time-LLM` on ETTh1 according to the authors' suggested specifications, and we compare the performance of both models with and without context.

**`UniTime`:** We train `UniTime` (Liu et al., 2024c) with their codebase (`https://github.com/liuxu77/UniTime`) using a single seed on one AMD Instinct MI200 GPU for approximately 14 hours. It features a lightweight transformer with maximum context length of 210 and a pre-trained GPT2 language model as backbone, of which only the first half of the transformer layers are used. The time series baseline employs non-overlapping patch embeddings generated with a kernel size and stride of 16, and a maximum input sequence length of 96. When the total tokenized length exceeds the architecture's capacity, we truncate the context.

Unlike `Time-LLM`, `UniTime` is jointly trained on all datasets simultaneously. Batches were generated by first choosing a dataset uniformly at random then returning a batch from the associated data loader. To account for domain convergence speed imbalance, a mask rate of 0.5 is used and the training batch size is varied according to the dataset (details in the data config directory of the `UniTime` GitHub repository). Training was conducted for 10 epochs of the mixed dataset, with cosine decay from an initial learning rate of 1e-4 to a minimum of 1e-6 over a maximum period of 20 epochs. The results of our training on the original datasets are given in Tab. 10.

Finally, in order to accelerate training, we added BF16 automatic mixed precision training and gradient accumulation to the original training procedure.

**`Time-LLM`:** We train `Time-LLM` (Jin et al., 2024) with their codebase (`https://github.com/KimMeen/Time-LLM`) on the ETTh1 dataset (Zhou et al., 2021) with a prediction length of 96. We train using a single seed on four AMD Instinct MI200 GPUs, with an average training time per run of approximately 13 hours. Training was conducted using a batch size of 8 per device and 4 gradient accumulation steps, along with a 1Cycle learning rate schedule with a maximum learning rate of 1e-3. In addition, runs were accelerated using DeepSpeed Stage 2 and BF16 automatic mixed precision.

Training was conducted over a maximum of 50 epochs with early stopping, and a time-based split of 70% for training, 10% for validation, and 20% for testing, where the most recent windows were reserved for the test set. All runs were trained with

*Table 10.* Evaluation results for `UniTime` on their test splits. Results are comparable to the original paper, although MSE on Illness is approximately 20% higher for prediction lengths 36,48,60.

| Dataset | Mean Squared Error (MSE) | | | |
|---|---|---|---|---|
| Prediction Length | 96 | 192 | 336 | 720 |
| ETTh1 | 0.395 | 0.435 | 0.469 | 0.468 |
| ETTh2 | 0.291 | 0.368 | 0.413 | 0.422 |
| ETTm1 | 0.336 | 0.377 | 0.409 | 0.465 |
| ETTm2 | 0.181 | 0.248 | 0.315 | 0.417 |
| Exchange | 0.090 | 0.180 | 0.322 | 0.862 |
| Weather | 0.179 | 0.224 | 0.278 | 0.354 |
| Electricity | 0.198 | 0.202 | 0.217 | 0.257 |
| | 24 | 36 | 48 | 60 |
| Illness | 2.284 | 2.515 | 2.572 | 2.455 |

an input sequence length of 512, with overlapping patch embeddings generated with a kernel size of 16 and a stride of 8. The results on the ETTh1 test set are given in Tab. 11.

When evaluating on CiK tasks which do not conform to `Time-LLM`'s requirements, we make the following modifications to the method:

- For short history tasks where the history length $|\mathbf{X_H}|$ is less than 5, we change the `topk` operator's $k$ value from 5 to $|\mathbf{X_H}|$ in the `calculate_lags()` function.

- For tasks where the length of the prediction window $|\mathbf{X_F}|$ exceeds the trained projection head's output dimension (in our case, 96), we repeat the last predicted value $|\mathbf{X_F}| - 96$ times. This occurs for very few tasks (3 tasks) with prediction windows of 97 or 98 steps depending on the sampled instance, which we assume leads to a negligible impact on evaluated results.

*Table 11.* ETTh1 test set results for Time-LLM trained on ETTh1.

| Time-LLM | MSE | MAE |
|---|---|---|
| ETTh1-pl96 | 0.3846123 | 0.4149854 |

**Why Do `Time-LLM` and `UniTime` Not Benefit (More) From Context?**  Looking at table Appendix C.1, we see that context actually harms the performance of `Time-LLM`'s forecasts. Two possible reasons for this are: 1) `Time-LLM`'s adaptation procedure is unlikely to retain the backbone LLM's language-processing capabilities, and 2) `Time-LLM`'s single-dataset training procedure is unlikely to generalize to unseen time series patterns. Part of `Time-LLM`'s model adaptation involves training linear layers at the input and output of the language model. Although the backbone LLM remains frozen, these linear layers must be trained, and `Time-LLM` opts for a highly structured prompting format which involves domain knowledge, task instructions and input statistics. Since the training data for the linear layers consists of output representations based on these highly structured prompts, it is not evident that the resulting architecture will generalize to more diverse contextual descriptions such as those found in CiK. Furthermore, although we have not conducted a formal analysis of the diversity of the ETTh1 dataset, it is not a priori obvious that such a dataset would have a sufficient diversity of patterns to train a time series foundation model.

Interestingly, `UniTime`'s performance does benefit from context for some tasks (see Figure 29). However, the aggregate RCRPS and rank of `UniTime` with respect to other models indicate that it still struggles to produce forecasts competitive with even quantitative forecasting methods.

### D.5. `Lag-Llama`

We use the publicly available implementation of `Lag-Llama` (Rasul et al., 2023) located at `https://github.com/time-series-foundation-models/`, and its associated pre-trained weights. The model inference was done on a single H100 GPU.

### D.6. `Chronos`

We use the publicly available implementation of `Chronos` (Ansari et al., 2024) located at `https://github.com/amazon-science/chronos-forecasting`. We evaluated (see Appendix C.1) our tasks on all 5 available models: chronos-tiny, chronos-mini, chronos-small, chronos-base and chronos-large, and reported the results of the best performing model, chronos-large in Tab. 1. The model inference was done on a single H100 GPU.

### D.7. `Moirai`

We use the publicly available implementation of `Moirai` (Woo et al., 2024) located at `https://github.com/SalesforceAIResearch/uni2ts`. We evaluated (see Appendix C.1) our tasks on the 3 following models: moirai-1.0-R-small (located at `https://huggingface.co/Salesforce/moirai-1.0-R-small`), moirai-1.0-R-base (located at `https://huggingface.co/Salesforce/moirai-1.0-R-base`) and moirai-1.0-R-large (located at `https://huggingface.co/Salesforce/moirai-1.0-R-large`) and reported the results of the best performing model, moirai-1.0-R-large in Tab. 1. The model inference was done on a single H100 GPU.

**Context:** "Suppose that in the forecast, the values are bounded above by 6.29."

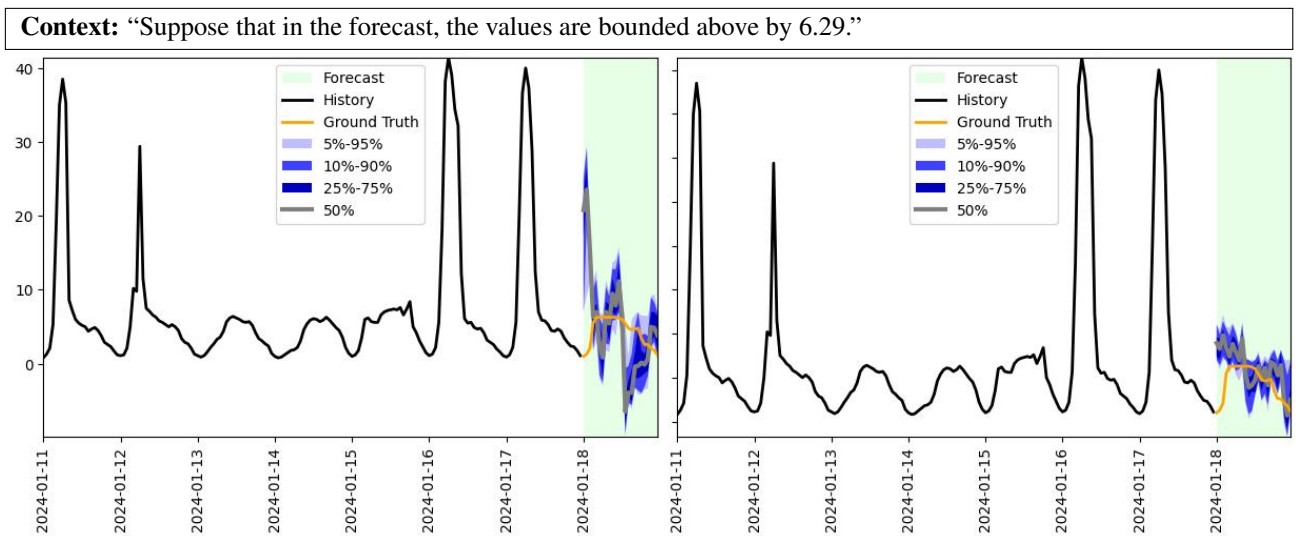

*Figure 29.* A comparison of forecasts from `UniTime` without context (left) and with context (right). On average across 5 instances, `UniTime`'s RCRPS is 64% better with context than without on the "Bounded Prediction Constraint Based On Prediction Quantiles" task.

### D.8. `TimeGEN`

We access `TimeGEN-1`, an optimization of the `TimeGPT` model (Garza et al., 2023), using the API made available through the `nixtla` Python package. Unlike all other baselines, we only generate point forecasts with `TimeGEN` due to its probabilistic mode requiring much longer historical data than is available in instances evaluated in the benchmark. This is the reason the RCPRS values for `TimeGEN` have zero standard error.

### D.9. `Exponential Smoothing`

We used the `Exponential Smoothing` implementation from the `statsmodels` Python package, namely the `statsmodels.tsa.holtwinters.ExponentialSmoothing` class. Both trend and seasonal components of the models are set to be additive. The seasonal period length is either set manually for tasks where the simple guess using the time series frequency is incorrect. If there is not at least two full seasonal periods in the history window of the time series, we disable the seasonal component of the model. Since some of the benchmark tasks can have as few as 3 time steps in the history window, we also disable the trend component if we have less than 5 time steps in said window.

### D.10. `ETS` and `ARIMA`

We used the implementations of `ETS` and `ARIMA` from the `forecast` R package, using `rpy2` for compatibility with Python. For `ETS`, we use the `ets` method, which we call with automatic error, trend, and seasonality components. In the rare cases where the `ETS` forecast contains NaN values, we manually switch off the trend component and rerun the forecast. The `ARIMA` results are computed using the `auto.arima` method. If the `ARIMA` fits fail, we rerun it with restricted parameter and disabled seasonality.

## E. Details of the proposed metric

The CiK benchmark is designed to determine whether models can improve their probabilistic forecasts by leveraging associated textual information (see Sec. 2). To support this goal, the evaluation metric:

1. should be a **proper scoring rule**, such that a model who perfectly knows what the correct forecast is should have no reason to favor another prediction;

2. must be **easy to compute** using a finite sample from the forecast distribution, since many models do not provide a functional form for their forecasts.

To account for the importance of leveraging relevant context, the metric should also:

1. **penalize obviously impossible forecasts**, i.e. that can be inferred as implausible from the contextual information;

2. **take a similar range of values across different tasks**, to prevent some tasks to dominate the score as we average the results across tasks;

3. **prioritize forecast quality for timesteps with relevant context**, even if these timesteps are a small portion of the forecast horizon.

To satisfy the first two properties, we start with the Continuous Ranked Probability Score (CRPS) (Gneiting & Raftery, 2007), a reliable strictly proper scoring rule for univariate probability distribution, and take its mean over all time steps. To compute the CRPS from a finite number of samples, we use the estimator based on its probability weighted moment form (Taillardat et al., 2016), since it is unbiased (Zamo & Naveau, 2018). See Appendix E.3 for more details about this estimator.

Many of our tasks are built to include information about a hard constraint on $\mathbf{X}_F$ in their $\mathcal{C}$, which can be written as $v_{\mathcal{C}}(\mathbf{x}_F) = 0$. If we were only interested to measure by how much a forecast breaks the constraint, we could take inspiration from the threshold-weighted CRPS (Gneiting & Ranjan, 2011) by using $v_{\mathcal{C}}$ as its chaining function (Allen et al., 2023):

$$\text{twCRPS}_{v_{\mathcal{C}}}(\widetilde{\mathbf{X}}_F, \mathbf{x}_F) \equiv \text{CRPS}\left(v_{\mathcal{C}}(\widetilde{\mathbf{X}}_F), v_{\mathcal{C}}(\mathbf{x}_F)\right), \tag{1}$$

where $\widetilde{\mathbf{X}}_F$ is the forecast of $\mathbf{X}_F$ to be evaluated. Since, by construction, the ground-truth $\mathbf{x}_F$ always satisfy the constraints, we have $v_{\mathcal{C}}(\mathbf{x}_F) = 0$. But since we do not care only about whether forecasts break constraints, we sum both the original CRPS and this twCRPS, but we weight the later by a factor of $\beta = 10$, to denote the additional interest we show to these errors. See Appendix E.4 for the various $v_{\mathcal{C}}$ used in the benchmark.

One common approach to normalize the CRPS to get similar ranges for multiple problems is to divide it by the mean absolute value of the target ground-truth of the forecasted series (Alexandrov et al., 2020). This has two issues: the metric is no longer proper, and it leads to much larger values for series close to zero than those far from it. To solve the first issue, we take advantage that we can generate many more instances from each of our tasks, by computing a normalization factor $\alpha$ from 25 instances not included in the benchmark. The details of this calculations are in Appendix E.1.

Many tasks in our benchmark contains contextual information which is highly relevant for a small fraction of the time steps in the forecasting window, while being only marginally relevant for the majority of the time steps. If we were to weight these two categories equally, then the score for a model which ignores the context would be hard to distinguish from the score of one who does not. We correct this issue by identifying the subset of time steps with relevant information, which we call the Region of Interest (RoI). We then weight the CRPS to give half weight to the RoI time steps and half weight to the non-RoI time steps. Therefore, we obtain our metric, which we call the Region-of-Interest CRPS (RCRPS):

$$\text{RCRPS}(\widetilde{\mathbf{X}}_F, \mathbf{x}_F) := \begin{cases} \alpha \cdot \left[\frac{1}{2|\mathcal{I}|} \cdot \sum\limits_{i \in \mathcal{I}} \text{CRPS}\left(\widetilde{X}_i, x_i\right) + \frac{1}{2|\neg\mathcal{I}|} \cdot \sum\limits_{i \in \neg\mathcal{I}} \text{CRPS}\left(\widetilde{X}_i, x_i\right) + \beta \cdot \text{CRPS}\left(v_{\mathbf{C}}(\widetilde{\mathbf{X}}_F), 0\right)\right] & \text{if } |\mathcal{I}| > 0 \\ \alpha \cdot \left[\frac{1}{|\neg\mathcal{I}|} \cdot \sum\limits_{i \in \neg\mathcal{I}} \text{CRPS}\left(\widetilde{X}_i, x_i\right) + \beta \cdot \text{CRPS}\left(v_{\mathbf{C}}(\widetilde{\mathbf{X}}_F), 0\right)\right], & \text{if } |\mathcal{I}| = 0 \end{cases}$$

where $\mathcal{I}$ is the set of time steps in the RoI, $\neg\mathcal{I}$ is the set of time steps in the forecast but not in the RoI, $\alpha$ is the aforementioned scaling factor, and we drop the factor of two and the first sum for tasks where there is no meaningful RoI.

### E.1. Scaling for cross-task aggregation

The rationale behind scaling the RCPRS is to allow us to average its value from diverse tasks without the average being dominated by the forecast quality for tasks with time series with large values. An alternative argument is: all other conditions being equal, a forecaster that is wrong by 10 in its forecast for a time series which goes from 25 to 30 is worse than one that is wrong by 100 in its forecast for a time series which goes from 2500 to 3000. Furthermore, we have multiple tasks for which some instances have constant $\mathbf{x}_F$ or nearly so, often with values close to zero. Due to these tasks, we cannot simply use a scaling which only depends on said instances $\mathbf{x}_F$. Instead, we take advantage of our benchmark ability to generate a very large number of instances for each tasks by using $M = 25$ instances not included in our benchmark. Given

the ground-truth future values $\mathbf{x}_F^m$ for these instance, the scaling factor $\beta$ for an individual task is as follow:

$$\alpha = \left[ \frac{\sum_m \left( \max_i x_i^m - \min_i x_i^m \right)}{M} \right]^{-1}. \tag{2}$$

**Properness** In an ideal scenario, all instances of a tasks would be fully independent. In that case then Equation (2) would not contain any information about the target time series in the benchmark instances, making the RCPRS a proper scoring rule. However, due to possible overlaps in the time windows used when creating the instances and to auto-correlations, we cannot guarantee independence between instances, and thus we cannot guarantee that the RCPRS is actually a proper scoring rule. Note that this deviation from a proper scoring rule is minor, and has a much smaller effect than the one due to the common approach of normalizing the CRPS using the Mean Absolute Value of the ground-truth.

### E.2. CRPS and twCRPS

Given a univariate forecast $\widetilde{X}$ and a ground-truth realization $x$, the Continuous Ranked Probability Score (CRPS) can be defined in its integral as follow:

$$\mathrm{CRPS}(\widetilde{X}, x) = \int_{-\infty}^{\infty} dy \left[ \Phi_{\widetilde{X}}(y) - \mathbb{1}(y \geq x) \right]^2, \tag{3}$$

where $\Phi_{\widetilde{X}}(y)$ is the Cumulative Distribution Function of $\widetilde{X}$, and $\mathbb{1}$ is the indicator function.

There are multiple ways to compute the CRPS, but a particularly interesting one which showcases its link to the Mean Absolute Error is the energy form of the CRPS:

$$\mathrm{CRPS}(\widetilde{X}, x) = \mathbb{E}_{X \sim \widetilde{X}} |X - x| - \frac{1}{2} \mathbb{E}_{X, X' \sim \widetilde{X}} |X - X'|. \tag{4}$$

We get the threshold-weighted CRPS (twCRPS) from Equation (4) by adding a weighting function $w(x)$ to it:

$$\mathrm{twCRPS}(\widetilde{X}, x) = \int_{-\infty}^{\infty} dy w(y) \left[ \Phi_{\widetilde{X}}(y) - \mathbb{1}(y \geq x) \right]^2. \tag{5}$$

To get the energy form of the twCRPS, we must compute the chaining function $v(x)$ from $w(x)$:

$$v(x) - v(x') = \int_{[x, x')} dy w(y). \tag{6}$$

Using $v(x)$, we can write the twCRPS as:

$$\mathrm{twCRPS}(\widetilde{X}, x) = \mathbb{E}_{X \sim \widetilde{X}} |v(X) - v(x)| - \frac{1}{2} \mathbb{E}_{X, X' \sim \widetilde{X}} |v(X) - v(X')|. \tag{7}$$

Equation (7) can readily be generalized to a multivariate forecast, by using any $\mathbb{R}^d \to \mathbb{R}$ chaining function.

### E.3. Estimating the CRPS using samples

Computing the CRPS using Equation (3) or Equation (4) directly would be extremely hard for most of the baselines included in our experiments. Instead, it is more computationally convenient to use an estimator of the CRPS which uses a finite number of samples $x_1, ..., x_M$ from the forecasting distribution. An unbiased estimator of the CRPS created from Equation (4) is:

$$\mathrm{CRPS}(\widetilde{X}, x) \approx \frac{1}{M} \sum_{n=1}^{M} |x_n - x| - \frac{1}{2M(M-1)} \sum_{n=1}^{M} \sum_{n'=1}^{M} |x_n - x_{n'}|. \tag{8}$$

However, this estimator is relatively costly, having a $O(M^2)$ time complexity.

A faster estimator which gives the same result as Equation (8) (up to numerical accuracy) is the one based on the probability weighted moment form of the CRPS (Taillardat et al., 2016; Zamo & Naveau, 2018):

$$\mathrm{CRPS}(\widetilde{X}, x) \approx \frac{1}{M} \sum_{n=1}^{M} |x_n - x| + \frac{1}{M} \sum_{n=1}^{M} x_n - \frac{2}{M(M-1)} \sum_{n=1}^{M} (n-1) x_n, \tag{9}$$

where the $x_n$ have been sorted in ascending order. We used Equation (9) in our metric, since it is as accurate as Equation (8), while only having a $O(M \log M)$ time complexity.

### E.4. Constraint-violation functions

In selecting constraint-violation functions $v_{\mathcal{C}}$ for our various tasks, we have the following requirements: it should be invariant to the number of timesteps in the forecasting window and it should be multiplied by $\alpha$ if all numerical data in a task is transformed using $x \to \alpha x + \beta$. Here are the $v_{\mathcal{C}}$ we use in some of our benchmark tasks:

- *Constant upper-bound constraint $x_i \leq \tau^+$:*

$$v_{\mathcal{C}}(\mathbf{x}_F) = \frac{1}{T-t} \sum_{i=t+1}^{T} \max(0, x_i - \tau^+),$$

- *Constant lower-bound constraint $x_i \geq \tau^-$:*

$$v_{\mathcal{C}}(\mathbf{x}_F) = \frac{1}{T-t} \sum_{i=t+1}^{T} \max(0, \tau^- - x_i),$$

- *Constant lower-bound and upper-bound constraints $\tau^- \leq x_i \leq \tau^+$:*

$$v_{\mathcal{C}}(\mathbf{x}_F) = \frac{1}{T-t} \sum_{i=t+1}^{T} \max(0, \tau^- - x_i) + \max(0, x_i - \tau^+),$$

- and *variable upper-bound constraints, on a subset of time steps $x_i \leq \tau_i^+ \ \forall i \in C$:*

$$v_{\mathcal{C}}(\mathbf{x}_F) = \frac{1}{|C|} \sum_{i \in C} \max(0, x_i - \tau_i^+).$$

### E.5. Covariance of two CRPS estimators

One approach to compute standard error on the RCRPS is to compute the empirical standard deviation based on the 5 instances we use for each task. However, such a method would overestimate the standard error, since it would consider both the variance coming from the selection of instances of a given task, and the variance coming from the models sampling processes. Since all models are tested using the exact same instances, the variance coming from their selection is not relevant, and thus we need a way to ignore it.

To do so, we take advantage that the RCRPS is a weighted sum of multiple CRPS estimates. Since those estimates are not independent from one another, we can compute an estimate of the variance of the RCPRS under the sampling process by computing an estimate of the covariance matrix between the various CRPS estimates, followed by the appropriate weighted sum.

Let says we want to compute the covariance between the CRPS for variable $i$ and the CRPS for variable $j$, using $M$ independent and identically distributed samples from the joint distribution of $\widetilde{X}_i$ and $\widetilde{X}_j$.

$$\text{Cov}\left(\text{CRPS}\left(\widetilde{X}_i, x_i\right), \text{CRPS}\left(\widetilde{X}_j, x_j\right)\right) =$$
$$\text{Cov}\left(\frac{1}{M} \sum_n |\widetilde{X}_{i,n} - x_i| - \frac{1}{2M(M-1)} \sum_{n \neq n'} |\widetilde{X}_{i,n} - \widetilde{X}_{i,n'}|,\right.$$
$$\left.\frac{1}{M} \sum_n |\widetilde{X}_{j,n} - x_j| - \frac{1}{2M(M-1)} \sum_{n \neq n'} |\widetilde{X}_{j,n} - \widetilde{X}_{j,n'}|\right),$$

where the sums are over the various samples $n$ and $x_i$ and $x_j$ are the ground-truth values.

After some tedious algebraic manipulations, we obtain the final formula for the covariance of two CRPS estimates:

$$\text{Cov}\left(\text{CRPS}\left(\widetilde{X}_i, x_i\right), \text{CRPS}\left(\widetilde{X}_j, x_j\right)\right) = -\frac{1}{M} \mathop{\mathbf{E}}_{\widetilde{X}_i} |\widetilde{X}_i - x_i| \mathop{\mathbf{E}}_{\widetilde{X}'_j} |\widetilde{X}'_j - x_j|$$

$$+ \frac{1}{M} \mathop{\mathbf{E}}_{\widetilde{X}_i} |\widetilde{X}_i - x_i| \mathop{\mathbf{E}}_{\widetilde{X}'_j \widetilde{X}''_j} |\widetilde{X}'_j - \widetilde{X}''_j|$$

$$+ \frac{1}{M} \mathop{\mathbf{E}}_{\widetilde{X}_i \widetilde{X}'_i} |\widetilde{X}_i - \widetilde{X}'_i| \mathop{\mathbf{E}}_{\widetilde{X}''_j} |\widetilde{X}''_j - x_j|$$

$$- \frac{2M - 3}{2M(M-1)} \mathop{\mathbf{E}}_{\widetilde{X}_i \widetilde{X}'_i} |\widetilde{X}_i - \widetilde{X}'_i| \mathop{\mathbf{E}}_{\widetilde{X}''_j \widetilde{X}'''_j} |\widetilde{X}''_j - \widetilde{X}'''_j|$$

$$+ \frac{1}{M} \mathop{\mathbf{E}}_{(\widetilde{X}_i, \widetilde{X}_j)} |\widetilde{X}_i - x_i| \cdot |\widetilde{X}_j - x_j|$$

$$- \frac{1}{M} \mathop{\mathbf{E}}_{(\widetilde{X}_i, \widetilde{X}_j)} \mathop{\mathbf{E}}_{\widetilde{X}'_i} |\widetilde{X}_i - \widetilde{X}'_i| \cdot |\widetilde{X}_j - x_j|$$

$$- \frac{1}{M} \mathop{\mathbf{E}}_{(\widetilde{X}_i, \widetilde{X}_j)} \mathop{\mathbf{E}}_{\widetilde{X}'_j} |\widetilde{X}_i - x_i| \cdot |\widetilde{X}_j - \widetilde{X}'_j|$$

$$+ \frac{1}{2M(M-1)} \mathop{\mathbf{E}}_{(\widetilde{X}_i, \widetilde{X}_j)} \mathop{\mathbf{E}}_{(\widetilde{X}'_i, \widetilde{X}'_j)} |\widetilde{X}_i - \widetilde{X}'_i| \cdot |\widetilde{X}_j - \widetilde{X}'_j|$$

$$+ \frac{M - 1}{M(M-1)} \mathop{\mathbf{E}}_{(\widetilde{X}_i, \widetilde{X}_j)} \mathop{\mathbf{E}}_{\widetilde{X}'_i \widetilde{X}''_j} |\widetilde{X}_i - \widetilde{X}'_i| \cdot |\widetilde{X}_j - \widetilde{X}''_j|,$$

where variables with the same number of apostrophes ($'$) are drawn together and those with different number of apostrophes are independent variables.

To get an estimate of the covariance using our $M$ samples, we can estimate each of these terms using their respective unbiased estimators. Once we have computed an estimate of the variance for a single task instance, the overall variance for a full task is computed using the formula for the variance of the average of multiple independent variables. One slight disadvantage of using this method is that it offers no guarantee that the RCPRS variance estimate will be non-negative, so in the rare cases where the estimate for the variance of a full task is negative, we clip it to 0.

### E.6. Comparison of Statistical Properties of Various Scoring Rules

*Table 12.* Comparison of Statistical Properties of Various Scoring Rules. The * indicates that, to be proper, it would need that different seeds are independent, which cannot be guaranteed by CiK, but could happen in other applications.

| Metric | Proper Scoring Rule | Domain | Invariance | |
| --- | --- | --- | --- | --- |
| | | | Additive | Multiplicative |
| Brier Score | Yes | Discrete | Yes | Yes |
| CRPS | Yes | Continuous | Yes | No |
| TwCRPS | Yes | Continuous | Yes | No |
| CRPS skill score | No | Continuous | Yes | Yes |
| MAV-Scaled CRPS | No | Continuous | No | Yes |
| RCRPS | No* | Continuous | Yes | Yes |

Tab. 12 describes a few statistical properties for both commonly used scoring rules and our RCPRS. For the invariance (additive and multiplicative) properties, we indicate whether the scoring rule remains unchanged if all relevant quantities (forecast, ground truth, threshold, and constraint parameters) are modified by adding a constant to them or by multiplying them by a constant. By MAV-Scaled CRPS, we denote the common approach in the forecasting literature to normalize the CRPS by dividing it by the Mean Absolute Values of the ground-truth, instead of reporting the original CRPS values.

