# OpenReview forum: "Context is Key: A Benchmark for Forecasting with Essential Textual Information"
_ICML.cc/2025/Conference — ICML 2025 poster_

### Official Review · Reviewer_hoT6 · 2025-02-14

**Overall Recommendation:** 3

**Summary:**

This paper introduces the "Context is Key" benchmark to evaluate the capability of models in leveraging textual information for time series forecasting. By designing 71 tasks and evaluating them with both human and large language model (LLM) annotators, the study confirms the significant role of contextual information in enhancing prediction accuracy, especially during relevant time periods. The authors propose a novel evaluation metric combining CRPS and twCRPS to more accurately assess model performance. Experimental results show that providing contextual information significantly improves forecasting quality, offering valuable insights for future developments in multimodal forecasting and automated predictive systems.

**Claims And Evidence:**

This paper provides evidence supporting the significant role of contextual information in time series forecasting by designing 71 tasks and evaluating them with both human and large language model (LLM) annotators. It introduces a novel evaluation metric (CRPS and twCRPS) to more accurately measure prediction performance. Experimental results show that providing contextual information improves forecasting quality, demonstrating the effectiveness of this approach. Overall, the main findings of the paper are clear and convincing, highlighting the critical importance of context in forecasting tasks.

**Essential References Not Discussed:**

I believe the related work section of this paper is sufficient.

**Experimental Designs Or Analyses:**

The experiments in this paper are relatively comprehensive. However, I have reservations about the training approach for UniTime and Time-LLM. Such a stringent comparison setup fails to fully leverage the strength of the baseline models and may lead to an overestimation of the role of event text. Furthermore, if the authors could include some multimodal time series forecasting baselines, such as dataset-specific models (e.g. TimeMMD) and foundational models (e.g. ChatTime), it would make the work even more complete.

**Methods And Evaluation Criteria:**

This paper collects a large number of datasets and designs reasonable evaluation metrics, filling a gap in the field.

**Other Comments Or Suggestions:**

Refer to the above comments.

**Other Strengths And Weaknesses:**

Refer to the above comments.

**Questions For Authors:**

Refer to the above comments.

**Relation To Broader Scientific Literature:**

This paper further advances the development of multimodal time series forecasting.

**Theoretical Claims:**

I have reviewed the correctness of the theoretical proofs in this paper.

---

> ### Author Rebuttal · Authors · 2025-04-01
>
> Thank you for your thoughtful response. We appreciate your recognition of our work as addressing a gap in the field, offering valuable insights for future developments, and providing clear and convincing findings. We address your concerns below and are happy to clarify any further points.
>
> ---
>
>
> ## ***Training Approach of Time-LLM and UniTime***
>
> Thank you for your questions regarding the training approaches for Time-LLM and UniTime. We included Time-LLM and UniTime in our evaluation due to their respective contributions to multimodal forecasting. These models require training to adapt them for context-aided forecasting, however CiK is an evaluation benchmark, and has no training set.  Therefore, we rely on the original papers’ training recipes to train their models, which adapt them for specific time series datasets with paired templates of context text. We discuss these limitations of the models in Appendix D.3 (“Why Do Time-LLM and UniTime Not Benefit (More) From Context?”). To make this more clear, we propose to:
>
> - \[**Expanded Results Discussion**\] Add to the main text an abridged version of the discussion in Appendix D.3, which describes the limitations we faced when evaluating Time-LLM and UniTime.
>
> Finally, if you can suggest a more appropriate training recipe, we are happy to rerun the models and report the results in the camera-ready version.
>
> ---
>
> ## ***Stringent comparison setups for Time-LLM and UniTime, and potential overestimation of the importance of event text***
>
> Thank you for pointing this out. We acknowledge the reviewer’s claim that the comparison setups for Time-LLM and UniTime are stringent, and may fail to fully leverage the strength of the baseline models. We attribute this to the training approaches of the respective models (described above), which adapt the baseline models’ capabilities to process contextual text in highly specialized templates, and for specific time series domains. CiK is however intended to evaluate context-aided forecasting across several time series domains and types of textual context, which is a different, more broader setup compared to those used in the papers of Time-LLM and UniTime. The training dataset of the models would have to be much more diverse to enable these models to generalize well enough to perform well on CiK, which contains linguistic variations and requires many types of reasoning. To make this more clear, we propose to
>
> - \[**Addition to Limitations**\] Add to limitations the impact of the lack of training datasets when evaluating dataset-specific approaches, such as UniTime and Time-LLM.
> - \[**Addition to Future Work**\] Add to future work the need for training datasets for context-aided forecasting.
>
> ---
>
> ## ***Additional baselines***
>
> \[**TimeMMD**\] We contacted the authors of [Time-MMD](https://arxiv.org/abs/2406.08627) for access to checkpoints of models from their paper, but have yet to hear back. Unfortunately, we found that their paper does not contain the hyperparameters used to train these models and thus we could not recover their checkpoints ourselves. We will however add the results of these models to the paper if we receive the checkpoints or manage to reproduce their original results.
>
> \[**ChatTime**\] Thank you for these suggestions. We evaluate the publicly available checkpoints ([Base](https://huggingface.co/ChengsenWang/ChatTime-1-7B-Base), [Chat](https://huggingface.co/ChengsenWang/ChatTime-1-7B-Chat)) of [ChatTime](https://arxiv.org/abs/2412.11376) on CiK. Here are the aggregate results:
>
> | Model | Average RCRPS (± std) |
> |---|--|
> |**With Context**||
> | ChatTime-Base | 0.735 ± 0.002 |
> | ChatTime-Chat | 0.747 ± 0.005 |
> |**Without Context**||
> | ChatTime-Base | 0.725 ± 0.002 |
> | ChatTime-Chat | 0.781 ± 0.015 |
>
> Compared to other models evaluated on CiK (shown in Table 1), these results (with and without context) are not competitive and are, in fact, much worse than those of statistical models that cannot process context. ChatTime-Base shows a very small improvement with context, while ChatTime-Chat degrades with context. This is likely because ChatTime’s training dataset is limited to 3 time series datasets with highly specific contextual text templates. Similar to Time-LLM and UniTime, the training approach likely fails to generalize outside its training domains, and fails to leverage the strength of the baseline models. We will add these results to the paper.

---

> > ### Comment · Reviewer_hoT6 · 2025-04-02
> >
> > I appreciate the author's patient response and detailed experiments, which largely addressed my doubts. Considering the limitations of the baseline training approach, I will maintain my score.  Additionally, to my knowledge, similar to TimeLLM, TimeMMD also requires separate training for each dataset. Perhaps you could include it as a baseline following the training approach of TimeLLM and UniTime.

---

### Official Review · Reviewer_F7ED · 2025-03-13

**Overall Recommendation:** 4

**Summary:**

The paper introduces Context is Key (CiK), which is a benchmark aiming to evaluate forecasting models’ ability to integrate both numerical time-series data and essential textual context. Unlike traditional forecasting benchmarks that rely solely on numerical data, CiK explicitly requires models to process and leverage natural language context to improve prediction accuracy. CiK consists of 71 manually designed forecasting tasks spanning seven real-world domains (climatology, economics, energy, mechanics, public safety, transportation, and retail). The authors evaluate statistical models, time-series foundation models, and large language model-based forecasters. Notably, the authors introduced a new forecasting metric, RCRPS, to specifically evaluate forecasts when context is relevant.

**Claims And Evidence:**

Firstly, the authors explicitly designed various tasks where textual context is necessary to make accurate forecasts. And with human and LLM-based evaluations, it further confirmed that with contextual guidance, the forecast quality is improved. The experiments are comprehensive and detailed.

**Essential References Not Discussed:**

Essential references are discussed but several suggested references are listed in Suggestion section for time series reasoning.

**Experimental Designs Or Analyses:**

The paper’s experimental designs are well-structured and comprehensive, which are one of the key strengths. The paper evaluates a wide range of models, including Statistical Models (ARIMA, ETS, Exponential Smoothing), Time-Series Foundation Models (Lag-Llama, Chronos, Moirai, TimeGEN), LLMs (GPT-4o, Llama-3, Mixtral-8x7B, Qwen-2.5), Multimodal Models (UniTime, Time-LLM). This allows for a fair and meaningful comparison between traditional, modern, and context-aware forecasting methods. Additionally, the authors conducted detailed ablation studies. For example, in figure 5, it shows that removing textual context significantly reduces forecasting accuracy. And the authors also designed a new evaluation metrics specifically for context-aware forecasting. They also included various examples tasks, which would be really good for further implementation.

**Methods And Evaluation Criteria:**

The experiments in this paper are exceptionally well-designed and detailed, effectively filling a critical gap in the literature by providing time series data paired with contextual information—a resource that has been largely missing until now. Additionally, the authors introduce a new evaluation metric, RCRPS, which prioritizes context-sensitive forecast accuracy. This ensures that models are assessed based on their ability to effectively integrate and utilize textual context, making the evaluation more meaningful and relevant to real-world forecasting scenarios.

**Other Comments Or Suggestions:**

In addition to citing Merrill et al.’s work on the challenges large language models (LLMs) face in zero-shot time-series reasoning, several other studies have explored this area:

1. “Towards Time-Series Reasoning with LLMs”: they propose a novel multi-modal approach that integrates a lightweight time-series encoder with an LLM.

2. “Beyond Forecasting: Compositional Time Series Reasoning for End-to-End Task Execution”: they introduce a program-aided inference agent that leverages LLMs’ reasoning capabilities to decompose complex time-series tasks into structured execution pipelines.

3. “Implicit Reasoning in Deep Time Series Forecasting”: This study delves into how deep learning models implicitly perform reasoning during time-series forecasting, shedding light on the internal mechanisms that contribute to their predictive capabilities.

4. “XForecast: Evaluating Natural Language Explanations for Time Series Forecasting”: The authors of this paper focus on the generation and evaluation of natural language explanations accompanying time-series forecasts, aiming to enhance the interpretability and transparency of predictive models.

5. “Position: Empowering Time Series Reasoning with Multimodal LLMs”: This work advocates for the integration of multimodal data—such as textual descriptions, visual data, and audio signals—with LLMs to bolster time-series reasoning.

**Other Strengths And Weaknesses:**

The paper is among the first to systematically evaluate how textual context improves time-series forecasting. The CiK benchmark fills a major gap in the literature by providing a dataset where context is essential. The proposed evaluation metric (RCRPS) is specifically designed for context aware. The findings have practical implications for industries like finance, energy, and public safety, where human forecasters already use external knowledge to refine numerical predictions. The paper is well-structured and clearly written, with detailed explanations of the benchmark design, evaluation methods, and experimental setup. The supplementary material provides extensive details on dataset construction, evaluation metrics, and failure cases, which ensures reproducibility.

The weakness lies on the lack of theoretical justification. If a theoretical framework could be designed (for example using Bayesian reasoning or causal inference), this would strengthen the argument. RCRPS is well-motivated but not formally compared against other scores (e.g., CRPSS, Brier Score) in terms of statistical properties.

**Questions For Authors:**

1. Did you experiment with any alignment strategies to help LLMs better process time-series data?
2. Could the lower performance of certain LLM-based forecasting approaches be attributed to the fact that LLMs struggle to natively process time-series sequences?

**Relation To Broader Scientific Literature:**

This paper is one of the early contributions in the emerging field of integrating contextual information with time-series forecasting, primarily due to the lack of paired datasets in this domain. It effectively fills a gap by introducing a benchmark specifically designed to test models’ ability to incorporate textual context into forecasting. Additionally, the lack of appropriate evaluation metrics has been a significant challenge in this area. The authors address this by proposing Region of Interest CRPS, which has the potential to enhance the evaluation of context-aware forecasting models.

**Theoretical Claims:**

The paper does not include formal theoretical proofs to support its claims about context-aided forecasting. Instead, it relies primarily on empirical evidence to demonstrate that textual context improves forecasting accuracy. While RCRPS is intuitively appealing, the paper does not formally prove that it is optimal metric for context-aware forecasting.

---

> ### Author Rebuttal · Authors · 2025-04-01
>
> Thank you for your thoughtful response. We are grateful that you highlighted the thoroughness of our experiments, the value of the RCRPS metric, and the quality of the writing. We are pleased that you see the CiK benchmark as filling a critical gap in the literature, with real-world implications across multiple industries that rely on additional non-numerical information for forecasting.
>
> Below, we present point by point clarifications to your concerns. We are happy to clarify the points further.
>
> ---
>
> ## ***Formal theoretical proofs to support claims about context-aided forecasting***
>
> ### *RCRPS*
>
> We thank the reviewer for highlighting the added value of a formal justification of the RCRPS. We emphasize that we give further details on the RCRPS metric and why it is a good choice of metric for context-aided forecasting in Appendix E. We propose the following:
>
> \[**Addition to Main Text from Appendix E**\] We will include key elements of Appendix E in the main text, including the desiderata for the RCRPS and statistical properties such as properness.
>
> \[**Additional Appendix Section: Statistical Properties**\] We will include in the Appendix a detailed comparison of the statistical properties of different metrics, including the CRPS, CRPSS, twCRPS, Brier Score, and the proposed RCRPS.
>
> ### *Formal characterization of the context-aided forecasting setting*
>
> Thank you for your suggestion regarding a formal characterization of the problem setting of context-aided forecasting. We emphasize that the problem setting is defined formally in Section 2, according to which, we are interested in models where, in expectation, forecasts that leverage context perform better. Critically, all tasks in the benchmark are designed accordingly.
>
> **We seek clarification on this.** We are happy to include aspects of formalization beyond what is discussed in the problem setting. Please let us know what you had in mind.
>
> ---
>
> ## ***Related Work***
>
> \[**Addition to Related Work**\] Thank you for providing these valuable references; we will make sure to reference them all in the revised paper.
>
> ---
>
> ## ***Clarifications to Questions***
>
> \[**Time series alignment strategies for LLMs**\] Thank you for your question. We hope that the analyses we presented are of use:
>
> - \[**Sensitivity to input format and post-training**\] We find that the forecasting performance of the LLMs is sensitive to the input-output format, as shown by differences between the LLMP and DP methods which each prompt the model differently. We also find that the post-training strategy plays a role, as can be evidenced by the gap in performance of models with the DP and LLMP models. Sec 5.3 (“Comparing LLMP and DIRECT PROMPT”) contains more details on both these results. Both these factors could play important roles in the ability of LLMs to process and forecast time series.
> - \[**Addition to Future Work**\] Future work that explores better input/output formatting strategies would benefit the community. As also suggested by reviewer M7ph, the gap between DP and LLMP suggests that prompting strategies might elicit different capabilities too. Further, fine-tuning LLMs on time series data may further adapt them to process time series better. We leave all these directions to future work.
>
> \[**Impact of the ability of LLMs to natively process time series**\] Thank you for this insightful question.
>
> - **\[Comparison of performance of LLMs to other models in the no-context setup\]**  We compare the performance of LLMs to that of quantitative models that can natively process time series (such as foundation models and statistical models) in a no-context setup where all models consume the same input (only the numerical historical data). These results are presented in App C.1. We find that many of the LLMs (with both DP and LLMP) are competitive and sometimes better than the quantitative models. With that said, as all LLMs consider time series as text, we agree that the lower performance of some LLMs could be due to their inability to “natively” process time series, among other reasons (such as poor forecasting ability). Improving the ability of LLMs to process time series is an active area of research, and we believe CiK can be of high value in this front.
> - \[**Extended discussion of catastrophic failures**\] While successful examples of LLMs generating good forecasts hints at the capability to process time series data as text, the catastrophic failures (Sec 5.4)  may suggest that LLMs are not extremely reliable when processing time series, and more research is needed in understanding when they can fail catastrophically.

---

### Official Review · Reviewer_m7ph · 2025-03-14

**Overall Recommendation:** 3

**Summary:**

This paper introduces "Context is Key" (CiK), a benchmark for time-series forecasting models that incorporate textual context alongside numerical data. The authors design 71 tasks across seven domains where textual information is essential for accurate forecasting, propose a Region of Interest CRPS (RCRPS) evaluation metric, and evaluate statistical models, time series foundation models, and LLM-based forecasters. Their "Direct Prompt" method performs better than other tested approaches, showing the value of textual context while highlighting current limitations.

**Claims And Evidence:**

The paper's main claims are backed by empirical results:
- The authors demonstrate context necessity through their task design, with 95% of instances confirmed by evaluators as benefiting from contextual information.
- Figure 6 shows quantified performance gains from textual context, with larger models seeing significant improvements (67.1% for Llama-3.1-405B-Inst), with statistical significance confirmed in Appendix C.6.
- The Direct Prompt method outperforms other approaches in comprehensive evaluations (Table 1), though no single method excels across all context types.

However, I have concerns about real-world applicability due to the data transformations used to mitigate memorization. The authors apply several techniques described in Appendix A.1, including prioritizing live data sources, using derived time series, and adding noise or shifting timestamps. While these modifications help prevent LLMs from leveraging memorized data during evaluation, they create an artificial evaluation setting that may not reflect performance on unmodified real-world data. For instance, shifting timestamps could disrupt natural seasonal patterns that models might otherwise leverage. The paper claims these transformations are used "sparingly," but provides no quantitative analysis of how extensively they were applied or how they might affect model behavior. This creates a disconnect between benchmark performance and what practitioners might experience in deployment scenarios. The main text should explicitly acknowledge this limitation rather than burying it in an appendix, ideally with ablation studies showing performance differences between original and transformed data where possible.

**Essential References Not Discussed:**

None that I'm aware of.

**Experimental Designs Or Analyses:**

The task creation process addresses data contamination through several methods. The sample size (5 instances × 25 forecasts per task) seems adequate for statistical analysis.

Table 1 shows comprehensive evaluation results with significance testing in the appendix. The failure analysis in §5.4 reveals important model limitations, like GPT-4o struggling with scientific notation.

The decision to clip RCRPS scores for major failures at 5 prevents outliers from dominating results but also hides how badly models can fail. The authors acknowledge this trade-off and include win-rate analyses that aren't affected by clipping.

Figure 7's parameter-efficiency analysis provides practical deployment insights beyond raw performance numbers.

**Methods And Evaluation Criteria:**

The benchmark draws from diverse data sources and includes various context types (intemporal, future, historical, covariate, causal). The authors address memorization issues by applying transformations to the data, though this potentially affects realism.

The RCRPS metric improves on standard metrics by accounting for context-sensitive regions and constraint satisfaction, making it more suitable for evaluating contextual forecasting.

The experiment design uses multiple instances per task and multiple forecasts per instance, with a weighting scheme to handle task similarity clusters. This helps provide statistical reliability.

The model comparison covers different architectural approaches and scales, allowing for meaningful performance comparisons across the forecasting methodology spectrum.

**Other Comments Or Suggestions:**

The paper could benefit from a more systematic analysis of the failture patterns across context types and model architectures. Developing clearer guidelines for matching specific context types to the most appropriate models based on your findings could also be useful. Have you considered extensions to domain-specific ontexts like healthcare or finance? The specialized terminology may present additional challenges for contextual integration.

**Other Strengths And Weaknesses:**

Strengths:

- Manual creation and validation of tasks ensures quality
- Context type taxonomy provides structure for analyzing text-forecast relationships
- Model failure and inference cost analysis offers practical insights

Weaknesses:

- Missing analysis of which linguistic patterns most impact forecast quality
- Limited exploration of computational costs for real-time applications
- One-way focus (text informing forecasts) misses potential bidirectional interactions
- Limited insight into how models could better use contextual information

**Questions For Authors:**

1. What specific aspects of instruction tuning hurt LLMP performance while not affecting Direct Prompt? Does this suggest fundamental differences in how these methods use model capabilities?
2. Besides scientific notation issues, have you found other systematic failure patterns that could guide model selection?
3. Have you explored compression techniques that might preserve contextual integration while reducing computational costs?
4. How might the benchmark extend to structured data formats (databases, tables) common in operational forecasting?
5. Since no method excels across all context types, would you recommend ensemble approaches based on context detection, or do you think unified models could overcome these limitations?

**Relation To Broader Scientific Literature:**

The paper positions itself between numerical approaches (time series foundation models) and language-integrated methods. It acknowledges prior benchmarks (Merrill et al., 2024; Zhang et al., 2023; Liu et al., 2024a) while explaining CiK's focus on essential contextual information.

The Direct Prompt approach builds on prior prompt-based forecasting methods (Requeima et al., 2024; Gruver et al., 2024), though the paper could better discuss architectural innovations in multimodal forecasting.

**Theoretical Claims:**

The paper is mostly empirical. The RCRPS formulation extending CRPS is mathematically sound. The context-aided forecasting problem formalized in probabilistic terms in Section 2 seems correct.

---

> ### Author Rebuttal · Authors · 2025-04-01
>
> Thank you for your thorough feedback. We appreciate the recognition of our contributions, including the comprehensive empirical evaluations, task diversity and quality, the soundness of the proposed metric, and the thoroughness in handling data contaminations. Below, we clarify several points raised in the review.
>
> ---
>
> ## Transformations to prevent data contamination
>
> We acknowledge your concerns. We present the time series domains and the exact transformations used in App A.1. We include a summary table [here](https://raw.githubusercontent.com/anon-forecast/benchmark_report_dev/refs/heads/main/rebuttal_images/data_transformation_table.png).
>
> We’d like to note that a significant portion of our tasks did not require any transformation. And wherever required, mitigation strategies were chosen extremely carefully and applied intelligently (the exact details of which are in App A.1). The tasks where we shifted the date by 24 hours are tasks with solar irradiance data, where the date shift would have minimal impact as the seasonal effects change smoothly over the year. For the tasks to which we added a very small amount of gaussian noise, we visually inspected that the noise barely changed the data.
>
> We’d also like to emphasize that the code uses transformations as an option, and the tasks can be obtained without the transformation too. Nevertheless, we will explicitly point to these transformations and their potential impact in the main text, as suggested. For the camera-ready version, we will also analyze the performance of all models to the sensitivity to the transformations.
>
> ---
>
> ## Failure pattern analysis
>
> - [**Specific linguistic patterns and systematic failures**] We thank the reviewer for this point. The analysis of specific linguistic patterns could yield interesting further insights. More analyses are also required to understand systematic failure specific models (such as for DP - GPT-4o with scientific notation). We propose to add a note on these points in the future work.
> - [**Additional analyses**] We would like to point out that we provide the number of catastrophic failures per model (in App. C.5.). In the camera-ready version, we will add a view of this table broken down by context type and domain.
>
> ---
>
> ## Other comments
>
> Thank you for bringing these aspects to our attention.
>
> - [**Computational costs**] Apart from the pareto analysis which was aimed at this direction (Fig 7), we currently also provide the average inference time of all models per task (App C.4). We propose to further complement this analysis that provides the per-timestep average inference time, enabling estimation of real-world performance. We are open to any other suggestions.
> - [**Bidirectional interactions**] We focus on the forecasting task in CiK, but other tasks that use numerical data to predict text are very relevant as well. We will add this point to future work.
> - [**How models can better use contextual information**] We believe encouraging models to “think” or reason about the context explicitly can be useful here. We also believe fine tuning LLMs on datasets with text and time series would allow them to generalize better to tasks in CiK. We add these directions in our future work section.
> - [**Domain-specific contexts**] Extending CiK to build domain-specific tasks and benchmarks is definitely a valuable direction of future work that will present additional changes in context integration. We will add this point to future work.
> ---
>
> ## Clarifications to other questions
>
> - [**Impact of instruction tuning**] We agree that the differences in the performance of LLMP and DP may point to fundamental differences in how the methods use contextual information. We hypothesize that, while the instruction-tuned models respond better to DP’s direct instructions, the iterative prediction strategy used by LLMP may rely on a base (non instruction-tuned) model’s proper calibration for forecasting, which might be more calibrated (as instruction-tuning might affect calibration as per [the GPT-4 technical report](https://arxiv.org/abs/2303.08774) (Fig 8)). We agree that this requires further investigation and leave it to future work.
> - [**Compression**] We have not tested this but agree that it is a very interesting idea, especially given recent works on LLM-based lossless compression such as [FineZip](https://arxiv.org/abs/2409.17141). We will add this point to future work.
> - [**Structured context**] While we restrain this work to study textual context, the codebase is extensible and natively supports adding more modalities. We agree this is a very interesting direction of future work, that we have already noted in the future work section.
> - [**Context-source specific models**] We believe both approaches, i.e. unified models for context-aided forecasting, and ensemble approaches such as expert-based methods and [learning to defer](https://arxiv.org/abs/1711.06664) merit investigation. We will add this point to future work.

---

### Decision · Program_Chairs · 2025-05-01

**Decision:**

Accept (poster)

**Comment:**

The paper received three reviews, all of which were positive (two weak accepts and one accept). Reviewers commended the paper’s comprehensive experimental design, which spans 71 tasks across seven domains, where textual context is essential for accurate time series forecasting. The paper proposes a region-of-interest variant of the CRPS evaluation metric (RCRPS), and provides a structured evaluation of classical statistical models, time series foundation models, and  LLM-based predictors. Overall, the reviewers recognized this work as one of the early contributions to the emerging area of integrating contextual information into time series forecasting.

The authors provided a rebuttal addressing the reviewers' comments, which were mostly requests for clarification and additional analysis. No reviewer raised further objections or concerns during the discussion phase. The consensus remained positive.

The AC concurs with the reviewers' assessments. This work offers a timely and well-structured benchmark in a rapidly growing area, and makes a valuable contribution by proposing a new metric and systematically evaluating a broad range of models. AC recommends acceptance. The AC would also like to note that a shorter version of this work was presented at NeurIPS 2024 Workshop. After confirming that the workshop paper was under 4 pages and does not violate the dual submission policy outlined at https://icml.cc/Conferences/2025/CallForPapers, AC discussed the matter with the SAC and both agreed that no further action is necessary.